



# GHOST: A globally harmonised dataset of surface atmospheric composition measurements

Dene Bowdalo[1,2*], Sara Basart[1,2*], Marc Guevara[1], Oriol Jorba[1], Carlos Pérez García-Pando[1,3], Monica Jaimes Palomera[4], Olivia Rivera Hernandez[4], Melissa Puchalski[5], David Gay[6], Jörg Klausen[7], Sergio Moreno[2], Stoyka Netcheva[2,8*], and Oksana Tarasova[2]

[1]Barcelona Supercomputing Center, Barcelona, Spain
[2]World Meteorological Organization (WMO), Geneva, Switzerland
[3]Catalan Institution for Research and Advanced Studies (ICREA), Barcelona, Spain
[4]Secretaría del Medio Ambiente de la Ciudad de México (SEDEMA), Mexico City, Mexico
[5]Environmental Protection Agency (EPA), Washington D.C., United States
[6]National Atmospheric Deposition Program (NADP), Wisconsin State Laboratory of Hygiene, Madison, United States
[7]Federal Office of Meteorology and Climatology MeteoSwiss, Zurich, Switzerland
[8]Environment and Climate Change Canada (ECCC), Toronto, Canada
[*]Current affiliation

**Correspondence:** Dene Bowdalo (dene.bowdalo@bsc.es)

**Abstract.**

GHOST: Globally Harmonised Observations in Space and Time, represents one of the biggest collection of harmonised measurements of atmospheric composition at the surface. In total, 7,275,148,646 measurements from 1970-2023, of 227 different components, from 38 reporting networks, are compiled, parsed, and standardised. Components processed include gaseous species, total and speciated particulate matter, and aerosol optical properties.

The main goal of GHOST is to provide a dataset that can serve as a basis for the reproducibility of model evaluation efforts across the community. Exhaustive efforts have been made towards standardising almost every facet of provided information from the major public reporting networks, saved in 21 data variables, and 163 metadata variables. Extensive effort in particular is put towards the standardisation of measurement process information, and station classifications. Extra complementary information is also associated with measurements, such as metadata from various popular gridded datasets (e.g. land use), and temporal classifications per measurement (e.g. day / night). A range of standardised network quality assurance flags are associated with each individual measurement. GHOST own quality assurance is also performed and associated with measurements. Measurements prefiltered by some default GHOST quality assurance are also provided.

In this paper, we outline all steps undertaken to create the GHOST dataset, and give insights and recommendations for data providers based on experiences gleaned through our efforts.

The GHOST dataset is made freely available via the following repository: https://doi.org/10.5281/zenodo.10637449 (Bowdalo, 2024).



## 1 Introduction

The 20th century bore witness to a revolution of scientific understanding in the atmospheric composition field. In the early
1950's, ozone ($O_3$) was identified as the key component of photochemical smog in Los Angeles (Haagen-Smit, 1952), and
sulphur dioxide ($SO_2$) was identified as the key component of the "London smog" (Wilkins, 1954). These findings instigated a
number of clean air laws to be implemented in the most developed regions of the world (e.g. UN, 1979), and with it an explosion
of monitoring activity, with measuring networks created to continuously measure the concentrations of key components. Over
the next few decades the importance of particulate matter (PM) as a pollutant became better understood (Whitby et al., 1972;
Liu et al., 1974; Hering and Friedlander, 1982). However, it took until the 1980s and 1990s respectively for PM exposure to be
more rigorously monitored via aerodynamic size fractions, namely PM10 and PM2.5 (Cao et al., 2013).

In the present day we know of hundreds of atmospheric components which act as pollutants to human and plant health
(Monks et al., 2015; Mills et al., 2018; Agathokleous et al., 2020; Vicedo-Cabrera et al., 2020), and 100s more which directly
or indirectly affect the concentration of these components. Furthermore, some of these pollutants impact climate forcings in
some capacity via direct, semi-direct, and indirect effects (Forster et al., 2021).

A critical approach for our understanding of the complex, non-linear processes which control the concentration levels of
components in the atmosphere, is through the use of Chemical Transport Models (CTMs) and Earth System Models (ESMs).
In order to evaluate the veracity of these models, observations are required. Unfortunately, the limited availability and quality
of these observations serves as a major impediment to this process. From the 1970s onwards, atmospheric components have
been extensively measured around the world by long-term balloon borne measurements (Tarasick et al., 2010; Thompson et al.,
2015), suitably equipped commercial aircraft (Marenco et al., 1998; Petzold et al., 2015), research aircraft (Toon et al., 2016;
Benish et al., 2020), ships (Chen and Siefert, 2003; Angot et al., 2022), and satellites (Boersma et al., 2007; Krotkov et al.,
2017). However, each of these measurement types carry drawbacks associated with the temporal, horizontal or vertical resolu-
tion of measurements. Near global coverage by satellites exist for some components (e.g. CO, $NO_2$), but these require complex
corrections, and can not yet isolate concentrations at the surface (Kang et al., 2021; Pseftogkas et al., 2022), the air most rele-
vant for humans and vegetation. The most temporally consistent measurements have been made at the surface by established
measurement networks, although the spatial coverage of these measurements is typically limited, being predominately located
in the most developed regions.

The ultimate purpose for measurements at in situ surface stations are wide ranging, from providing information regarding
urban air quality exceedances, to monitoring long term trends, or simply for the purpose of advancing scientific understanding
of atmospheric composition. Owing to this, numerous different institutions or networks manage the reporting of this informa-
tion, meaning information is reported in a plethora of different formats and standards. As a consequence, the aggregation and
harmonisation of both data and metadata, from across these networks, requires extensive effort.

Efforts at synthesising measurements across surface networks have been previously made, but these have often been limited
to a single compound of interest, e.g. $O_3$ (Sofen et al., 2016; Schultz et al., 2017). The AeroCom project represents one of
the most complete efforts at creating a model evaluation framework, harmonising both measurements (from satellites, and



surface) and model output, although this project is solely limited to aerosol components (Kinne et al., 2006; Gliß et al., 2021). The Global Aerosol Synthesis and Science Project (GASSP) is another project that has made efforts at harmonising global aerosol measurements, in this case from the surface, ships, and aircrafts (Reddington et al., 2017). An interesting approach

to overcome the limited spatial coverage of surface observations, has been to create synthetic gridded observations (Cooper et al., 2020; van Donkelaar et al., 2021), by combining satellite data with CTM output, and calibrating to surface observations, although naturally this approach comes with significant uncertainties. There are existing efforts which parse near real time surface measurements globally (IQAir; OpenAQ; WAQI), or citizen science project utilising low-cost sensors (PurpleAir; UN Environment Programme). However, these efforts are typically more tailored for public awareness purposes than for actual

science, with little to no quality control procedures, limited historical extent (maximum of ∼5 years), and a limited number of processed components. Rather than harmonising existing datasets, there have been other efforts to create universal standards to which measurement stations can comply with. The World Meteorological Organization (WMO) (WMO, b, c, d) have made significant efforts through the WMO Integrated Global Observing System (WIGOS) (WMO, 2019a, 2021) framework to this purpose. The Aerosol, Clouds and Trace Gases Research Infrastructure (ACTRIS) (ACTRIS), and EBAS (NILU) are two other

examples of efforts to create extensive reporting standards. The number of measurement stations following these standards however represents a small fraction of those available globally.

There have been numerous model evaluation studies which utilise data from one or more surface measurement networks. However, there is typically little to no detail given about the methodology used in combining data / metadata from across different networks, the quality assurance (QA) applied to screen measurements, and the station classifications employed to

subset stations (e.g. Colette et al., 2011; Solazzo et al., 2012; Katragkou et al., 2015; Schnell et al., 2015; Badia et al., 2017). Therefore evaluation efforts from different groups are often incomparable, and non-reproducible.

In response to this, we established GHOST: Globally Harmonised Observations in Space and Time. The main goal of GHOST is to provide a dataset of atmospheric composition measurements, that can serve as a basis for the reproducibility of model evaluation efforts across the community. Exhaustive efforts are made towards standardising almost every facet of

provided information from the major public reporting networks that provide measurements at the surface. Unlike other major synthesis efforts, no data is screened out. Rather, each measurement is associated with a number of standardised QA flags, providing users a way to flexibly subset data. Although this work focuses on surface based measurements, GHOST was designed to be extensible, both to more surface network data, as well as the incorporation of other types of measurements, e.g. satellite, aircraft.

This paper fully details the processing procedures that have resulted in the GHOST dataset. In Sect. 2 of this paper we outline the reporting networks contributing to this work. Section 3 details the processing used to transform native network data to the finalised GHOST dataset. Section 4 describes the temporal and spatial extent of the finalised dataset. Finally, Sect. 5 gives some insights and recommendations for data providers based on experiences gleaned through this work.



## 2   Contributing datasets

GHOST ingests data from the 38 networks listed in Table 1. 227 atmospheric components, across 13 distinct component types (or matrices), are processed per network. These matrices serve as a way of being able to more simply classify the many types of components, and are specifically: gas (all gas-phase components), pm (all particulate matter), pm10 (particulate matter with a diameter <= 10$\mu$m), pm2.5 (particulate matter with a diameter <= 2.5$\mu$m), pm1 (particulate matter with a diameter <= 1$\mu$m), aod (aerosol optical depth), extaod (extinction aerosol optical depth), absaod (absorption aerosol optical depth), ssa (aerosol

single scattering albedo), asy (aerosol asymmetry / sphericity factors), rin (aerosol refractive indices), vconc (aerosol total volume concentration), and size (aerosol size distribution). The components processed within GHOST are outlined per matrix in Table 2, with more detailed information given per component in Table A3.

It is important to state, the term "network" is used loosely through this work. Many of the "networks" that data are sourced from could be better classified better as "projects", "frameworks" or "reporting mechanisms". However, for the purposes of

simplicity, we define "network" to be the most common name of an available dataset, from a specific data source. For WMO data for example, this means what is typically called the Global Atmosphere Watch Programme (GAW) network, is separated out across 3 networks, as the data is reported in a discretised form, across 3 data centres.

The geographic coverage of the contributing networks range from the global to sub-national scale. The operational objectives of the networks are wide ranging, with some of the networks setup to monitor the background concentrations of atmospheric

components in rural areas (e.g. US EPA CASTNET), whereas others exist for regulatory purposes, monitoring compliance with national or continental air quality limits (e.g. EEA AQ e-Reporting). Many of the networks have substantive, well documented internal QA programs.

We recognise that the datasets ingested in GHOST do not represent all of the observations of atmospheric components made globally. However, other datasets are not readily available (i.e. not available online), unlikely to conform to the QA protocols

followed by the included networks, or have too few stations to justify the time spent processing. In total, the resultant processed data collection, across all components, comprises of 7,275,148,646 measurements, beginning in 1970 with measurements from the Japan NIES network, going through to January 2023.

Some of the datasets come with restrictive data permissions that typically mean redistributing the data is impossible. Through dialogue with each of the data reporters, the majority of this data is included in the public GHOST dataset, however there are

a few networks which are not able to be redistributed, indicated in the data rights column of Table 1.



**Table 1.** General description of the reporting networks from whom data is sourced in GHOST. For each network, the temporal extent of processed data, the available matrices of processed components, the data source from which the original data was downloaded, and an indication if the data rights of the network permit the data to be redistributed as part of the GHOST dataset, are given.

| Network | Temporal Extent | Matrices | Data Source | Data Rights |
|---|---|---|---|---|
| ACTRIS (ACTRIS) | 2002 – 2023 | gas, pm, pm2.5, pm10, pm1 | NILU | ✓ |
| AERONET v3 Level 1.5 | 1993 – 2022 | aod, extaod, absaod, ssa, asy, rin, vconc, size | NASA | ✓ |
| AERONET v3 Level 2.0 | 1993 – 2022 | aod, extaod, absaod, ssa, asy, rin, vconc, size | NASA | ✓ |
| AMAP (Arctic Council Member States) | 1980 – 2022 | pm | NILU | ✓ |
| BJMEMC | 2013 – 2023 | gas, pm10, pm2.5 | BJMEMC | ✕ |
| CAMP (OSPAR Commission) | 1990 – 2022 | gas, pm, pm10, pm2.5 | NILU | ✓ |
| Canada NAPS | 1974 – 2022 | gas, pm, pm10, pm2.5 | Canada NAPS | ✓ |
| CAPMoN | 1988 – 2018 | gas, pm10 | CAPMoN | ✓ |
| Chile SINCA | 1993 – 2021 | gas, pm10, pm2.5 | Chile MMA | ✓ |
| CNEMC | 2014 – 2023 | gas, pm10, pm2.5 | CNEMC | ✕ |
| COLOSSAL (COLOSSAL) | 2018 | pm2.5 | NILU | ✓ |
| EANET | 1999 – 2021 | gas, pm, pm10, pm2.5 | EANET | ✕ |
| EEA AirBase | 1973 – 2013 | gas, pm, pm10, pm2.5, pm1 | EEA (a) | ✓ |
| EEA AQ e-Reporting | 2011 – 2023 | gas, pm, pm10, pm2.5, pm1 | EEA (b) | ✓ |
| EMEP (MET Norway; Tørseth et al., 2012) | 1971 – 2023 | gas, pm, pm10, pm2.5, pm1 | NILU | ✓ |
| EUCAARI (Kulmala et al., 2011) | 2007 – 2010 | pm10, pm2.5 | NILU | ✓ |
| EUSAAR (Cavalli et al., 2010) | 2006 – 2010 | pm, pm10, pm2.5, pm1 | NILU | ✓ |
| HELCOM (HELCOM) | 1996 – 2012 | pm, pm2.5 | NILU | ✓ |
| HTAP (Gusev et al., 2012) | 2002 – 2007 | gas | NILU | ✓ |
| IMPACTS (Aas et al., 2007) | 2001 – 2004 | gas, pm | NILU | ✓ |
| Independent (EBAS) | 2008 – 2022 | gas | NILU | ✓ |
| Japan NIES | 1970 – 2020 | gas, pm10, pm2.5 | Japan NIES | ✕ |
| Mexico CDMX | 1986 – 2022 | gas, pm10, pm2.5 | SEDEMA | ✓ |
| MITECO | 2001 – 2022 | gas, pm10, pm2.5 | Spain MITECO | ✓ |

*table continued on next page*



| Network | Temporal Extent | Matrices | Data Source | Data Rights |
|---|---|---|---|---|
| NADP AMNet | 2008 – 2021 | pm2.5 | NADP (a) | ✓ |
| NADP AMoN | 2007 – 2022 | gas | NADP (b) | ✓ |
| NILU (NILU et al.) | 1971 – 2023 | gas, pm, pm10, pm2.5, pm1 | NILU | ✓ |
| NOAA-ESRL (NOAA-ERSL) | 1973 – 2022 | gas, pm10, pm1 | NILU | ✓ |
| NOAA-GGGRN (NOAA-GGGRN) | 2001 – 2017 | gas | NILU | ✓ |
| OECD (OECD) | 1972 – 1980 | gas, pm | NILU | ✓ |
| UK AIR | 1973 – 2023 | gas, pm, pm10, pm2.5 | UK DEFRA | ✓ |
| UK DECC (University of Bristol et al.) | 2012 – 2019 | gas | NILU | ✓ |
| US EPA AirNow DOS | 2008 – 2023 | gas, pm10, pm2.5 | US EPA (a) | ✓ |
| US EPA AQS | 1980 – 2022 | gas, pm, pm10, pm2.5 | US EPA (b) | ✓ |
| US EPA CASTNET | 1987 – 2022 | gas, pm, pm2.5 | US EPA (c) | ✓ |
| WMO GAW WDCA (WMO, b) | 1981 – 2022 | pm, pm10, pm2.5, pm1 | NILU | ✓ |
| WMO GAW WDCGG | 1979 – 2022 | gas | WMO (c) | ✓ |
| WMO GAW WDCRG (WMO, d) | 1971 – 2023 | gas | NILU | ✓ |



**Table 2.** Names of the standard components processed in GHOST, grouped per data matrix. The "sconc" prefix is used for all components which can vary significantly with height. More information regarding these components can be found in Table A3.

| Matrix | GHOST Component Name |
|---|---|
| gas | sconco3, sconcno, sconcno2, sconcso2, sconcco, sconcch4, sconcc2h4, sconcc2h6, sconcc3h6, sconcc3h8, sconcisop, sconcc6h6, sconcc7h8, sconcc10h16, sconcnmvoc, sconcvoc, sconnmhc, sconchc, sconcnh3, sconchno3, sconcpan, sconchcho, sconchcl, sconchf, sconch2s |
| pm | sconcal, sconcas, sconcbc, sconcc, sconcca, sconccd, sconccl, sconccobalt, sconccr, sconccu, sconcec, sconcfe, sconchg, sconck, sconcmg, sconcmn, sconcmsa, sconcna, sconcnh4, sconcnh4no3, sconcni, sconcno3, sconcoc, sconcpb, sconcse, sconcso4, sconcso4nss, sconcso4ss, sconcv, sconczn |
| pm10 | pm10, pm10al, pm10as, pm10bc, pm10c, pm10ca, pm10cd, pm10cl, pm10cobalt, pm10cr, pm10cu, pm10ec, pm10fe, pm10hg, pm10k, pm10mg, pm10mn, pm10msa, pm10na, pm10nh4, pm10nh4no3, pm10ni, pm10no3, pm10oc, pm10pb, pm10se, pm10so4, pm10so4nss, pm10so4ss, pm10v, pm10zn |
| pm2.5 | pm2p5, pm2p5al, pm2p5a, pm2p5bc, pm2p5c, pm2p5ca, pm2p5cd, pm2p5cl, pm2p5cobalt, pm2p5cr, pm2p5cu, pm2p5ec, pm2p5fe, pm2p5hg, pm2p5k, pm2p5mg, pm2p5mn, pm2p5msa, pm2p5na, pm2p5nh4, pm2p5nh4no3, pm2p5ni, pm2p5no3, pm2p5oc, pm2p5pb, pm2p5se, pm2p5so4, pm2p5so4nss, pm2p5so4ss, pm2p5v, pm2p5zn |
| pm1 | pm1, pm1al, pm1as, pm1bc, pm1c, pm1ca, pm1cd, pm1cl, pm1cobalt, pm1cr, pm1cu, pm1ec, pm1fe, pm1hg, pm1k, pm1mg, pm1mn, pm1msa, pm1na, pm1nh4, pm1nh4no3, pm1ni, pm1no3, pm1oc, pm1pb, pm1se, pm1so4, pm1so4nss, pm1so4ss, pm1v, pm1zn |
| aod | od500aero, od500aerocoarse, od500aerofine, fm500frac, od380aero, od440aero, od550aero, od675aero, od870aero, od1020aero, ae440-870aero |
| extaod | extod440aero, extod440aerocoarse, extod440aerofine, extod675aero, extod675aerocoarse, extod675aerofine, extod870aero, extod870aerocoarse, extod870aerofine, extod1020aero, extod1020aerocoarse, extod1020aerofine, extae440-870aero |
| absaod | absod440aero, absod675aero, absod870aero, absod1020aero, absae440-870aero |
| ssa | sca440aero, sca675aero, sca870aero, sca1020aero |
| asy | asy440aero, asy440aerocoarse, asy440aerofine, asy675aero, asy675aerocoarse, asy675aerofine, asy870aero, asy870aerocoarse, asy870aerofine, asy1020aero, asy1020aerocoarse, asy1020aerofine, sphaero |
| rin | rinreal440, rinreal675, rinreal870, rinreal1020, rinimag440, rinimag675, rinimag870, rinimag1020 |
| vconc | vconcaero, vconcaerofine, vconcaerocoarse |
| size | vconcaerobin1, vconcaerobin2, vconcaerobin3, vconcaerobin4, vconcaerobin5, vconcaerobin6, vconcaerobin7, vconcaerobin8, vconcaerobin9, vconcaerobin10, vconcaerobin11, vconcaerobin12, vconcaerobin13, vconcaerobin14, vconcaerobin15, vconcaerobin16, vconcaerobin17, vconcaerobin18, vconcaerobin19, vconcaerobin20, vconcaerobin21, vconcaerobin22 |





## 3 GHOST processing workflow

Synthesising such a large quantity of data from disparate networks, is as much a challenge from a logistical and computational processing standpoint, as it is a scientific one. For this purpose we designed a fully parallelised workflow, based in Python, tailored to fully exploit the resources of the MareNostrum4 supercomputer, housed at the Barcelona Supercomputing Center (BSC). The workflow processes data per network, per component, through a pipeline of multiple processing stages, described visually in Fig. 1.

There are 9 stages in the piepline, which can be grouped broadly into 5 different stage types: data acquisition (stage 0), standardisation (stages 1 and 2), data addition (stages 3-5), temporal manipulation (stage 6), and data aggregation (stages 7 and 8).

There are two layers to the workflow parallelisation. Firstly, data per network, per component, is processed through the pipeline, in parallel. Secondly, the workload in each stage of the pipeline is divided into multiple smaller jobs, which are then processed in parallel also.

The processing in each pipeline ultimately results in harmonised netCDF4 files across all networks, per component. We will now describe the operation of each of the pipeline stages, in detail.





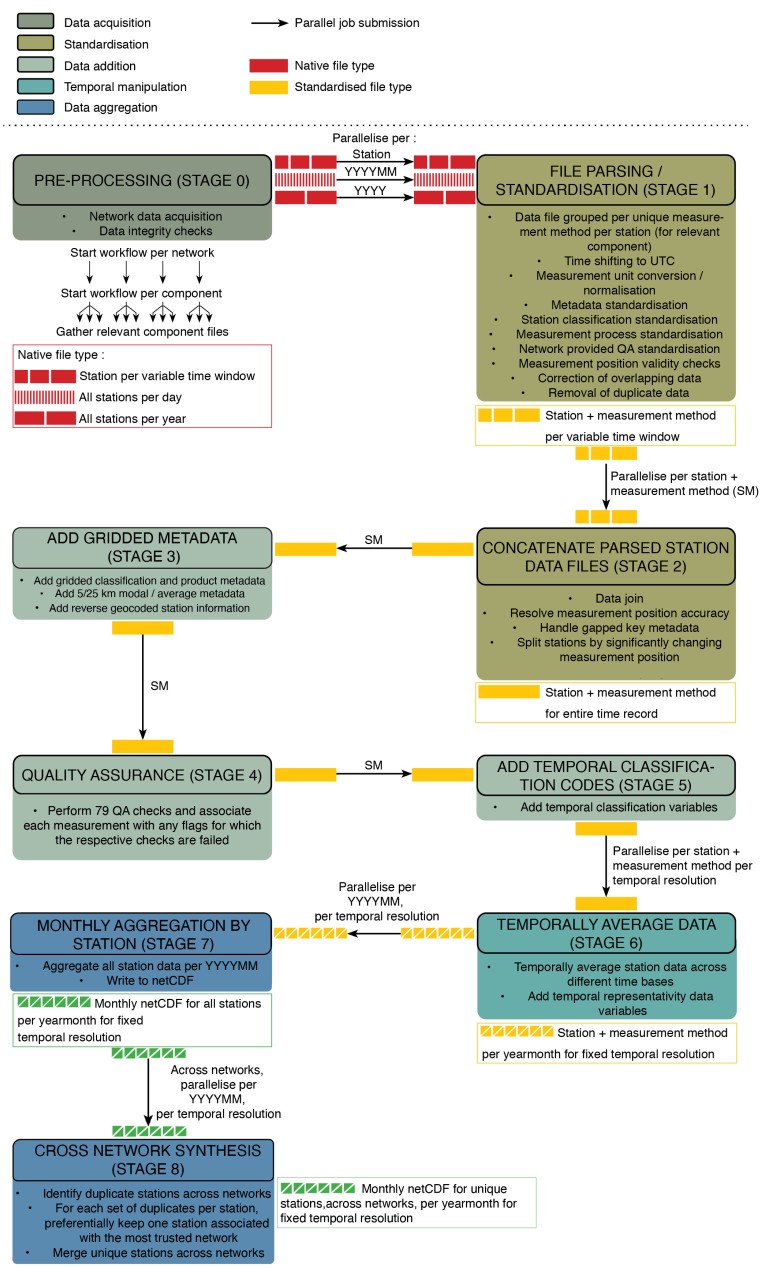

**Figure 1.** Visual illustration of the GHOST workflow, with data processed through a pipeline of 9 different stages. There are 5 broad stage types: data acquisition (stage 0), standardisation (stages 1 and 2), data addition (stages 3–5), temporal manipulation (stage 6) and data aggregation (stages 7 and 8). Data per network, per component, is processed through the pipeline, in parallel. The workload in each individual stage is divided into multiple smaller jobs, which are also processed in parallel (the arrows between the different stages indicating the type of parallelisation). The processing in each pipeline ultimately results in harmonised netCDF4 files across all networks, per component.





### 3.1 Pre-processing (Stage 0)

Starting the workflow, a processing pipeline per network, per component, is created. Before any processing can begin, in each pipeline, the relevant data for each network and component pair needs to be procured, and some initial checks performed to ensure the data integrity of the downloaded data.

#### 3.1.1 Data acquisition

All available measurement data between January 1970 and January 2023, from each of the 38 networks, for the listed components in Table 2, is downloaded. The available data matrices, temporal extent, and data source, are outlined per network in Table 1.

The data files come in a variety of formats, with no real consistency between any of them. Inconsistencies in file formats also 145 exist within some networks, e.g. Canada NAPS. In addition to the data files, there are often standalone metadata files, detailing the measurement operation at each station. The format of these files also varies considerably across the networks, and there can also be multiple files per network, e.g. EEA AQ e-Reporting.

For some networks, key details describing the measurement operation are published in network data reports / documentation. All available additional documentation across the networks was downloaded and read, greatly aiding the parsing / standardis-150 ation process described in Sect. 3.2.

#### 3.1.2 Data integrity checks

For some networks, some basic checks are first implemented before doing any file parsing, to ensure no fundamental problems exist with the data files. This is done in cases where information in the data filename and size can be used to identify potential data irregularities. For example, in the case of the EEA AQ e-Reporting network, data is reported per component, with unique 155 component codes contained within the filenames. In some cases, the component code in the filename is not correct for the component downloaded. In such cases, these files are excluded from any further processing, although such files represent a tiny fraction of all files.

With valid data files now gathered for the relevant network and component pair, file parsing can begin.

### 3.2 File parsing and standardisation (Stage 1)

In this stage, the relevant data files for a network and component pair are parsed, and the contained data / metadata is standardised. We define "data" variables to be those which vary per measurement, and "metadata" variables to be those which are typically applicable for vast swathes of measurements, varying on much longer timescales. Upon completion of the stage, the relevant parsed data from each data file is saved in standardised equivalent files, per station.

The type of parallelisation within stage 1 is dependant on how the data files are structured. If the data files include all 165 measurement stations per year, then parallelisation is done per year. If the files include all measurement stations per day, then



parallelisation is done per year and month. If the data files are separate for each station per time interval, then parallelisation is done per unique station.

The standardisation efforts made within GHOST are extensive, and cover a number of facets. As well as harmonising the data / metadata information provided by the networks, additional information is included in the form of gridded metadata,

GHOST QA flags, and temporal classification codes. The main standardisation types undertaken in GHOST are summarised in Table 3. Greater detail associated with each standardisation type is outlined in the referenced sections / summary tables, and the standard fields defined for each standardisation type are detailed in the referenced appendix tables.

Table 4 outlines the different types of data and metadata variables standardised in GHOST. The majority of these standardisations are performed in stage 1, with the processes involved in these standardisations described in the following subsections.



**Table 3.** Summary of the main standardisation types undertaken in GHOST. Per standardisation type, a brief description of the type, the number of variables associated with the type, the section where the type is discussed in the manuscript, and the numbers of the tables in the manuscript and appendix outlining the type, are detailed.

| Type | Description | N Variables | Section Detailed | Summary Table | Appendix Table |
|---|---|---|---|---|---|
| data | Information which is variable per measurement point, e.g. qa flags. | 21 | 3.2 | 4 | A1 |
| metadata | Quantitative and qualitative information associated with measurements, which is typically valid across large swathes of time, e.g. station latitude. | 163 | 3.2 | 4 | A2 |
| components | Specific information associated with each measured component, e.g. standard units. | 227 | 2 | 2 | A3 |
| station classifications | Variables used to classify the typical types of air parcels seen at a station, e.g. land use. | 6 | 3.2.10 | 8 | A4 |
| sampling types | Names of types of processes used to sample air, e.g. low volume continuous. | 8 | 3.2.8 | — | A5 |
| sample preparation types | Names of types of processes used to prepare samples for subsequent measurement, e.g. filter pack. | 10 | 3.2.8 | — | A6 |
| measurement methods | Names of the methods used for measuring component samples, e.g. ultraviolet photometry. | 104 | 3.2.8 | — | A7 |
| network QA | Standardised network QA flags. | 186 | 3.2.4 | 5 | A8 |
| simple network QA | Simplified standardised network QA flags. | 6 | 3.2.4 | 6 | — |
| GHOST QA | GHOST QA flags, each associated with GHOST implemented quality control checks. | 79 | 3.2.5 | 10 | A9 |
| temporal classifications | Temporal classifications of the station local time e.g. day / night. | 3 | 3.6 | 11 | — |





**Table 4.** Summary of the different types of data / metadata variables standardised in GHOST. For each type, a description is given, as well as the total number of associated variables. Definitions of all data / metadata variables are given in Tables A1 and A2 respectively.

| Group Type | N Variables | Description |
|---|---|---|
| **Data** | | |
| measurements | 2 | Unfiltered and filtered measurements. |
| time | 3 | Start times of measurement windows referenced against different time standards. |
| network QA | 1 | Standardised network QA flags. |
| simple network QA | 1 | Simplified standardised network QA flags. |
| GHOST QA | 1 | GHOST QA flags, each associated with GHOST implemented quality control checks. |
| measurement uncertainties | 2 | Reported and derived measurement uncertainties. |
| temporal classifications | 3 | Temporal classifications of the station local time. |
| data representativity | 8 | Variables providing the percentage data representativity of native measurements across multiple temporal periods. |
| **Metadata** | | |
| GHOST version | 1 | Version number of GHOST. |
| station information | 31 | Information associated with the measurement station. |
| station classifications | 6 | Variables used to classify the typical types of air parcels seen at a station. |
| gridded classifications | 29 | Station classes derived from various gridded classification types. |
| gridded products | 38 | Station products, i.e. numeric information, derived from various gridded product types. |
| measurement information | 45 | Information associated with the measurement process. |
| contact information | 6 | Contact information for the principal data investigators and station contact. |
| further detail | 6 | Additional information provided by the network, which cannot be easily standardised. |
| process warnings | 1 | Information regarding any assumptions made in the GHOST processing pipeline. |



### 3.2.1 Data grouping, by station reference and measurement method

Firstly, each data file is read into memory. All non-relevant component data is removed, and a list of unique reference station IDs associated with remaining file data is generated, henceforth referred to as station references.

In some cases, stations operate multiple instruments to measure the same component, often utilising differing measurement methods. There can therefore be data in a file, associated with the same station reference, but resultant from differing measurement methods. To handle such instances, station data in GHOST is grouped via a station reference, as well as a standard

measurement method. Each station group is associated with a GHOST station reference, defined as: "[network station reference]_[standard measurement methodology abbreviation]", and is saved in the GHOST metadata variable: "station_reference". The standardisation of measurement methodologies is detailed in Sect. 3.2.8.

The data in each of the station groups, are then parsed independently.

### 3.2.2 Measured values

Measurements are typically associated with a measurement start date / time, as well as the measurement end date / time, or the temporal resolution of measurement. The period between the measurement start time and end time can be termed the measurement window. In almost all cases, the measurement values reflect an average across the measurement window. Occasionally, there are multiple reported statistics per measurement window e.g. average, standard deviation, percentiles. Only measurements which represent an average statistic are retained.

Missing measurements are often recorded as empty strings, or a network defined numeric blank code. For these cases, the values are set to Not a Number (NaN). Measurements for which the start time or temporal resolution cannot be established, are dropped. Any measurements which do not have any associated units, or have unrecognisable units, are dropped. All measurements are converted to GHOST standard units (see Sect. 3.2.13).

In the case of one specific component, aerosol optical depth at 550nm (od550aero), the measurement is derived synthetically,

using multiple other components (od440aero, od675aero, od875aero, and extae440-870aero), following the Ångström power law (Ångström, 1929). All dependent component measurements are needed to be non-NaN for this calculation, otherwise od550aero is set as NaN. All od550aero values are associated with the GHOST QA flag "Data Product" (code 45), and any instances where od550aero cannot be calculated, are associated with the flag "Insufficient Data to Calculate Data Product" (code 46). The concept for these flags is explained in Sect. 3.2.5.

At this point, if there are no valid measurements remaining, then the specific station group does not carry forward in the pipeline. If there are valid measurements, these are then saved to a data variable named by the standard GHOST component name (see Table 2), e.g. sconco3 for $O_3$.

### 3.2.3 Date, time, and temporal resolution

Some of networks provide the measurement start date / time in local time, thus a unified time standard is needed to harmonise

215 times across the networks. We choose to shift all times to Coordinated Universal Time (UTC), for which many of the networks



already report in. For most cases where the time is not already in UTC, the UTC offset or local timezone is reported per measurement, or in metadata / network documentation (i.e. constant over all measurements). However, in the case that no local timezone information exists, this is obtained using the Python timezonefinder package (Michelfeit), as detailed in Sect. 3.4.5.

In order to store the measurement start date / time in one single data variable, it is transformed to be minutes from a fixed reference time (0001-01-01 00:00:00 UTC). Note, these units differ from the end units of the "time" data variable in the finalised netCDF4 files (see Sect. 3.7).

A small number of stations have consistent daily gaps on the 29th February during leap years. An assumption is made that this is an actual missing day of data, imposed by erroneous network data processing, and that data labeled for the 1st of March is indeed for the 1st of March. Some networks also report measurement start times of 24:00. Thus is assumed to be referring 225 to 00:00 of the next day.

For some networks, the temporal resolution of measurements are provided, and for others both measurement start and end dates / times are given, from which the temporal resolution can be derived. In some other cases, the temporal resolution is fixed for the entire data file, either stated in the filename, or in network documentation.

In some instances, the measurement start time is also not provided, with measurements provided in a fixed format, e.g. 230 24 hours per data line, with the column headers: "hour 1", "hour 2", etc. In these cases, there is some ambiguity as to where measurements start and stop. For example, does "hour 1" refer to $00:00 - 01:00$, $01:00 - 02:00$, or $00:30 - 01:30$? An assumption is made in these cases that the column header refers to the end of the measurement window, i.e. hour 1 = $00:00 - 01:00$. The temporal resolution of measurements can vary widely (e.g. hourly, 3 hourly, daily), all of which are parsed in GHOST. When later wishing to temporally average data to standard resolutions (Sect. 3.7), the temporal resolution of each original 235 measurement is required, and therefore this information is stored through the processing.

### 3.2.4 Network quality assurance

Many of the networks provide QA flags associated with each measurement. These can be used to represent a number of things, but are typically used to highlight erroneous data, or to inform of potential measurement concerns. It is also often the case that one measurement is associated with multiple QA flags. Network QA flag definitions were found through the investigation of 240 reports / documentation.

GHOST handles these flags in a sophisticated manner, mapping all the different types of network QA flags to standardised network QA flags. Table 5 shows a summary of the different types of standard flags, ranging from basic data validity flags, to flags informing on the weather conditions at the time of measurement. The standard flags are saved in the GHOST data variable: "flag", as a list of numeric codes per measurement, i.e. each measurement can be associated with multiple flags. Each 245 individual standard flag name (and associated flag code) is defined in Table A8. Whenever a flag is not active, a fill value (255) is set instead.

The large number of standard network QA flags gives the user a great number of options for filtering data, but for users who are looking to more crudely remove obviously bad measurements, the wealth of options could be overwhelming. For such cases we also implement a greatly simplified version of the standard network QA flags, defined in Table 6, and saved in the "flag_



simple" variable. These definitions follow those defined in the WaterML2.0 open standards (Taylor et al., 2014). As opposed
to the "flag" variable, each measurement can only be associated with one simple flag.



**Table 5.** Summary of the standard network QA flag types, stored in the "flag" variable. These flags represent a standardised version of all the different QA flags identified across the measurement networks. For each type, a description is given, as well as the number of flags associated with each type. Definitions of the individual flags are given in Table A8.

| Flag Types | N Flags | Description |
|---|---|---|
| basic | 5 | Simple flags which inform about the level of validity of the data. |
| estimated | 7 | Flags informing that data has been estimated in some fashion. |
| extreme / irregular | 13 | Flags informing of irregular measurement data, or close to detection limits. |
| measurement issue | 18 | Flags informing of issues associated with the measurement process. |
| operational maintenance | 12 | Flags informing of instrument maintenance activities being undertaken. |
| data formatting issue | 2 | Flags informing of issues associated with the formatting or processing of data files. |
| representativity | 8 | Flags informing of the temporal representativity of measurements. |
| weather | 79 | Flags informing of the specific local weather conditions at time of measurement. |
| local contamination | 29 | Flags which inform of local contamination events, or atmospheric obscuration of some kind. |
| exceptional event | 11 | Flags informing of exceptional local events. |
| meteorological infinites | 2 | Flags informing of meteorological conditions that cannot be digitised, i.e. infinite. |





**Table 6.** Definitions of the simplified standard network QA flags, stored in the "flag_simple" variable. These flags represent a simplified version of network QA flags defined in Table A8. These definitions follow those defined in the WaterML2.0 open standards (Taylor et al., 2014).

| Flag Code | Flag Name | Description |
| --- | --- | --- |
| 0 | estimate | Data is an estimate only, not a direct measurement. |
| 1 | good | Data has been examined and represents a reliable measurement. |
| 2 | missing | Data is missing. |
| 3 | poor | Data should be considered as low quality and may have been rejected. |
| 4 | suspect | Data should be treated as suspect. |
| 5 | unchecked | Data has not been checked by any qualitative or quantitative method. |





### 3.2.5 GHOST quality assurance

Each of the native network QA flags often come with an associated validity recommendation, informing whether a measurement is of sufficient quality to be trusted or not. For example, if the network QA flag is informing of rainfall at the time of measurement, the recommendation would most probably be that the measurement is valid, whereas if the flag is informing of instrumental issues, the recommendation would likely be that the measurement is invalid.

This creates a binary classification, where data can be filtered out based on the recommendation of the data provider. This is extremely useful when an end user simply wants to have data that they know is of a reliable standard, and do not wish to preoccupy themselves choosing which network QA flags to filter by.

As well as writing standard network QA flags per measurement, GHOST own QA flags are also set, with each flag relating to a GHOST implemented quality control check. These flags are stored as a list of numeric codes per measurement, in the "qa" data variable. A summary table outlining the different GHOST QA flag types is given in Table 10, and individual standard flag names (and associated flag codes) are defined in Table A9. Whenever a flag is not active, a fill value (255) is set instead. The majority of these flags are set in stage 4 of the pipeline (Sect. 3.5), however a few are set in stage 1. For example, one of those set is the network recommendation that a measurement should be invalidated: "Invalid Data Provider Flags – Network Decreed" (code 7).

In many instances the network suggestions to invalidate measurements are entirely subjective, and the person who should decide whether a measurement should be retained or not, is the end user themselves. For example, the data provider can recommend that a measurement should be invalidated due to windy conditions, but the end user may well be interested in such events. We therefore create a GHOST set of binary validity classifications, which are less prohibitive than the original data provider ones. Only in the case that a data flag informs that there has been a technical issue with the measurement, or that the measurement has not met internal quality standards, is a measurement recommended to be invalidated. This is again written as a GHOST QA flag: "Invalid Data Provider Flags – GHOST Decreed" (code 6).

Further GHOST QA flags which are set in stage 1 relate to assumptions / errors found when standardising the metadata associated with measurement processes (described in Sect. 3.2.8), and when an assumption has been made in converting measurement units (described in Sect. 3.2.13).

### 3.2.6 Metadata

Networks provide metadata in both quantitative and qualitative forms. Metadata is either provided in an external file, stored in the data file header, or given line by line.

Across the networks there is a large variation in the quantity and detail of metadata reported. In GHOST there is an attempt to ingest and standardise as much available metadata as possible from across the networks, which can be broadly separated into 6 different types, as illustrated in Fig. 2. Table 4 outlines the types of metadata variables standardised in GHOST, and Table A2 defines each of these variables individually.





The standardisation process for the majority of metadata variables consists of mapping the slightly varying variable names, across the networks, to a standard name, e.g. "lat", "degLat" to "latitude"; converting units (if a numeric variable) to standard ones; and standardising string formatting (if a string variable). For some variables, detailed work is needed to be done to standardise information from across the networks, i.e. station classifications and measurement information, the processes for which are discussed in subsequent sections. Standardisations are not performed for descriptive variables, for which it would be impossible to do so, represented in Fig. 2 by the "Further Detail" grouping. If any metadata variable is not provided by a network, or the variable value is an empty string, the value in GHOST is set to be NaN.

In GHOST, metadata is treated dynamically, i.e. it is allowed to change with time. A limitation of previous data synthesis efforts is that the metadata is static for a station throughout the entire time record. If a station has measured a component from the 1970s to the present day, the typical air sampled at the station could change in a number of ways. For example, a road may be built nearby, the population of the nearest town may swell, or the sampling position may be moved slightly. Significant changes can also occur in the physical measurement of the component. Measurement techniques have evolved over time, and consequently the accuracy and precision of measurements have improved. All of these factors impact upon the measurements. Having dynamic metadata allows for inconsistencies or jumps in the measurements over time to be understood, something not possible with static metadata.

The way the dynamic metadata is stored in GHOST, is in columns. Per station, blocks of metadata are associated with a start time, from which they apply. For data files which report metadata line by line, this leads to vast number of metadata columns, in most cases with no metadata changing between columns. To resolve this duplication, after all metadata parsing and standardisation is complete, each metadata column is cross compared with the next column, going forwards in time. If all of certain key metadata variables in the next column are identical to the current column, the next column is entirely removed. These key variables are defined, per metadata group type, in Table A12.

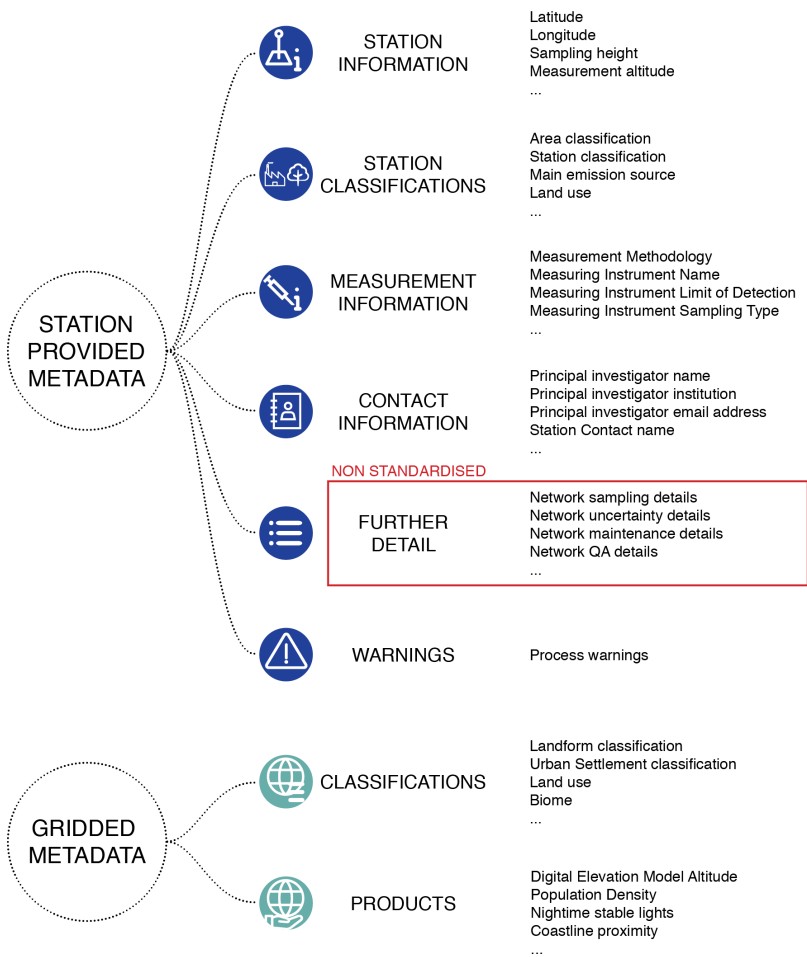

**Figure 2.** Visual summary of the types of metadata ingested and standardised in GHOST. The metadata can be separated into 2 distinct categories, station provided metadata, and gridded metadata.





### 3.2.7 External metadata join

When metadata is reported in external file/s, separate to the data, it is typically associated with the data using the network station reference. In some cases, the association is made using a sample ID, with individual measurements tagged with an ID that is associated with a specific collection of metadata. Stations for which external metadata cannot be associated, and there is no other source of metadata (i.e. in the data files), are excluded from further processing.

The metadata values in the external files are assumed to be valid across the entire time record. For the specific case of Japan NIES, external metadata files are provided per year, permitting updates to the metadata with time.

For some networks there are several different external metadata files provided, e.g. EEA AQ e-Reporting. Some of the metadata variables across these files are repeated, whereas some are unique to specific files. To solve this, the external files are given priority rankings, so that when variables are repeated, it is known which file to preferentially take information from.

For some networks, no metadata is provided, either in the data files or in external files, therefore the metadata for key variables (e.g. longitude, latitude, station classifications) is compiled manually in external files. This is done principally using information gathered from network reports / documentation. For other networks, the provided metadata is very inconsistent station to station, and therefore external metadata files are compiled manually to ensure some key variables are available across all stations, e.g. station classifications. Manually compiled metadata is only ever accepted for a variable when there is no other network provided metadata for that variable available through the time record.

When station classifications are manually compiled, this is first attempted to be done following network documentation on how the classifications are exactly defined. If no documentation exists, this is then done by assessing the available network station classifications in conjunction with their geographical position using Google Earth, to attempt to empirically understand the classification procedures. The stations are then classified following this empirically obtained logic.

### 3.2.8 Measurement process standardisation

The type of measurement processes implemented in measuring a component can have a huge bearing on the accuracy of measurements. Despite most networks providing information which details some aspects of the measurement processes, this information is incredibly varied, both in terms of detail and format.

Within GHOST, substantive efforts are made to fully harmonise all information relating to the measurement of a component. As there are 227 components processed within GHOST, there is naturally a huge number of differing measurement processes used to measure all of these different components. For example, for $O_3$, as it is relatively easy to measure, a standalone instrument both samples and measures the concentration continuously. For speciated PM10 measurements, a filtering process is first needed to separate the PM by size fraction, and then a speciated measurement of the relevant size fraction is performed.

In GHOST, an attempt is made to standardise all measurement processes across 3 distinct measurement steps: sampling, sample preparation, and measurement. The "sampling" step refers to type of sampling used to gather the sample to be measured, "sample preparation" refers to processes used to prepare the sample for measurement, and "measurement" refers to the ultimate measurement of the sample.





Combining information across these 3 different steps can be used to subsequently describe all different types of measurement processes. Figure 3 visually shows some typical measurement configurations that can be described by mixing these steps. For example, the measurement of $O_3$ is represented by the "automatic" configuration, where information from the sampling and measurement steps is sufficient to describe the measurement process, i.e. there is no preparation step.

In GHOST, a database has been created, identifying and storing information from across the measurement steps, in a standardised format. For the "sampling" step, 8 different sampling types, and 83 different instruments which employ the sampling types are identified, defined in Table A5. For the "sample preparation" step, 10 different preparation types, and 20 specific techniques which employ the preparation types are identified, defined in Table A6. For the "measurement" step, 104 different measurement methods, and 508 different instruments which employ the methods are identified, defined in Table A7.

For each specific sampling / measuring instrument, there is typically documentation published outlining the relevant specifications of the instrument, e.g. providing information about the limits of detection, flow rate. Where this documentation is made available online, it is downloaded and parsed, and the relevant specifications are associated with the standard instruments in the database.

In order to connect network reported metadata with the standard information in the database, firstly, all network provided metadata associated with measurement processes is gathered and concatenated to one string. These strings are then manually mapped to standard elements in the database. This mapping procedure is a huge undertaking but ultimately returns a vast quantity of standardised specification information that can be associated with measurements. Table 7 outlines all the types of measurement metadata variables that information is returned for, with the full list of available variables defined in Table A2, in the "Measurement Information" section. All measurements are therefore associated with a standard measurement method, the abbreviation for which (defined in Table A7) forms the second part of the "station_reference" variable, defined in Sect. 3.2.1. In some cases, the networks themselves provide some measurement specification information. This can differ in some cases from the documented instrument specifications, as there may be station made modifications to instrumentation, therefore improving upon the documented specifications. This reported information is also ingested in GHOST, for the exact same specification variables as ingested in the documented case. There are therefore 2 variants for each of these variables. All variables which contain the "reported" string contain information from the network, whereas variables containing the "documented" string contain information from the instrument documentation.

Multiple QA checks are also performed throughout the standardisation process. Each standardised sampling type / instrument, sample preparation type / technique and measurement method / instrument is associated with a list of components for which they are known to: 1. be associated with the measurement of, and 2. be associated with the accurate measurement of.

For example, for the first point, the "gravimetry" measurement method is not associated with the measurement of $O_3$, therefore this method would identified as being erroneous, and associated measurements flagged by GHOST QA ("Erroneous Measurement Methodology", code 22, in this case). For the second point, the "chemiluminescence (internal molybdenum converter)" method is associated with the measurement of $NO_2$, but there are known major measurement biases (Winer et al., 1974; Steinbacher et al., 2007), therefore these instances would be also flagged by GHOST QA ("Invalid QA Measurement Methodology", code 23).



Table A7 details the components each standard measurement method is known to be associated with the measurement of, as well as the components that each method can accurately measure. Additional GHOST QA flags are set when the specific
names of types / techniques / methods / instruments are unknown, as well as when any assumptions have been made in the mapping process. All of these flags are defined in Table A9, in the "Measurement Process Flags" section.





MEASUREMENT STEPS        TYPICAL MEASUREMENT CONFIGURATIONS

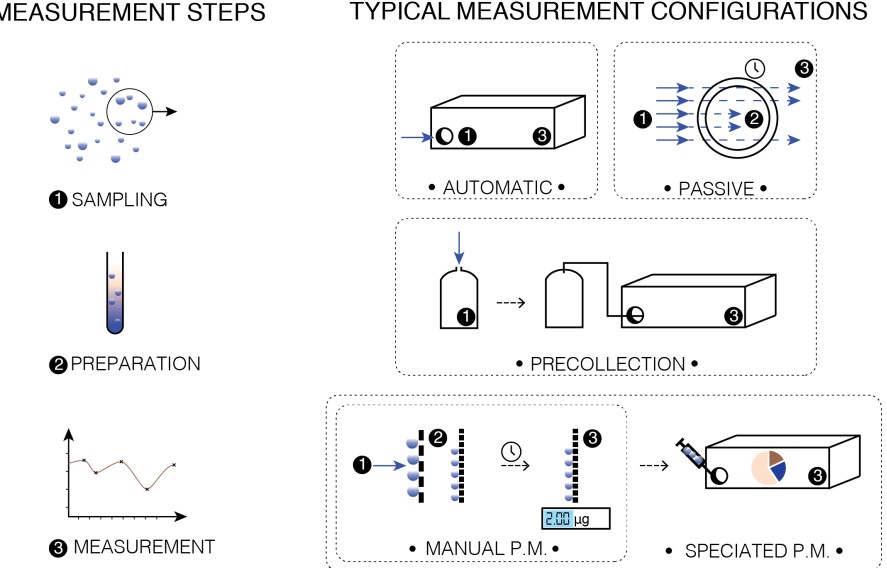

**Figure 3.** Visual illustration of the 3 GHOST standard measurement process steps, and how those steps are combined in the most typical measurement configurations. The 3 standard steps are sampling, preparation, and measurement.





**Table 7.** Outline of the type of standard metadata variables in GHOST associated with the measurement process. A description is given per variable. Many of these variables types will have have two associated variables, one giving network reported information, and another giving information stemming from instrument documentation. More information is available in Table A2.

| Variable Type | Description |
| --- | --- |
| sampling type | Type of process used to sample air. |
| sampling preparation types | Types of processes used to prepare sample for subsequent measurement. |
| sampling preparation techniques | Specific technique of a utilised preparation type. |
| measurement methodology | Methodology used for measuring component. |
| instrument name | Specific name of the sampling / measuring instrument. |
| flow rate | Volume of fluid sampled per unit time. |
| lower limit of detection | Lower limit of measurement detection. |
| upper limit of detection | Upper limit of measurement detection. |
| accuracy | Difference between a measured value and the actual value of a known part. |
| precision | Measure of the variation seen when the same part is measured repeatedly with the same instrument. |
| uncertainty | Measurement uncertainty. |
| measurement resolution | Smallest level of change of a measured quantity that the instrument can detect. |
| zero drift | Measurement drift across the full scale caused by slippage, or due to undue warming up of the electronic circuits. |
| span drift | Measurement drift which proportionally increases along the upward scale. |
| zonal drift | Measurement drift which occurs only over a portion of the full scale. |
| absorption cross section | Assumed molecule cross-section for the component being measured (for optical measurement methods). |
| inlet information | Description of the sampling inlet of the measuring instrument. |
| calibration scale | Name of the scale used for the calibration of the measuring instrument. |
| retrieval algorithm | Name of the retrieval algorithm associated with measurement (for remote sampling). |
| volume standard temperature | Temperature associated with the volume of the sampled gas. |
| volume standard pressure | Pressure associated with the volume of the sampled gas. |
| reported units | Units that the measured component are natively reported in. |
| manual name | Name of the sampling / measuring instrument manual. |
| further details | Further miscellaneous details associated with the measurement process. |
| process details | Miscellaneous details about assumptions made in the standardisation of the measurement process. |



### 3.2.9 Measurement limits of detection and uncertainty

In some cases, measurements will be associated with estimations of uncertainty, and limits of detection (LODs), both lower and upper, by the measuring network. These can be provided per measurement, or as constant metadata values. This information is incredibly useful scientifically, as it allows for the screening of unreliable measurements.

In GHOST this information is captured as GHOST QA flags whenever LODs are exceeded: "Below Reported Lower Limit of Detection" (code 71), and "Above Reported Upper Limit of Detection" (code 74), and as a data variable for the measurement uncertainty: "reported_uncertainty_per_measurement".

This information can be complemented with documented information associated with the measuring instrument (if known). If documented LODs for an instrument are exceeded, then this sets the GHOST QA flags: "Below Documented Lower Limit of Detection" (code 70), and "Above Documented Upper Limit of Detection" (code 73). Typically, the reported network information is to be preferred over the documented instrument information, as any manner of modifications may have been made to the instrument post sale. Two GHOST QA flags encapsulate this concept neatly, first trying to evaluate LOD exceedances by the reported information if available, and if not then by documented instrument information: "Below Preferential Lower Limit of Detection" (code 72), and "Above Preferential Upper Limit of Detection" (code 75).

In some cases the measurement uncertainty is not provided directly, but can be calculated from other associated metadata information (again network reported information being preferred to instrument documentation). This is done using the quadatric addition of measurement accuracy and precision metrics, and is saved as the data variable: "derived_uncertainty_per_measurement".

All of this information, is converted to the standard units of the relevant component (see Sect. 3.2.13) before setting QA flags, or metadata / data variables.

### 3.2.10 Station classification standardisation

The networks provide a variety of station classification information, which can be used to inform of the typical types of air parcels seen at a station. Within GHOST, all this classification information is standardised to 6 metadata variables, as outlined in Table 8.

For each standard classification variable, the available class fields are also standardised, done through an extensive assessment of all available fields across the networks. This process is inherently associated with some small inconsistencies, as there is not always a perfect alignment between the available class fields across the networks, as well as significant variation in the granularity of fields in some cases, e.g. for station area classifications: "urban" vs "urban centre". In order to account for variations in field granularity, all standard class fields can consist of a primary class and a sub-class, separated by a "-", e.g. "urban", or "urban-centre". These fields are defined per variable in Table A4.





**Table 8.** Outline of the GHOST standard station classification metadata variables, the standard fields per variable, and a description of each
variable. In Table A4 each of the fields per variable are defined.

| Metadata Variable | Standard Fields | Description |
| --- | --- | --- |
| area_classification | urban, urban-centre, urban-suburban, rural, rural-near_city, rural-regional, rural-remote | Classification of the type of area a station is situated in. |
| station_classification | background, point_source, point_source-industrial, point_source-traffic | Classification of the type of air dominantly measured by a station. |
| main_emission_source | agriculture, commercial_and_residential_ combustion, extraction_of_fossil_fuels, industrial_ combustion, natural, other_mobile_sources_and_ machinery, production_processes, power_ production, road_transport, solvents, waste_ treatment_and_disposal | Main emission source influencing air measured at a station. |
| land_use | barren, barren-beach, barren-desert, barren-rock, barren-soil, forest, open, open-grassland, open-savanna, open-shrubland, snow, urban, urban-agricultural, urban-blighted, urban-commercial, urban-industrial, urban-military, urban-park, urban-residential, urban-transportation, water, wetland | Dominant land use in the area of a station. |
| terrain | coastal, complex, flat, mountain, rolling | Dominant terrain in the area of a station. |
| measurement_scale | micro, middle, neighbourhood, city, regional | Denotation of the geographic scope of the air measured at a station. |



### 3.2.11 Check measurement position validity

After all metadata information has been parsed, some checks are done to ensure if the measurement position metadata is sensible in nature, with the checks done as follows:

1. Check if the longitude and latitude are outside valid bounds, outside of -180° <-> 180° and -90° <-> 90° bounds respectively.

2. Check if both the longitude and latitude are equal to 0.0, i.e. in the middle of the ocean. In this case the position is assumed to be erroneous.

3. Check if the altitude and measurement altitude are < -413m, i.e. lower than the lowest exposed land on Earth, the Dead Sea Shore.

4. Check if the sampling height is < -50m. Such a sampling height would be extremely strange to be so far below the station altitude.

Any measurement position metadata failing any of the these checks is set to be NaN. Any stations associated with longitudes or latitudes equal to NaN, are excluded from further processing.

### 3.2.12 Correcting duplicate or overlapping data

Some network data files contain duplicated or overlapping measurement windows. Work is done to correct these instances, as well as ensuring measurements and all other data variables (e.g. "qa", "flag") are ordered to be ascending across time.

Measurement start times are first sorted in ascending order. If any measurement windows are identically duplicated, i.e. same start and end time, windows are iteratively screened by the GHOST QA: "Not Maximum Data Quality Level" (code 4), "Preliminary Data" (code 5), "Invalid Data Provider Flags – GHOST Decreed" (code 6), in that order, until the duplication is resolved. If there is still a duplication after screening, then the first indexed measurement window is kept preferentially, and the others dropped.

After removing the duplicate windows, it is next checked whether any measurement window end times overlap with the next window start time. If an overlap is found, again windows are screened iteratively by the GHOST QA flags: 4, 5, 6, in that order, until the duplication is resolved. If there is still an overlap, the remaining windows with the finest temporal resolution are kept, e.g. hourly resolution is preferred to daily. If this still does not resolve the overlap, then the first indexed remaining measurement window is kept preferentially.

Both of these processes are done recursively until each measurement window does not overlap with any other, and has no duplicates.

### 3.2.13 Measurement unit conversion

A major challenge in a harmonisation effort such as GHOST, is that components are often reported in various different units, and in many instances report entirely different physical quantities, requiring complex conversions.





In GHOST, each component is assigned standard units, listed in Table A3, for which all native provided units are converted to. The units for all components in the gas and particulate (pm, pm10, pm2.5, pm1) matrices are reported as either mole fractions (e.g. ppbv = $\text{nmol mol}^{-1}$ = $1 \times 10^{-9}\ \text{mol mol}^{-1}$) or mass densities (e.g. $\mu\text{g m}^{-3}$), in a range of different forms across the networks. All gas components are standardised to be mole fractions, whereas all particulate components are standardised to be mass densities. Components in the other matrices are all unitless, except for vconc and size, which are standardised to be $\mu\text{m}^3\ \mu\text{m}^{-2}$. Components for these two matrices all stem from the AERONET v3 Level 1.5 and AERONET v3 Level 2.0 networks, and are already reported in GHOST standard units. Unit conversion is therefore only handled for gas and particulate matrix components.

Almost all gas and particulate measurement methodologies fundamentally measure in units of number density (e.g. molecules $\text{cm}^{-3}$), or as a mass density, not as a mole fraction. The conversion from a number density to a mass density is simply:

$$\rho_C = \frac{\rho_{NC} \cdot M_C}{N_A}, \tag{1}$$

where $\rho_C$ is the mass density of the component ($\text{g m}^{-3}$), $\rho_{NC}$ is the number density of the component (molecules $\text{m}^{-3}$), $M_C$ is the molar mass of the component ($\text{g mol}^{-1}$), and $N_A$ is Avogadro's number ($6.0221 \times 10^{23}\ \text{mol}^{-1}$).

The conversion from mass density to mole fraction, depends on both temperature and pressure:

$$V_C = \rho_C \cdot \frac{RT}{M_C P}, \tag{2}$$

where $V_C$ refers to the component mole fraction ($\text{mol mol}^{-1}$), $R$ is the gas constant ($8.3145\ \text{J mol}^{-1}\ \text{K}^{-1}$), $P$ is pressure (Pa), and $T$ is temperature (K). The temperature and pressure variables refer to the internal temperature and pressure of the measuring instrument, not ambient conditions, physically relating to the volume of the air sampled.

Some component measurements are reported in units of mole fractions per element, e.g. ppbv per carbon, ppbv per sulphur. These units are converted to the mole fractions of the entire components by:

$$V_C = \frac{V_C}{A_{EC}}, \tag{3}$$

where $V_{EC}$ is the mole fraction per element ($\text{mol mol}^{-1}$), and $A_{EC}$ is the number of relevant element atoms in the measured component (e.g. 2 carbon atoms in $C_2H_4$).

In a small number of instances, for measurements of total VOCs (Volatile Organic Compounds), total NMVOCs (Non-Methane Volatile Organic Compounds), total HC (Hydrocarbons), total NMHC (Non-Methane Hydrocarbons), are reported as mole fractions per carbon. As these measurements sum over various components, there is no fixed number of carbon atoms. It is assumed that these measurements are normalised to $CH_4$, i.e. 1 carbon atom, as is done typically.

In order to ensure measurements are comparable across all stations, measurements are typically standardised by each network to a fixed temperature and pressure, i.e. no longer relating to the actual sampled gas volume. The standardisation applied differs per network, but in almost all cases follows EU or US standards. The EU standard sets the temperature and pressure as 293 K and 1013 hPa (European Parliament, 2008), whereas the US standard is 298.15 K and 1013.25 hPa (US EPA, 2023). The



differences applied standards can lead to significant differences in the reported values of the same initial measurements. For example, a CO measurement of $200\,\mu\mathrm{g\,m^{-3}}$, with an internal instrument temperature and pressure of $301.15\,\mathrm{K}$ and $1000\,\mathrm{hPa}$, is $3.55\,\mu\mathrm{g\,m^{-3}}$ higher following EU standards compared to US ones ($208.2$ vs $204.7\,\mu\mathrm{g\,m^{-3}}$). This means the same measurements using EU standards will be always slightly higher (1.7%) than those using US standards.

To attempt to remove this small inconsistency across networks, after measurement unit conversion, all gas and particulate

matrix measurements are re-standardised to a GHOST defined standard temperature and pressure: $293.15\,\mathrm{K}$ and $1013.25\,\mathrm{hPa}$, equivalent to the normal temperature and pressure (NTP). An assumption is made that the original units of measurement are either a mass or number density, i.e. the measurement is dependent on temperature and pressure.

This standardisation is only done when there is confidence in the sample gas volume associated with measurements, i.e. the volume standard temperature and pressure are reported, or there is a known network standard temperature and pressure for

a component. When any assumptions are made when performing this standardisation, or the sample gas volume is unknown, then GHOST QA flags are written, outlined in the "Sample Gas Volume Flags" section in Table A9.

Where the standard units are a mass density, the standardisation is done by:

$$S_C = \rho_C \cdot \frac{T_N}{293.15} \cdot \frac{1013.25}{P_N} \tag{4}$$

Where the standard units are a mole fraction, the conversion is by:

$$S_C = MR_C \cdot \frac{293.15}{T_N} \cdot \frac{P_N}{1013.25}, \tag{5}$$

where $S_C$ is the GHOST standardised values, $T_N$ is the known standard temperature, and $P_N$ is the known standard pressure.

### 3.3 Concatenate parsed station data files (Stage 2)

Now that all data files for a network and component pair have been parsed, and saved in standardised equivalent files, the next step is to concatenate all files associated with the same station, creating a complete time series.

Typically this is a very easy process, simply joining the files together through the time record. However, it quickly becomes very complex when there are duplicated or overlapping files. Choosing which file to take data from for each file conflict is a tricky issue, for which a number of factors need to be taken into consideration.

In stage 2 of the pipeline, a methodology is implemented to systematically resolve each of these file conflicts, per station. Additional work is done to fill gaps in metadata across the time record, and finally a check is undertaken to determine if the

515 station measurement position is consistent across the time record. Where there are significant changes in the measurement position, station data is split apart to reflect the significantly different air masses being measured. Figure 4 visually describes the stage 2 operation.

Parallelisation is done per unique station (via station_reference) in the stage.



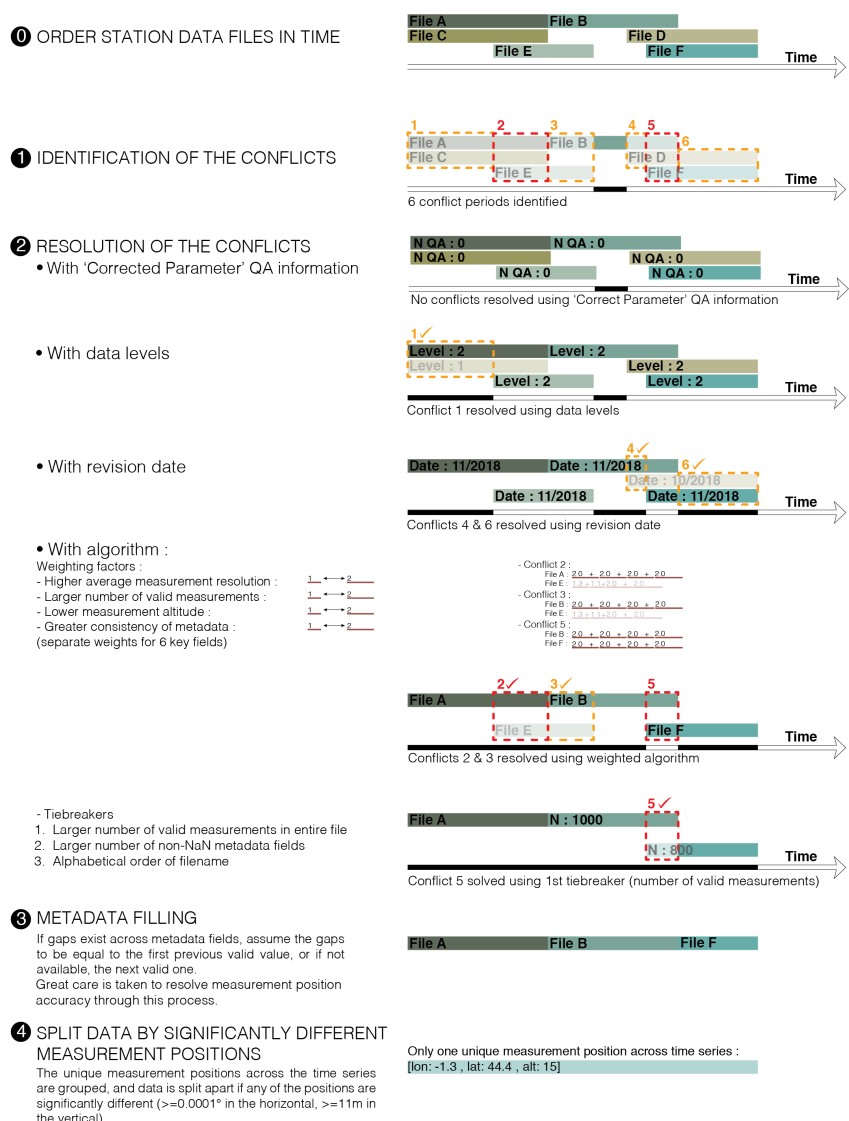

**Figure 4.** Visual illustration of the resolution process for temporally conflicting parsed station data files, in stage 2 of the GHOST pipeline, when concatenating station data across time.





### 3.3.1 Data join

For each unique station (via station_reference), all associated stage 1 written files are gathered, and read into memory.

An assessment is first made if there are any data overlaps between any of the files through the time record. If no overlaps are found, then the data / metadata in the files is simply joined together. If any overlaps are found, the relevant periods and files are logged, and a stepped process is undertaken to determine which file should be retained in each overlap instance:

1. First, the overlap is attempted to be resolved by the number of measurements associated with the GHOST QA flag: "Cor-
525 rected Parameter" (code 24). This flag applies to measurements for which there is typically a known issue with the measurement methodology, and some type of correction has been applied to improve the accuracy of the measurement. The maximum number of measurements associated with the QA flag are taken across the conflicting files, and only files with equal to the maximum number of associated measurements are kept.

2. Second, priority data levels are used. Networks often publish the same data files multiple times, with continuously im-
530 proved QA, e.g. near real time, then with automatic QA, and finally with manual QA validation. Each type of data release is associated with a defined data level (stored in the "data_level" metadata variable), which are each given a hierarchical priority ranking. For example, EEA provide data in 2 separate streams: E1a (validated), and E2a (near real time). E1a is preferred to E2a in this case. The maximum ranking across the conflicting files is taken, and only files with that ranking are retained.

3. Third, the data revision date is used. Data files are often published with the same data level, but with different data revision
dates, with files often needing to be republished, after processing errors are identified and corrected. The data revision date is used to differentiate between these files. The latest revision date across the conflicting files is taken, and only files with that revision date are retained.

4. Fourth, a ranking algorithm is used. For each file, a number of weighting factors contribute a normalised ranking score between 1 and 2, which are then summed to give total ranking score. The file with the highest score is then selected. The
540 weighting factors considered in the ranking algorithm are as follows:

• Average temporal resolution in the overlap period. A finer temporal resolution (i.e. smaller number) gives a higher weighting.

• Number of valid measurement points in the overlap period (after screening by the GHOST QA flag: "Invalid Data Provider Flags – GHOST Decreed" (code 6)). A higher number gives a higher weighting.

• Measurement altitude. Designed to deal with instances where measurements are made on towers, simultaneously measuring components at different altitude levels. Lower measurement altitudes are given a higher weighting.

• Consistency of metadata in the overlapping files with that across all other files across the entire time record. A weighted score is calculated for each of longitude, latitude, altitude, measurement altitude, measurement methodology, and measuring instrument name variables. Files with values which occur more frequently over the time record are given a higher weighting.

After this, only files with summed rankings equal to the maximum score are retained.

5. Finally, if there are still 2+ remaining files for an overlap instance, some tiebreak criteria are used to select a file:





- First, by the maximum number of valid measurement points across the whole data files, i.e. not just the valid values for the overlap period (after screening by the GHOST QA flag: "Invalid Data Provider Flags – GHOST Decreed" (code 6)).

- Second, by the maximum number of non-NaN metadata variables provided in each data file.

• Finally, if there is still a tie, after sorting the filenames alphabetically, the first file is chosen.

After selecting a file in each overlapping period, the data / metadata in the files are simply joined together across the time record.

### 3.3.2    Resolve measurement position accuracy

After joining the data files, a consistent time series now exists for each station, however some irregularities may exist in the

stored metadata through the time record. This is of specific concern for the variables associated with the measurement position, i.e. longitude, latitude, altitude, sampling height, and measurement altitude.

In some instances, the level of accuracy of the network provided measurement position metadata, varies over time. This can cause significant ramifications, with the difference of a decimal place or two being able to significantly shift the subsequent evaluation of station data, e.g. placing a station incorrectly over the sea, or in an erroneous valley / peak in mountainous terrain.

Most of these instances are simply explained by errors in the creation of the data files, or due to the number of reported decimal places changing over time.

To attempt to rectify the majority of these cases a 2-step procedure is undertaken:

1. First, for each measurement position variable, all non-NaN values across the time record are grouped together within a certain tolerance ($0.0001° = {\sim}11m$ for longitude / latitude, 11m for altitude / sampling height / measurement altitude). Values

that are within the tolerance of at least 1 other position, would all be grouped together e.g. [10m, 17m, 21m]. However, without the 17m value, [10m] and [21m] would be in separate groups. The weighted modal measurement position in each group is then determined, using the number of sampled minutes that each metadata value represents as weights, and value of this position is then used to overwrite the original measurement position values in the group, through the time record.

2. Second, for each variable, all values which are sub-strings of any of the other positions across the time record, are grouped

together, e.g. 0.01 is a sub-string of 0.012322. In each group, an assumption is made that each sub-string is actually referring to the most detailed version of the position in the group, i.e. that with the most decimal places. If there are 2+ positions with the same maximum level of decimal places, then the position which represents the greater number of sampled minutes is chosen. This chosen position is then used to overwrite the original measurement position values in the group, through the time record.

In both steps, information is written to the "process_warnings" metadata variable, informing of the assumptions made in

these procedures.

### 3.3.3    Handle gapped key metadata

Generally speaking, the level of detail in the reporting of metadata has improved over time. This means in many cases, metadata variables that were not reported in the past, are now. In some instances, a metadata variable is inexplicably not included in a file,





when it was previously or subsequently reported, in most cases presumably due to a formatting error. As metadata is handled
dynamically in GHOST, both circumstances lead to gaps in the metadata variables, throughout the time record.

In most cases the provided metadata is constant over large swathes of time, therefore taking metadata reported previously or
subsequently in the time record can be justifiably assumed to be applicable for the missing periods. The missing metadata for
each variable is thus attempted to be filled. This is done by taking the closest non-NaN value going backwards in time for each
variable, or if none exists, then the closest non-NaN value going forwards in time. For positional metadata this stops stations
being separated out due to small inconsistencies through the time record (Sect. 3.3.5).

Some dependencies are required for this filling procedure for some metadata variables, to prevent incompatibilities in con-
current metadata variables, e.g. the documented lower limit of detection of a measuring instrument should not change if the
measuring instrument does not. These dependencies are defined in Table A13. Because of the importance of positional variables
being set (e.g. latitude), filling is attempted to be done through several passes, using progressively less stringent dependencies,
until ultimately requiring zero dependencies. The filling is not performed for any metadata variables that are highly sensitive
with time (these being the non-filled group in Table A13). If data is filled for any key variables, which are defined in Table
A12), a warning is written to the "process_warnings" variable.

### 3.3.4 Set altitude variables

The 3 GHOST measurement position altitude variables are all interconnected, in that the altitude + sampling height = measure-
ment altitude. A series of checks are performed to ensure this information is consistent through the time record, and modified
if not. For any variables that are modified, information is written to the "process_warnings" variable. Per metadata column, the
checks proceed as follows:

1. If all 3 altitude variables are set, i.e. non-NaN, then is is checked if all variables sum correctly. If not, the measurement
altitude variable is recalculated as altitude + sampling height.

2. If only 2 variables are set, the non-set variable is calculated from the others, e.g. altitude = 10m and sampling height =
2m, therefore measurement altitude is calculated to be 12m.

3. If only 1 variable is set, and it is the altitude or measurement altitude, then the other altitude variable is set to be equivalent,
i.e. altitude = measurement altitude, and the sampling height is set to be 0.

4. If no altitude or measurement altitude is set, then it is subsequently set using information from a digital elevation model
(DEM), detailed in Sect. 3.4.6.

### 3.3.5 Split stations by significantly changing measurement position

The final check in stage 2 is to determine if the measurement position of a station changes significantly through the time record,
i.e. one of the longitude, latitude, or measurement altitude changes. Where there are significant changes, the associated data /
metadata is separated out over the time record. Each separate grouping is then considered a new station, reflecting that the air
masses measured across the changing measurement positions, may be significantly different.



The unique measurement positions across the time record are firstly grouped within a certain tolerance ($0.0001° = \sim 11$m for longitude / latitude, and 11m for the measurement altitude), as in Sect. 3.3.2. Grouping like this ensures that if the measurement position changes, and then later reverts back to the previous position, then the associated data for the matching positions would be joined.

After the grouping process, some checks are performed to ensure that each of the groupings are of sufficient quality to continue in the GHOST pipeline:

1. If there are more than 5 unique groupings found, then the station is excluded from further processing, as the associated data is not considered to be trustworthy.

2. If any grouping has < 31 days of total data extent, then this group is dropped from further processing, as it is not considered
of sufficient relevance to continue processing.

3. For each grouping, if there are too many associated metadata columns per total data extent (<= 90 days per column), then the group is dropped from further processing, as the metadata is considered too variable to be trusted.

After these checks, if there is more than 1 remaining measurement position grouping, then the associated data / metadata is split, each associated with a new station_reference. The data which has the oldest associated time data retains the original
station_reference. Each chronologically ordered grouping after that is associated with a new station_reference, defined as "[station_reference]_S[N]", where N is an ascending integer starting from 1.

## 3.4  Add gridded metadata (Stage 3)

At this point in the pipeline, all station data / metadata for a component, reported by a given network, has been parsed, standardised, and concatenated, creating a complete time series per station. In the next three stages (3–5), the processed network
data is complemented through the addition of external information per station, giving added value to the dataset.

In many cases when observational data is used by researchers, it is used in conjunction with additional gridded metadata. This typically represents objective classifications, or measurements of some kind made over large spatial scales, i.e. typically continental to global. In some previous data synthesis efforts, some of the most frequently used gridded metadata in the atmospheric composition community were ingested, and associated per station.

GHOST follows this example, specifically looking to build upon the collection of metadata ingested by Schultz et al. (2017). A distinction made between the types of gridded metadata ingested, namely "Classification" and "Product", as outlined in Fig. 2. "Product" metadata is numeric in nature, whereas "Classification" metadata is not.

One key example of the added value of this gridded metadata, is when looking to filter out high altitude stations. When surface observations are used for model evaluation, it is typically desired to remove stations in hilly / mountainous regions,
as the models typically do not have the horizontal resolution to correctly capture the meteorological and chemical processes in these regions. The exclusion of stations is typically done by filtering out all stations above a certain altitude threshold, e.g. 1500m from mean sea level. This is a very simplistic approach, as it does not take into account the actual terrain at the stations, and means that low altitude stations which lie on very steep terrain are not removed, and high altitude stations which lie on flat plateaus are filtered out (e.g. much of the western US). A better approach would be to filter stations by the local terrain





type. There exist numerous sources of gridded metadata which globally classify the types of terrain, two of these ingested by GHOST being the Meybeck (Meybeck et al., 2001), and Iwahashi classifications (Iwahashi and Pike, 2007). Figure 5 shows these 2 classification types, in comparison with gridded altitudes from the ETOPO1 DEM. In areas such as southern and central Europe, the two terrain classifications indicate there is lots of very steep land, whereas the DEM indicates the majority of the land lies at relatively low altitudes (< 500m).

Table 9 shows a summary of the gridded metadata ingested in GHOST, with the associated temporal extents and native horizontal resolutions, per metadata variable. Table A11 provides more information about the ingested metadata, specifically spatial extents, projections, horizontal / vertical datums, and native file formats. All of the gridded metadata that are ingested in GHOST provide information on a global scale in longitudinal terms, but some do not provide full coverage to the poles, e.g. ASTER v3 altitude: -83:83°N.

The major processes involved in the association of gridded metadata in GHOST are described in the following subsections. As well as ingesting and associating gridded metadata per station, other globally standard metadata variables are also associated per station, i.e. reverse geocoded information and local timezones, described in Sect. 3.4.4 and Sect. 3.4.5 respectively.

Parallelisation is done per unique station (via station_reference) in the stage.



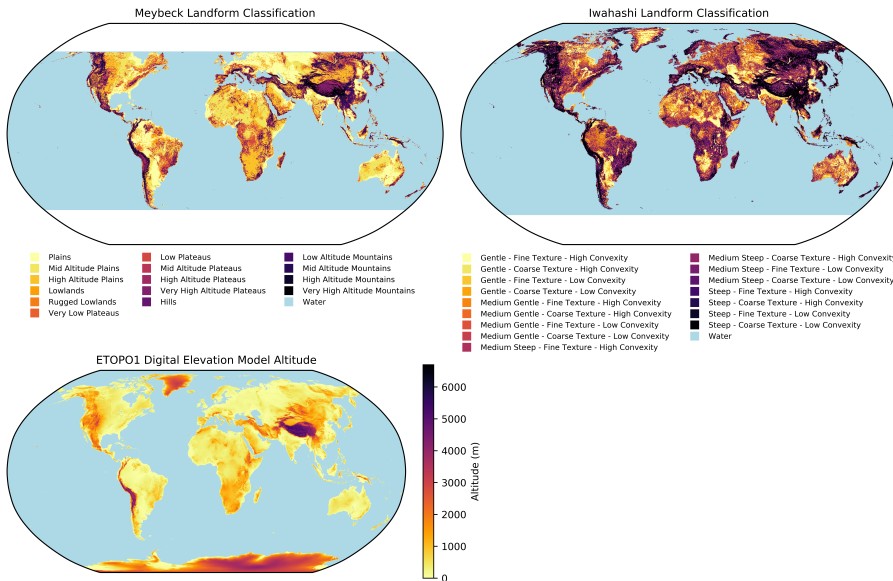

**Figure 5.** Comparison of the variety of gridded metadata available for the classification of terrain, ingested in GHOST. Shown are two landform classifications: Meybeck and Iwahasi, as well as the ETOPO1 DEM altitude.



**Table 9.** Summary of the gridded metadata which are ingested in GHOST. The temporal extent of each metadata type is given, as well as the native horizontal resolution of each type. More information is given in Table A11.

| Metadata Name | Temporal Extent | Resolution |
|---|---|---|
| ASTER v3 altitude (NASA et al., 2018) | 2000 – 2014 | 1" |
| ETOPO1 altitude (NOAA NGDC, 2009) | 1940 – 2008 | 1' |
| EDGAR v4.3.2 annual average emissions (Crippa et al., 2018; EC JRC and Netherlands PBL) | 1970, 1975, 1980, 1985, 1990, 1995, 2000, 2005, 2010, 2012 | 6' |
| ESDAC Iwahashi landform classification (Iwahashi and Pike, 2007; ESDAC) | 2007 | 30" |
| ESDAC Meybeck landform classification (Meybeck et al., 2001; ESDAC) | 2001 | 30" |
| GPW population density, v3: (CIESIN and CIAT, 2005), v4: (CIESIN, 2018) | v3: 1990, 1995 v4: 2000, 2005, 2010, 2015 | v3: 2.5' v4: 30" |
| GHSL built up area density (Corbane et al., 2018, 2019) | 1975, 1990, 2000, 2014 | 250m |
| GHSL population density (Freire et al., 2016; Schiavina et al., 2019) | 1975, 1990, 2000, 2015 | 250m |
| GHSL settlement model classification (Ehrlich et al., 2019; Pesaresi et al., 2019) | 1975, 1990, 2000, 2015 | 1km |
| GSFC coastline proximity (NASA OBPG) | 2009 | 36" |
| Koppen-Geiger classification (Beck et al., 2018) | 1980 – 2016 | 30" |
| MODIS MCD12C1 v6 IGBP land use (Friedl and Sulla-Menashe, 2015) | 2001, 2005, 2010, 2015, 2018 | 3' |
| MODIS MCD12C1 v6 UMD land use (Friedl and Sulla-Menashe, 2015) | 2001, 2005, 2010, 2015, 2018 | 3' |
| MODIS MCD12C1 v6 LAI (Friedl and Sulla-Menashe, 2015) | 2001, 2005, 2010, 2015, 2018 | 3' |
| NOAA-DMSP-OLS v4 nighttime stable lights (NOAA and US Air Force Weather Agency) | 1992, 1995, 2000, 2005, 2010, 2013 | 30" |
| OMI level3 column annual average $NO_2$ (Krotkov et al., 2017, 2019) | 2005, 2010, 2015, 2018 | 15' |
| OMI level3 column cloud screened annual average $NO_2$ (Krotkov et al., 2017, 2019) | 2005, 2010, 2015, 2018 | 15' |
| OMI level3 tropospheric column annual average $NO_2$ (Krotkov et al., 2017, 2019) | 2005, 2010, 2015, 2018 | 15' |
| OMI level3 tropospheric column cloud screened annual average $NO_2$ (Krotkov et al., 2017, 2019) | 2005, 2010, 2015, 2018 | 15' |
| WMO region (WMO, a) | 2013 | ——- |
| WWF TEOW terrestrial ecoregion (Olson et al., 2001) | 2006 | ——- |
| WWF TEOW biogeographical realm (Olson et al., 2001) | 2006 | ——- |
| WWF TEOW biome (Olson et al., 2001) | 2006 | ——- |
| UMBC anthrome classification (Ellis et al., 2010; University of Maryland Baltimore County) | 2000 | 5' |





### 3.4.1 Dynamic gridded metadata

For most of the gridded metadata types ingested in GHOST, the provided metadata is representative of an annual period, which is updated annually.

As with the network provided metadata, there is an conscious effort to capture the changes in the ingested gridded metadata across time. This is of specific importance for products directly affected by anthropogenic processes, e.g. land use or population density. However, processing gridded metadata for every year, in theory from 1970 to 2023, would place a major strain on the

675 processing workflow, therefore a compromise is needed to be struck. For each different gridded metadata type, the first and last available metadata years are ingested, as well as updates within this range, in years coinciding with the start and middle years of each decade, e.g. 2010, 2015. The specific ingested temporal extents for each type of gridded metadata are defined in Table 9. Each metadata column per station, is matched with the most temporally consistent gridded metadata, through the minimisation of the metadata column centre time, and gridded metadata centre extent time.

### 3.4.2 5km and 25km modal / average gridded metadata

The parsing and association of the gridded metadata per station, is in most cases done by taking the value of the gridcell in which the longitude and latitude coordinates of the station lie (i.e. nearest neighbour interpolation). Some gridded metadata is provided in non-uniform polygons, i.e. Shapefile and GeoJSON formats, adding additional complexity.

The extremely fine horizontal resolution of some of the ingested gridded metadata, e.g. 250m, means it may often be non-

685 comparable with data sources at coarser resolutions e.g. data from a global CTM. To help in situations such as this, for each ingested gridded metadata variable of fine enough horizontal resolution, extra variables are written taking the average, or mode in a 5km and 25km radius around the station coordinates. The mode is taken for "Classification" type variables, and the average is taken for "Product" type variables. No additional variables are created for gridded metadata which is natively provided in Shapefile and GeoJSON formats.

In order to calculate which gridboxes are taken into consideration in the modal / average calculations, perimeters 5km and 25km around the longitude and latitude coordinates are calculated geodesically, following (Karney, 2013). The percentage intersection of each gridcell with the perimeter is then calculated, i.e. how much of each gridcell is contained within the perimeter bounds?

When calculating the modal "Classification" variables, the class values are simply set as the class which appears most often

over all gridcells with an intersection > 0.0. When calculating the average "Product" variables, the weighted average is taken across all gridcells with an intersection > 0.0, using the percentage intersections as weights.





### 3.4.3   Coastal correction

Due to the nature of grids, stations which are located very close to the coast, could occasionally could fall in gridcells which are dominantly situated over water, and are thus associated with metadata which is not representative for the station. For the
regularly gridded "Classification" variables, a correction for this is attempted to be made.

In all cases where the metadata class is initially determined to be "Water", the modal class across the primary gridcell and its surrounding gridcells (i.e. share a boundary, including diagonally) is calculated, overwriting the initially determined class. If the primary gridcell is far from the coast, then the class will be maintained as "Water", but if it is close to the coast, then the set class will be more likely to be representative for the coastal station.

### 3.4.4   Reverse geocoded station information

Reverse geocoding is the process of using geographic coordinates to obtain address metadata. The Python reverse_geocoder package (Thampi) provides a library which provides this function. Specifically, for each provided longitude and latitude coordinate pair, metadata is returned for the following variables: "city", "administrative_country_division_1", "administrative_country_division_2", and "country". This is extremely useful, as it allows station address metadata to be standardised across
the networks.

In some cases, when stations are extremely remote, the returned search information is matched to a location extremely far from the original coordinates. To guard against such instances, the matched location is required to be within a tolerance of 5° of the station longitude and latitude.

### 3.4.5   Local timezone

As well as using the station coordinates to obtain standard address metadata, they can be used to obtain the local timezone. This is done by passing a station longitude and latitude coordinate pair to the Python timezonefinder package (Michelfeit). This returns a local timezone string, referencing the IANA time zone database (IANA), which is saved to the "station_timezone" metadata variable.

In some cases if the station is extremely remote, the timezonefinder package will not be able to identify a local timezone.
In these cases, the closest timezone is attempted to be identified within a set radius around the station, initially set as 1°. If no timezones are identified within this initial radius, the radius size is increased iteratively by 1°, until a timezone is found. This iteration is allowed to continue for 1 minute before timing out, and the station timezone is left unset.

If the timezonefinder package is used to obtain the local timezone in order to shift local time measurements to UTC (see Sect. 3.2.3), this of course carries some uncertainty, and thus any measurements shifted in such a fashion are accompanied with
the GHOST QA flag: "Timezone Doubt" (code 61).



### 3.4.6   Set missing altitude metadata using DEM

As referenced in Sect. 3.3.4, if no altitude or measurement altitude is set through the time record for a station, then it is set using information from a DEM.

This is first attempted to be done by taking altitudes from the ASTER v3 DEM (NASA et al., 2018). Missing altitude variable
metadata (i.e. NaN) is simply overwritten with the station specific ASTER v3 altitude. If sampling height is non-NaN, then the measurement altitude is set as the ASTER v3 altitude + sampling height, otherwise it is set as simply the ASTER v3 altitude.

Because ASTER v3 is only available between -83:83°N, there are some polar stations which would not be able to be handled. In these cases, the ETOPO1 DEM altitude (NOAA NGDC, 2009) is then used instead. ASTER v3 is preferred to ETOPO1, simply because it has a finer horizontal resolution (1" vs 1'). A warning is written to "process_warnings" to inform of any
assumption of altitude metadata through this process.

The ASTER v3 DEM, is also used to flag potential issues with network reported altitudes. This is determined whenever a reported station altitude is >= 50m different, in absolute terms, from the ASTER v3 station altitude, setting the GHOST QA flag: "Station Position Doubt – DEM Decreed" (code 40).

### 3.4.7   WIGOS link

In an effort to link GHOST with existing frameworks for storing atmospheric science data, a substantial effort was made to connect with WIGOS (WMO, 2019a, 2021). WIGOS is the framework employed for all WMO observing systems, and defines metadata standards for many variables (WMO, 2019b), of which there is a considerable overlap with those defined in GHOST.

All stations for which data is reported in a WMO observing system are associated with a WIGOS station identifier (WSI). Through the assistance of WMO, all stations in GHOST are cross-checked to see if they have an associated WSI. Any identified
WSIs are set in the "WIGOS_station_identifier" variable.

Any GHOST metadata variables which are equivalent (or very closely related) to a WIGOS metadata variable, will be accompanied with an attribute in the finalised netCDF: "WIGOS_name", which gives the respective name of the variable within WIGOS.

Some WIGOS variables are constant over the time record, e.g. "ApplicationArea". These variables are set as global attributes
in the finalised netCDF.

If the processed component is defined as one of the fields for the "ObservedVariableAtmosphere" WIGOS variable, then the relevant "WIGOS_name" and "WIGOS_number" is saved with the component data variable, as attributes, in the finalised netCDF.

### 3.5   Quality assurance (Stage 4)

The filtering of data by network QA flags goes a long way towards providing reliable measurements, however there are many instances where clearly erroneous or extreme data remains unfiltered. The level of detail of the network QA also varies greatly across the networks, with some networks not providing any QA whatsoever. For these reasons, a wide variety of GHOST own





QA checks are performed, returning GHOST QA flags. This attempts to ensure a minimum level of QA is associated with all measurements.

GHOST QA flags, as numeric codes, are written per measurement to the "qa" data variable. Some of these flags have already been described in previous sections, see Sect. 3.2.5 for some basic flag type definitions, Sect. 3.2.8 for measurement process flags, Sect. 3.2.9 for limit of detection and measurement resolution flags, Sect. 3.2.13 for sample gas volume flags, and Sect. 3.4.6 for positional metadata doubt flags.

Table 10 summarises the different types of GHOST QA flags, together with the number of associated flags per type. These

765 QA types range from "basic", e.g. checking for NaNs, negative values, zeros; to more advanced types such as the "monthly distribution consistency", classifying the consistency of monthly data across the years. Specific definitions for each GHOST QA flag are given in Table A9, and some of the more advanced flags are described in greater detail in the following subsections.

After all GHOST QA checks have been performed, some default GHOST QA is used to filter measurements, creating a prefiltered version of the measurements.

Parallelisation is done per unique station (via station_reference) in the stage.





**Table 10.** Summary of the GHOST QA flag types, stored in the "qa" variable. Each QA flag is derived from GHOST own quality control checks. For each type, a description is given, as well as the number of flags associated with each type. Definitions of the individual flags are given in Table A9.

| Flag Types | N Flags | Description |
|---|---|---|
| basic | 9 | Flags associated with basic data validity checks. |
| measurement process | 15 | Flags which indicate issues with measurement processes found when standardising measurement metadata. |
| sample gas volume | 4 | Flags which indicate if the sample gas volume is unknown, or has been assumed. |
| positional metadata doubt | 2 | Flags which indicate doubt regarding the validity of the metadata stated station position. |
| data product | 2 | Flags associated with the process of calculating data from multiple components. |
| local conditions | 5 | Flags which indicate different kinds of local measurement conditions, aggregated from network QA flags. |
| timezone | 2 | Flags which indicate irregularities with the reported data timezone. |
| limit of detection | 6 | Flags which indicate data that exceeds limits of detection. |
| measurement resolution | 4 | Flags which indicate data is of a coarse resolution. |
| recurring values | 3 | Flags which indicate data is recurring to some extent. |
| monthly fractional unique values | 7 | Flags which indicate the percentage of unique data values per month. |
| data outliers | 6 | Flags which indicate data is outlying in some aspect. |
| monthly distribution consistency | 14 | Flags which indicate how consistent a monthly distribution of measurements is with other distributions for the same month, across the years. |





### 3.5.1 Monthly adjusted boxplot

Data outliers are very obvious to the human eye, however detecting these extremities using a computer algorithm can be challenging. There are a number of well documented parametric methods for the detection of outliers, however there exist a vast range of distributions across the hundreds of different components processed within GHOST, thus a non-parametric method is required.

Tukey's boxplot (Tukey, 1977) is one such method. The method results in the definition of two sets of fences, on both the lower and upper ends of the distribution, termed the inner and outer fences. Where observations exceed the inner fence they are considered possible outliers, and where they exceed the outer fence they are considered probable outliers. The lower and upper inner fences are set as:

$$[Lif, Uif] = [Q1 - (IQR \cdot 1.5), Q3 + (IQR \cdot 1.5)], \tag{6}$$

where $Lif$ is the lower inner fence, $Uif$ is the upper inner fence, $Q1$ is the 25th percentile, $Q3$ is the 75th percentile, and $IQR$ is the interquartile range.

The lower and upper outer fences are set as:

$$[Lof, Uof] = [Q1 - (IQR \cdot 3.0), Q3 + (IQR \cdot 3.0)], \tag{7}$$

where $Lof$ is the lower outer fence, and $Uof$ is the upper outer fence.

Statistically speaking, for a Gaussian distribution, 0.7% of data will lie beyond the inner fences, and 0.0002% beyond the outer fences. The method works well for the detection of outliers when the data distribution is symmetric, however with asymmetric distributions, the fences end up being set too either too low or high, depending on the skew of the distribution.

Hubert and Vandervieren (2008) proposed an adapted method to overcome this problem, the adjusted boxplot. They attempted to adjust Tukey's technique with the use of a robust measure of skewness, the medcouple. However, this erroneously extended the fences on the skewed side of the distribution, meaning some clear outliers were not flagged. Adil and Irshad (2015) provided a solution for this, with the lower and upper inner fences set as:

$$[Lif, Uif] = [Q1 - 1.5 \cdot IQR \cdot e^{-SK \cdot |MC|}, Q3 + 1.5 \cdot IQR \cdot e^{SK \cdot |MC|}], \tag{8}$$

where $SK$ is the classical skewness, $MC$ is the medcouple. A restriction is imposed in the calculation of $SK$, capping it at a maximum of 3.5, preventing the fences to be erroneously extended for the case of a highly skewed distribution.

The lower and upper outer fences are set as:

$$[Lof, Uof] = [Q1 - 3.0 \cdot IQR \cdot e^{-SK \cdot |MC|}, Q3 + 3.0 \cdot IQR \cdot e^{SK \cdot |MC|}] \tag{9}$$

This corrected adjusted boxplot method is independently applied to each month of station data (by UTC month). Restricting the application of the method to just one month of data ensures that any impact from the seasonal and interannual variation of measurements is limited. Data is pre-screened by other GHOST QA flags (defined in Table A14), to ensure a minimum





level of data quality before the method is applied. The method does not work well with very low number of data points, so a minimum of 20 remaining values after pre-screening is conservatively required to apply the method. Measurements exceeding the inner and outer fences are associated with the GHOST QA flags: "Possible Data Outlier – Monthly Adjusted Boxplot", and "Probable Data Outlier – Monthly Adjusted Boxplot" respectively (codes 114 and 115).

Figure 6 shows the application of the method to hourly $NO_2$ data from a suburban Spanish station, Penausende, in comparison with the application of the Tukey boxplot. Due to the left-skewed distribution of the data, Tukey's boxplot sets both the lower and upper fences too low, incorrectly flagging a large number of measurements on the upper end of the distribution. The advantage of the adjusted boxplot is seen in comparison, with the fence construction taking into account the skew of the distribution, meaning only measurements which are obviously outlying to the eye are flagged.



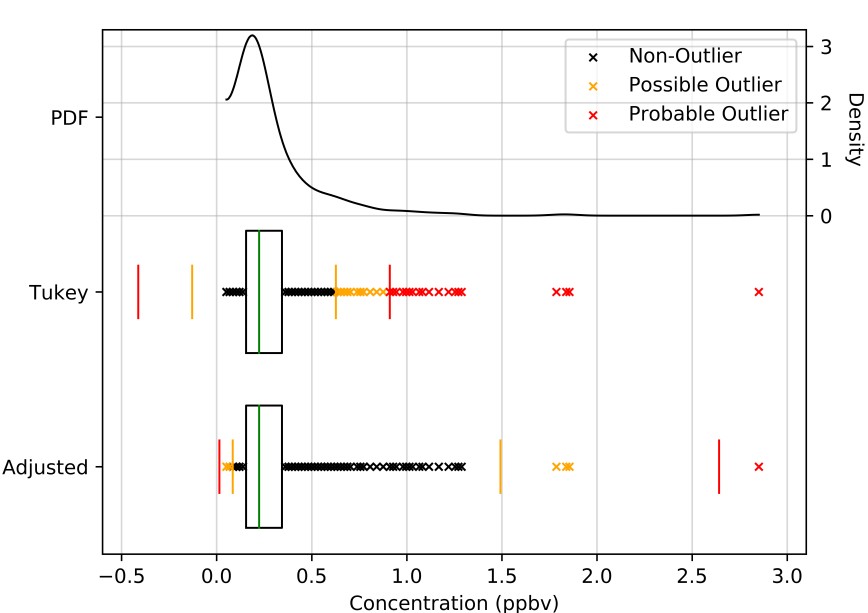

**Figure 6.** Illustration of the determination of possible (orange) and probable (red) data outliers using the Tukey boxplot and adjusted boxplot methods, for hourly $NO_2$ data in January 2018 at the suburban ES0013R_CL(IPC) station, Penausende, Spain. Also shown is the probability density function of the data in the month.



### 3.5.2 Monthly distribution consistency

Data outliers are most commonly thought of as values which are far from all other values, however, data can also be outlying as a collective. For example, the measurements in the month of July one year being significantly different to the collections of measurements in all previous Julys. These types of outliers can be entirely real in origin, e.g. driven by extreme meteorological conditions, or can be erroneous, e.g. due to measurement issues. In either case, these types of outliers should be flagged in some way.

One way of checking for these outliers is looking how the data distribution for one specific month, e.g. July 2016, at a station compares with the distributions for the same month, e.g. July, across the years. If one month's distribution is extremely different from the typical monthly distribution, then this is obviously suspicious, and should be flagged. The efficacy of this method is affected by long-term trends changing the station's distribution over time, but the impact of this can be constrained by only comparing against distributions in a limited range of years. Additionally, the variability of the distributions over time may vary significantly from station to station, which needs to be accounted for.

To allow for the quantification of the comparison of data distributions in different months, kernel density estimation is used to estimate the probability density function (PDF) of the data in each month. The intersection of the PDFs of two separate months can be used to objectively measure the consistency of monthly data distributions. An intersection score between 0.0 and 1.0 is returned, 0.0 being no intersection, and 1.0 being a perfect intersection. A PDF is only estimated for any given month when there are >= 100 valid values after screening by other GHOST QA flags (defined in Table A14), and when there is a minimum of 3 unique values in the month, to ensure there are sufficient values of quality to estimate the PDF.

It is attempted to estimate the consistency of the distribution for one specific month, termed the target month, with the distributions for the same month (e.g. July) across the years. By calculating the intersections of the PDF for the target month with PDFs of the same month in the surrounding ±5 years, a metric for the short term consistency of the target month is obtained. This is calculated by:

$$C_{ST} = 1.0 - \widetilde{I}, \tag{10}$$

where $C_{ST}$ is the short term consistency, and $\widetilde{I}$ is the median intersection of the PDF for the target month with PDFs of the same month in the surrounding ±5 years.

The short term consistency ranges between 0.0 and 1.0. A score of 0.0 indicates that the target month's data is perfectly consistent with a typical month, and a score of 1.0 indicates that it has no consistency with a typical month.

If the PDF for the target month cannot be estimated, or there are less than 2 estimated PDFs in total across the surrounding years, then there is not enough information to accurately assess the consistency of the target month's data, and a GHOST QA flag is written informing of this: "Monthly Distribution Consistency – Unclassified" (code 130).

By calculating the median short term consistency of the same month as the target month (e.g. July) over the time record, a measure for the standard consistency is obtained. When referenced against the short term consistency, this gives a metric for the deviation of the short term consistency from the standard consistency, termed the deviation of consistency. This is calculated



by:

$$C_D = \widetilde{C}_{ST} - C_{ST}, \tag{11}$$

where $C_D$ is the deviation of consistency, and $\widetilde{C}_{ST}$ is the median short term consistency of the same month over the time record, termed the standard consistency.

The deviation of consistency is normalised after calculation. If the score is less than 0.0 then it is set to 0.0, that is to say, any case where the short term consistency for the target month is equal to or greater than the standard consistency. Next, the score is scaled to be a ratio to the standard consistency. The deviation of consistency ranges between 0.0 and 1.0. A score of

0.0 indicates that the short term consistency is equal to or greater than the standard consistency, and a score of 1.0 indicates that the short term consistency is as far below the standard consistency that it can possibly be.

Finally, the short term consistency and deviation of consistency are summed to give a final consistency score for the target month:

$$C = C_{ST} + C_D, \tag{12}$$

where $C$ is the consistency score.

The consistency score ranges between 0.0 and 2.0, where 0.0 indicates that the target month has an extremely typical distribution, and 2.0 indicates that the target month has an extremely atypical distribution. The score is split into 10 zones (in range increments of 0.2), from the most typical distributions in Zone 1 (score of 0.0 to 0.2), to the most atypical distributions in Zone 10 (score of 1.8 to 2.0). All months for which a consistency score can be determined are associated with the appropriate

GHOST QA flag: "Monthly Distribution Consistency – Zone [N]" (codes 120-129), where [N] is the zone number of the consistency score. If 2/3, 4/6, or 8/12 consecutive months are classed as zone 6 or higher, then it is suspected there is a systematic reason for the atypical distributions, and the entire periods are flagged with the appropriate GHOST QA flags: "Systematic Inconsistent Monthly Distributions – 2/3 Months >= Zone 6" (code 131), "Systematic Inconsistent Monthly Distributions – 4/6 Months >= Zone 6" (code 132), and "Systematic Inconsistent Monthly Distributions – 8/12 Months >= Zone 6" (code 133).

Figure 7 visually describes this classification procecdure for hourly $O_3$ data at a rural background station, Cape Verde, for 2 different months: July 2009, and July 2012. The distribution of data in July 2009 is markedly different to the July data of the surrounding years, whereas the distribution in July 2012 is very similar to the surrounding years. July 2009 is classified as being zone 10, an extremely atypical July, whereas July 2012 is classified as zone 2, a very typical July.

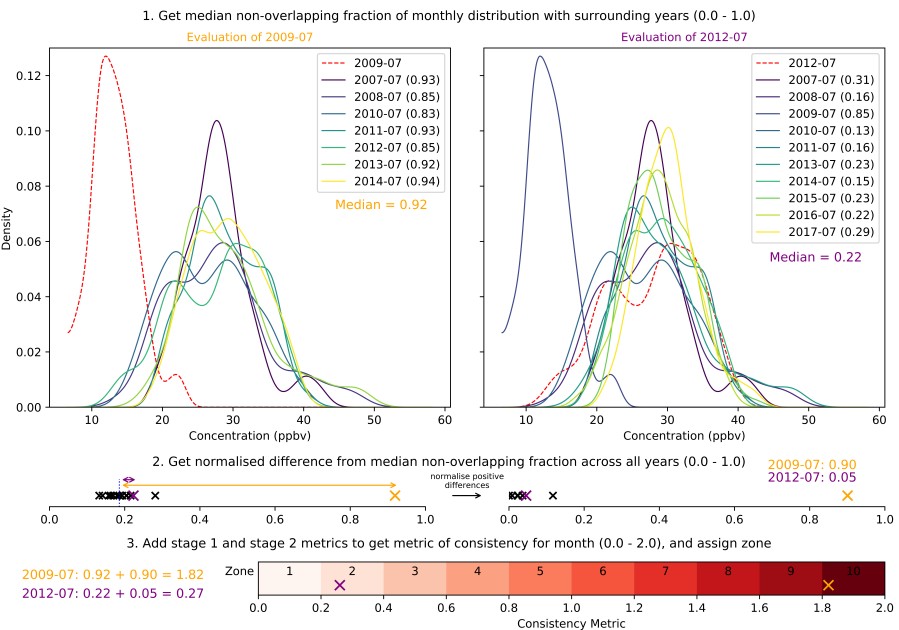

**Figure 7.** Illustration of the procedure for classifying the consistency of a monthly distribution of measurements with other distributions for the same month, across the years. The classification is demonstrated for hourly $O_3$ data at the rural background CV0001G_UVP station, Cape Verde, in 2 different months: July 2009 and July 2012. The distribution of data in July 2009 is markedly different to the July data of the surrounding years, whereas the distribution in July 2012 is very similar to the surrounding years. July 2009 is classified as being zone 10, an extremely atypical July, whereas July 2012 is classified as zone 2, a very typical July.



### 3.5.3    Prefilter data by default GHOST quality assurance

Although the extensive number of GHOST and network QA flags give users a wealth of options for filtering data, in many cases users simply want reliable data, with no major outliers, without having to worry about how to filter data. Therefore, such an option is provided, prefiltering data by some default GHOST QA, defined in Table A10. These QA flags are conservatively chosen, intended to remove only probable invalid values, therefore greater filtering may be required to solve other data issues. This is saved to the data variable "*GHOSTcomponentname*_prefiltered_defaultqa", where *GHOSTcomponentname* is the

standard GHOST name for the component, as defined in Table 2.

### 3.6    Add temporal classifications (Stage 5)

When evaluating station data, to better understand the driving temporal processes at play, it is common to screen data by some form of temporal classifications e.g. day / night. Thus, to streamline this process for end users of GHOST, some of the most widely used temporal classifications are calculated, and associated with station measurements. These are namely: day / night,

weekday / weekend, and season classifications.

These temporal classifications are added as data variables, with integer classification codes per measurement. Table 11 details the different temporal classification types, with a definition of a the class codes, and a description of the procedure used to calculate each of the classes. Whenever a temporal classification cannot be calculated, either because the temporal resolution is too coarse, or the local timezone is unknown, a fill value (255) is set instead.

Parallelisation is done per unique station (via station_reference) in the stage.



**Table 11.** Summary of the temporal classification data variables in GHOST. For each variable, the associated classification codes, calculation requirements, and the procedure for calculation are given.

| Data Variable | Class Codes | Calculation Requirements | Calculation Procedure |
|---|---|---|---|
| day_night_ code | Day = 0, Night = 1 | Known local timezone for station, and temporal resolution < 1 day | 1. The centre of each relevant measurement window is shifted to local time. 2. The solar elevation angle is calculated for each local time, at the station's location (longitude, latitude, and measurement altitude), using the Python ephem package (Rhodes). 3. Day = Solar elevation angle > 0.0° Night = Solar elevation angle <= 0.0° |
| weekday_ weekend_ code | Weekday = 0, Weekend = 1 | Known local timezone for station, and temporal resolution < 1 day | 1. The centre of each relevant measurement window is shifted to local time. 2. The day of the week for each local time is determined. 3. Weekday = Monday, Tuesday, Wednesday, Thursday, Friday Weekend = Saturday, Sunday |
| season_code | Spring = 0, Summer = 1, Autumn = 2, Winter = 3 | Temporal resolution < 31 days | 1. The month for the UTC centre of each relevant measurement window is determined. 2. The hemisphere of the station is determined using the latitude. NH = Northern Hemisphere, SH = Southern Hemisphere. 3. Winter = December, January, February (NH) / June, July, August (SH) Spring = March, April, May (NH) / September, October, November (SH) Summer = June, July, August (NH) / December, January, February (SH) August = September, October, November (NH) / March, April, May (SH) |





## 3.7 Temporally average data (Stage 6)

At this point in the pipeline, all reported station data / metadata for a component, for a given network, has been standardised, concatenated, and complemented with gridded metadata, GHOST QA, and temporal classifications. As measurements of all temporal resolutions are processed in GHOST (e.g. 30 minutes, 6 hours), the data for each station can be composed of a variety of temporal resolutions.

In this stage, station measurements are temporally standardised, temporally averaging data to standard temporal resolutions, namely: hourly, hourly instantaneous, daily, and monthly. Other data variables e.g. data flags, temporal classifications, are also temporally standardised.

Data variables informing on the representativity of the temporal averaging are also created, providing the percentage representativity of the native measurements that go into each temporal average. As well as having measurements associated with UTC, measurements are also associated with other reference times, namely, mean solar time, and local time.

Parallelisation is done per unique station (via station_reference) and standard temporal resolution (e.g. hourly, daily) pairings in the stage.

### 3.7.1 Temporal averaging procedure

First, station measurements with a coarser temporal resolution than the standard temporal resolution being averaged to, are dropped, e.g. monthly resolution measurements are dropped when processing hourly averages. Stations with no remaining data after this are excluded from further processing, for the particular standard temporal resolution.

Next, a regular grid of times between January 1970 and January 2023 UTC is created, with the spacing between each time being the relevant standard temporal resolution, e.g. for a monthly resolution: 1970-01-01 00:00, 1970-02-01 00:00, 1970-03-01 00:00, etc. These times are the start times of the temporally standardised measurements, that will be written out in the finalised netCDF4 file, as the "time" data variable. Each consecutive pair of times represent the start point and end point of each measurement, termed the standard measurement windows.

For some components, measurements are representative of a moment in time, rather than an average over time. All components that are not in the gas and particulate matrices, i.e. aerosol optical properties, have measurements which are instantaneous in nature. Measurements of this type are therefore extremely time sensitive, and averaging these measurements without care could result in nonsensical output. For example, when calculating hourly averages, instantaneous measurements at 00:01 and 00:59 would be averaged together, despite measurements being 58 minutes apart, and potentially being extremely different. To combat this, the hourly instantaneous resolution is added, for all instantaneously measured components. For this resolution, the standard measurement windows are adjusted to be centred around the top of the UTC hour, e.g: 1970-06-01 06:30 – 1970-06-01 07:30, 1970-06-01 07:30 – 1970-06-01 08:30. Rather than taking an average of the native measurements in each measurement window, the value closest to the top of each UTC hour is be taken to represent the window.

The temporal standardisation process is now started. The standard measurement windows are iterated through, chronologically, and in each window a value for every data variable is set e.g. measurements, data flags, temporal classifications. How



these values are set depends on the number of native resolution measurements that overlap with each standard window. A native measurement can be entirely contained within a window, can be equivalent to the window (i.e. same start and end point), or can lie across the bounds of two, or more, windows.

If zero native measurements lay in a window, then the measurement value of the window is set as NaN. For the "qa" variable, the value is set to be the GHOST QA flags that were set in the last window with a valid measurement, plus the "Missing Measurement" flag (code 0). This is done to ensure the GHOST QA flags do not jump wildly through the time record, but makes the assumption that the previously set flags are still applicable for the current window. All other data variable values are set to be NaN.

If there is just one native measurement in the window, then that measurement is taken to represent the entire window. The other data variables are also taken as they are.

If there is more than one native measurement in the window, then a procedure is undertaken to assign a measurement value for the window, as well as to assign values for the other data variables:

1. Invalid native measurements are first screened out using a defined set of GHOST QA flags, defined in the "Invalid QA" grouping in Table A15. This tries to ensure that any temporal average is not biased by erroneous data. The reciprocal values of the invalid native measurements across the other data variables are also screened out.

2. If there are zero remaining native measurements after screening, then for the hourly instantaneous resolution, the filtering is unapplied to ensure a value will be set for the window. For non-instantaneous resolutions, the measurement value of the window is set as NaN. For the "qa" variable, the value is set to be the GHOST QA flags that were set in the last window with a valid measurement, plus the "No Valid Data to Average" flag (code 8). All other data variable values are set as NaN, and processing proceeds to the next standard measurement window.

3. If there are remaining native measurements after screening for the hourly instantaneous resolution, the measurement closest to the UTC hour is simply taken to be value for the window. The reciprocal value of the chosen measurement in each other data variable is taken to set their values, and processing proceeds to the next standard measurement window.

4. If there are remaining native measurements after screening for non-instantaneous resolutions, the measurement value is set by taking a weighted average of the measurements in the window, with the weights being the number of minutes represented in the window per measurement. Values for the variables "reported_uncertainty_per_measurement", and "derived_uncertainty_per_measurement" are also calculated in the same way, after excluding NaNs.

5. For the "qa" variable, GHOST QA flags that were use to screen measurements in step 1 are dropped. Other flags are kept if they appear more often than not in the window (i.e. modally). These other flags are defined in the "Modal QA" grouping in Table A15.

6. For the "flag" variable, all network QA flags are dropped as these have already been indirectly filtered by the GHOST QA flag: "Invalid Data Provider Flags – GHOST Decreed" (code 6), in step 1. The "Valid Data" flag (code 0) is then set solely for the window.





7. For each of the the "day_night_code", "weekday_weekend_code", and "season_code" variables, the weighted mode over the respective codes in the window is taken to set their value, with the weights being the number of minutes represented in the window per associated measurement.

After all standard measurement windows have been iterated through, station data has been completely temporally standard-
965 ised.

### 3.7.2 Calculate temporal representativity

In parallel to the temporal averaging procedure, calculations of the temporal representativity of the native measurements across a variety of temporal periods are made. This is done as it is very useful, and often important, to know the representativity of the native measurements used for creating temporal averages. The different temporal periods evaluated are: hourly, daily, monthly,
and annual. The representativity is only calculated for periods as coarse or finer than the standard temporal resolution, e.g. for monthly averaged measurements, the evaluated periods would be monthly and annual.

All of the evaluated periods begin and end on UTC boundaries, and start in January 1970, going through to January 2023. For example, for the hourly period, 1970-01-01 00:00 – 1970-01-01 01:00 UTC and 1970-01-01 01:00 – 1970-01-01 02:00 UTC, would be the first two hourly periods evaluated.

For each temporal period, two metrics of representativity are calculated. The first metric is data completeness, i.e. the percentage of the relevant period that is represented by native measurements. The second metric is the maximum data gap, i.e. the percentage maximum data gap in the relevant period that is filled by native measurements, relative to the total period length. All representativity percentages are returned as rounded integers (0–100 %).

If the temporal resolution is hourly instantaneous, the representativity calculations are modified slightly. Rather then calcu-
980 lating the representativity over the total period, it is calculated as the percentage of all standard temporal resolution windows inside the relevant period that contain native measurements.

The calculated representativity variables are written to data variables with the syntaxes: "[period]_native_representativity_ percent", and "[period]_native_max_gap_percent", where [period] is replaced by the relevant temporal period, e.g. annual. All representativity variables are saved in the standard temporal resolution, e.g. if the standard temporal resolution is hourly, and
985 the evaluated temporal period is annual, then each annual UTC period is divided into hourly chunks, and all chunks assigned the calculated representativity metric for the annual period.

### 3.7.3 Local and mean solar time

As well as having measurements referenced to UTC, it is often useful to have measurements referenced to different time standards. As referenced previously, time manipulation is often a non-trivial affair, and to ensure end users do not need to
990 calculate this, station measurements are referenced against two other widely used other time standards: local time, and mean solar time.

Local time is defined simply as the local time at each station at the time of measurement. This is calculated by converting the standard UTC times using the pytz Python package (Bishop), fed with the local timezone determined in Sect. 3.4.5. The





calculated times are written to the "local_time" data variable. Unlike the standard UTC "time" variable, these times vary per
station.

Solar time is defined as the time measured by the Earth's rotation relative to the Sun. Apparent solar time is determined by
direct observation of the Sun, whereas mean solar time is time that would be measured by observation if the Sun traveled at
a uniform apparent speed throughout the year, rather than slightly varying across the seasons. More technically, it is defined
as the hour angle of the mean Sun plus 12 hours. The hour angles of each of the standard UTC times are calculated using
the Python ephem package (Rhodes), and station longitude. The calculated times are written to the "mean_solar_time" data
variable. These times also vary per station.

### 3.7.4   Station netCDF creation, per year and month

At this point, the associated data per station has been temporally standardised, and is ready to saved to its finalised form. Station
data, per standard temporal resolution, is grouped per year and month. Due to GHOST metadata being dynamic, it is possible
for there to be multiple values associated with a metadata variable in a month. For the purpose of simplicity, it is decided to
limit the number of values associated with each metadata variable in a month to just one. If there is more than one unique value
for any metadata variable in a month, then the value which is representative of the greater number of minutes in the month is
chosen to represent the variable. The data and metadata in each group is then written to a station specific netCDF4 file, for the
relevant year and month. Station specific files are written are for all year and month groups which contain station data.
All information associated with the data and metadata variables written in the netCDF4 files e.g. variable names, data types,
is defined in Tables A1 and A2 respectively.

### 3.8   Monthly aggregation by station (Stage 7)

Once all station specific netCDF4 files have been written for a network and component pair, the last remaining task is to aggre-
gate the files. All station specific netCDF4 files of the same standard temporal resolution, per year and month, are aggregated
into one netCDF4 file using NCO (The NCO Project). The resultant filenames have the syntax:
"*GHOSTcomponentname*_YYYYMM.nc", where *GHOSTcomponentname* is the standard GHOST name for the component,
as defined in Table 2. This is the finalised form of the GHOST data that is separated by network.

Parallelisation is done per year and month, and standard temporal resolution pairings in the stage.

### 3.9   Cross network synthesis (Stage 8)

At this point in the pipeline, finalised netCDF4 files for a component, for all standard temporal resolutions, across all networks
have been written. In order to maximise the usefulness of GHOST, with model evaluation specifically in mind, component data
across all networks is synthesised, resulting in a unified "network". This synthesis is done per year and month, per standard
temporal resolution.





During this process, any duplicate stations across networks are identified, and one is kept preferentially. The preference is made by prioritising some networks over others, with these determinations made using the experiences gleaned while processing data from each of the individual reporting networks in this work. These network preferences are not disclosed here out of respect to the data providers.

Identifying duplicate stations is done by geographically matching stations within a tolerance of 19.053 m. This tolerance is calculated by allowing for a tolerance of 11m in each of the 3 independent x, y, z dimensions, as is done in stage 2 of the GHOST pipeline to distinguish unique stations. Station longitudes, latitudes and measurement altitudes are converted to Earth-Centered, Earth-Fixed (ECEF) coordinates, and the distances between all stations are then calculated. Any geographically matched stations which use different measurement methods are not classed as duplicates.

The resultant filenames have the same syntax as the finalised network specific files described in Sect. 3.9, but are saved under the synthesised "network" name: "GHOST-PUBLIC".

Parallelisation is done per year and month, and standard temporal resolution pairings in the stage.

## 4 Finalised datasets

In this section, the file structure of the finalised GHOST dataset is detailed, and the temporal and spatial data extent for some key variables is described.

The GHOST dataset is made freely available via the following repository: https://doi.org/10.5281/zenodo.10637449 (Bowdalo, 2024).

The dataset consists of 7,275,148,646 total measurements from 1970–2023, of 227 different components, from 38 reporting networks.

Data is available in two forms. The first is separated out per network, per component. The second form is a synthesis across networks, per component, saved under the "GHOST-PUBLIC" name. Data is saved for both forms as netCDF4 files, per year and month, and in 4 different temporal resolutions: hourly, hourly instantaneous, daily, and monthly. The dataset includes data from all networks that we have the right to redistribute, indicated in the data rights column of Table 1.

Figure 8 shows the temporal data availability in GHOST of 4 key components: $O_3$, $NO_2$, CO, and total PM10. The evolution of the number of stations, per network, is shown across the time record (for monthly resolution data). The earliest measurements made for $O_3$ are in 1970, from the Japan NIES network. In general, the total number of stations has increased steadily across time for all components, however there is a large differential in the station numbers across the networks. The networks with the largest station numbers are those which exist for regulatory purposes, i.e. those which exist to monitor compliance with national or continental air quality limits (e.g. EEA AQ e-Reporting, Japan NIES, US EPA AQS).

In 2012 there was a major transition in the reporting framework of the major European database which exists to monitor air quality compliance of EU member states. The framework name changed from EEA AirBase to EEA AQ e-Reporting, and is treated in GHOST as two separate networks. Thus, this crossover is evident in Figure 8, as EEA AQ e-Reporting station numbers ramp up over 2012, and EEA AirBase goes offline in 2013.



For $O_3$, there is a clear seasonal trend in the number of stations from the US EPA AQS network, with the numbers increasing in the summer, and then decreasing in the winter. This is because the stations in US EPA AQS primarily monitor $O_3$ to check for air quality compliance, which is typically only of concern in the summer, when more light is available to drive $O_3$ production. Interestingly, the number of stations for CO and PM10 in the US EPA AQS network have dropped significantly since the 1990s.

Figure 9 shows the spatial data availability in GHOST of the same key 4 components, across the entire 1970-2023 time range, i.e. the unique stations per network, over the time record. There is excellent spatial coverage in North America, Europe, and Eastern Asia, across the components. However, there are consistent spatial gaps over Africa, central Asia, and South America (excluding Chile). In general, there is a large disparity between the number of stations in the northern hemisphere and the southern hemisphere. This disparity is less prevalent for CO, with the inclusion of flask samples from the WMO GAW WDGGG network providing excellent spatial coverage. Stations in networks which exist to measure rural background concentration levels (e.g. US EPA CASTNET) are far less densely distributed than they are in regulatory networks (e.g. US EPA AQS), where stations are mostly located in urban areas.



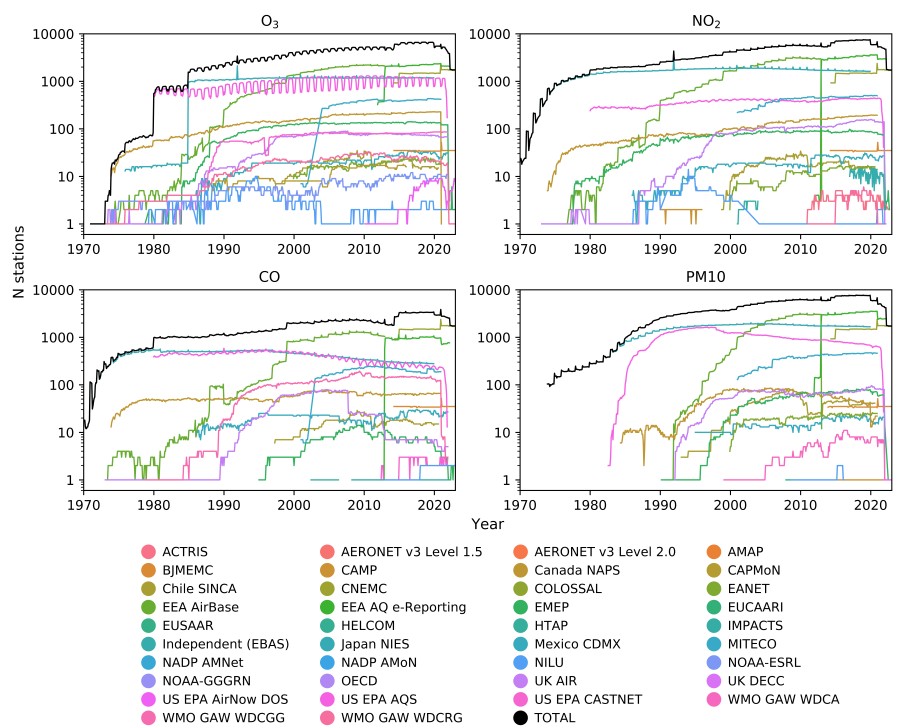

**Figure 8.** Evolution of the number of stations in GHOST, in each month across the time record (1970-2023), for 4 key components: $O_3$, $NO_2$, CO, and PM10. The differing number of stations per reporting network are represented by different coloured lines. The total number of stations across all networks is shown in black.



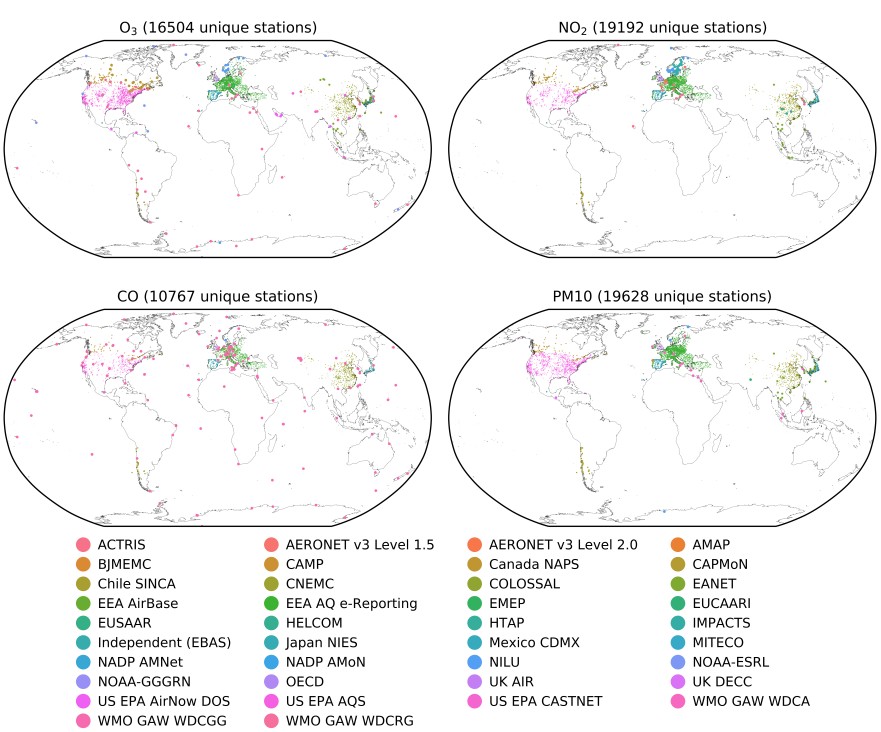

**Figure 9.** Spatial distribution of all unique stations in GHOST, across the time record (1970-2023), for 4 key components: $O_3$, $NO_2$, CO, and PM10. The stations are coloured by reporting network. The number of unique stations across the time record, per component, are given in the map titles.





## 5 Recommendations for data providers

The measurement of atmospheric components can often be costly, and require a huge amount of human labour, especially when low measurement uncertainty is required. We would like to thank all data providers for their work, which is of great benefit to the entire atmospheric composition community. The work done in creating GHOST however has highlighted several issues associated with the reporting of atmospheric composition data. In this section we will highlight some issues we identified

through this work, which we hope will be useful feedback for data providers.

In general, despite extensive efforts to gather as much available information from each reporting network as possible, there is simply a lack of detailed metadata associated with measurements. This lack of detail leads to many assumptions being made, and subsequently uncertainties being placed on measurements. In many cases, even basic metadata, such as the measurement altitude, sampling height, or even the longitude / latitude are not provided. Even when metadata is provided, the lack of explicit

detail can also lead to significant uncertainties. For example, providing a longitude / latitude with just a couple of decimal places can lead to the measurement position being erroneously located 10s of kilometres from the correct position. This was found to happen to even one of the most famous measurement stations, with its position being erroneously stated to be over the ocean.

The area where the reported metadata is most lacking is for that associated with measurement processes. In the majority of

1085 cases, the only measurement process information provided is a measurement methodology, and in some instances even that is not. Information such as the instrument name, sampling procedures, and limits of detection is very rarely provided, and more advanced information about measurement uncertainties, calibration procedures, etc. is almost never provided. Even when metadata is available, the lack of harmonisation across the reporting networks imposes a significant strain on the processing. For example, there are a number of methodologies which fundamentally measure concentrations of total PM through the scattering

of visible light, namely: nephelometry, light scattering photometry, and optical particle counting. Each of these methods operate in subtly distinct ways, and simply stating "light scattering" is not enough information to determine which exact method was used.

The conversion of measurement units was also made very challenging due to limited available information. In some cases the reported units were not provided with the data or metadata, and required rigorous investigation of network reports to find.

When converting from a mass density (e.g. $\mu$g m$^{-3}$) to a mole fraction (e.g. ppbv), or vice versa, the conversion requires the temperature and pressure associated with the air sampled. An additional complication to this is that many networks standardise measurements to a fixed temperature and pressure. The sample / network standard temperature and pressure is not commonly reported across the networks, and in some cases assumptions were needed to be made when converting units. Ideally, data providers would reference the applicable international measurement standards for their measurements, e.g. European standards.

The lack of metadata, for each of the cases outlined here, could probably be easily remedied by the data providers, as most of the information they most likely already have. A more deep rooted issue however is the reporting format used by networks to provide metadata. In the majority of cases, station metadata is provided in an external file, and is applicable for the entire time record. For stations which have measured for decades this can be problematic, as the type of air dominantly sampled at



a station can evolve over time, and should be reflected in the metadata, e.g. through station classes. Measurement techniques are also ever evolving, thus instrumentation is continuously being replaced or upgraded, which should also be reflected in the metadata.

One promising approach, which has been adopted by the EEA AQ e-Reporting network, is to associate all measurements with a sample ID. Each ID is associated with a specific collection of metadata, e.g. longitude, measurement method, instrument name. If one of the metadata values in this collection changes, e.g. a new instrument is installed, then the previous ID is no longer applicable, and a new ID is associated with measurements. Such an approach allows for the reporting of measurements from multiple instruments at one station. A potentially even cleaner approach would be to have a set of IDs for metadata associated with the station position, i.e. longitude, latitude, sampling height, and another set of IDs for metadata associated with measurement processes. This would ensure that a large number of metadata values are not needlessly duplicated between IDs, when just one value changes.

## 6 Conclusions

GHOST represents one of the biggest collection of harmonised measurements of atmospheric composition at the surface. In total, 7,275,148,646 measurements from 1970-2023, of 227 different components from 38 reporting networks, are compiled, parsed, and standardised. Components processed include gaseous species, total and speciated particulate matter, and aerosol optical properties. Data is made available in netCDF4 files, in 4 different temporal resolutions: hourly, hourly instantaneous, daily, and monthly.

The main goal of GHOST is to provide a dataset that can serve as a basis for the reproducibility of model evaluation efforts across the community. Exhaustive efforts have been made towards standardising almost every facet of provided information from the major public reporting networks, saved in 21 data variables, and 163 metadata variables. For this purpose, a fully parallelised workflow was created to enable the processing of such a large quantity of data. Through this process, a number of challenging issues are tackled, e.g. converting measurement units, shifting local time to UTC, handling measurement position changes. Extensive effort in particular is put towards the standardisation of measurement process information, and station classifications.

Rather than dropping any measurements which are labelled as potentially erroneous by the measurement provider, a range of standardised network QA flags are associated with each individual measurement. GHOST own QA is also performed and associated with measurements. For users who do not wish to worry about filtering data with the provided flags, measurements prefiltered by some default GHOST QA are also provided.

Measurements of all temporal resolutions are parsed in GHOST (e.g. 30 minutes, 6 hours), which are subsequently standardised by temporally averaging data to standard temporal resolutions (e.g. hourly). Data variables informing on the representativity of the temporal averaging are created, providing the percentage representativity of the native measurements that go into each temporal average. A variety of different reference times are associated with measurements: UTC, mean solar time, and local time.





Extra complementary information is also associated with measurements, such as metadata from various popular gridded datasets (e.g. land use), and temporal classifications per measurement (e.g. day / night). As the dataset spans more than 50 years, metadata is handled dynamically, and allowed to vary through the record, allowing changes in things such as the measurement

instrumentation, or measurement position, to be tracked.

We hope this work can be a spark for greater dialogue in the community regarding the reporting and standardisation of atmospheric composition data, and rather than being just a one off harmonisation effort, can be built upon and refined with the help of measurement experts from across the globe. We would warmly encourage any data providers who would wish to incorporate their data in GHOST to please contact us.

The GHOST dataset is made freely available via the following repository: https://doi.org/10.5281/zenodo.10637449 (Bowdalo, 2024).

## 7   Code availability

The code used to process GHOST is made available via GitLab: https://earth.bsc.es/gitlab/ac/GHOST.

GHOST processing software has been licenced with LGPLv3.

## 8   Data availability

The GHOST dataset is made freely available via the following repository: https://doi.org/10.5281/zenodo.10637449 (Bowdalo, 2024). The dataset has been licenced with CC BY 4.0. We would kindly ask any use of this dataset to cite both this publication and the dataset itself.

The dataset is 1.39 TB in total size (121 GB compressed), and includes data from all networks that we have the right to

redistribute, indicated in the data rights column of Table 1. The specific network data sources that GHOST draws from are listed in Table 1.

The data is separated out per network, per temporal resolution, per component, and is saved as netCDF4 files, per year and month. There is additionally one synthetic network entitled "GHOST-PUBLIC", which aggregates data across all networks. The dataset is compressed as .zip files per network. Beneath each network, collections of files per temporal resolution, per

component, are compressed as tar.xz files.

Each network .zip file can be decompressed via the following syntax:

*unzip [network].zip*

Component tar.xz files can be decompressed via the following syntax:

*tar -xf [component].tar.xz*



**Appendix A**





**Table A1.** Definitions of GHOST standard data variables. The variable name, data type, units, and a brief description are given. The "standard component units" refer to the standard units per component, as documented in Table A3.

| Variable | Data Type | Units | Description |
|---|---|---|---|
| time | uint32 | N hours / days / months from the start of the UTC month | Start time of measurement window, in UTC. |
| local_time | uint32 | minutes since 0001-01-01 00:00:00 | Start time of measurement window, in local time. |
| mean_solar_time | uint32 | minutes since 0001-01-01 00:00:00 | Start time of measurement window, in mean solar time. |
| *GHOSTcomponentname* | float32 | standard component units | Measured value of the component. |
| *GHOSTcomponentname_ prefiltered_defaultqa* | float32 | standard component units | Measured value of the component, prefiltered by default QA (defined in Table A10). |
| reported_uncertainty_per_ measurement | float32 | standard component units | Measurement uncertainty, as reported by the data provider. |
| derived_uncertainty_per_ measurement | float32 | standard component units | Derived measurement uncertainty, calculated as the quadratic addition of the measurement accuracy and precision metrics. The metrics used for calculation are network reported if available, else they from the instrument documentation. |
| flag | uint8 | unitless | List of standardised network QA flags, per measurement. |
| flag_simple | uint8 | unitless | List of simplified standardised network QA flags, per measurement. The template for the flags follows WaterML2.0 (Taylor et al., 2014). |
| qa | uint8 | unitless | List of GHOST QA flags, per measurement. |
| day_night_code | uint8 | unitless | Classification indicating if a measurement is made during the day (code 0), or night (code 1). |



| Variable | Data Type | Units | Description |
|---|---|---|---|
| weekday_weekend_code | uint8 | unitless | Classification indicating if a measurement is made on a weekday (code 0), or weekend (code 1). |
| season_code | uint8 | unitless | Classification indicating if a measurement is made during the spring (code 0), summer (code 1), autumn (code 2), or winter (code 3). |
| hourly_native_representativity_percent | uint8 | % | Percentage of an hourly UTC window represented by native resolution data. |
| daily_native_representativity_percent | uint8 | % | Percentage of a daily UTC window represented by native resolution data. |
| monthly_native_representativity_percent | uint8 | % | Percentage of a monthly UTC window represented by native resolution data. |
| annual_native_representativity_percent | uint8 | % | Percentage of an annual UTC window represented by native resolution data. |
| hourly_native_max_gap_percent | uint8 | % | Percentage maximum data gap in an hourly UTC window filled by native resolution data, relative to the total window length. |
| daily_native_max_gap_percent | uint8 | % | Percentage maximum data gap in a daily UTC window filled by native resolution data, relative to the total window length. |
| monthly_native_max_gap_percent | uint8 | % | Percentage maximum data gap in a monthly UTC window filled by native resolution data, relative to the total window length. |
| annual_native_max_gap_percent | uint8 | % | Percentage maximum data gap in an annual UTC window filled by native resolution data, relative to the total window length. |



**Table A2.** Definitions of GHOST standard metadata variables. The variable name, data type, units, and a brief description are given. The "standard component units" refer to the standard units per component, as documented in Table A3.

| Variable | Data Type | Units | Description |
|---|---|---|---|
| GHOST_version | str | unitless | Version number of GHOST. |
| **Network Provided Station Information** | | | |
| WIGOS_station_identifier | str | unitless | WIGOS station identifier (WSI). |
| station_reference | str | unitless | Reference ID for station. |
| station_timezone | str | unitless | Name of the local timezone that the station is located in. Calculated using the Python timezonefinder package (Michelfeit). |
| longitude | float64 | decimal degrees East | Geodetic longitude of the measuring instrument position, following a specific horizontal datum. |
| latitude | float64 | decimal degrees North | Geodetic latitude of the measuring instrument position, following a specific horizontal datum. |
| altitude | float32 | m | Altitude of the ground level at the station, relative to a specific vertical datum. |
| sampling_height | float32 | m | Height above the ground level of the measuring instrument sample inlet. |
| measurement_altitude | float32 | m | Altitude of the measuring instrument sample inlet, relative to a specific vertical datum. |
| ellipsoid | str | unitless | The ellipsoidal model of the earth used as a basis for 2D and 3D geographic coordinate systems. |
| horizontal_datum | str | unitless | Name of the horizontal datum used in defining geodetic latitudes and longitudes on the Earth's surface. |
| vertical_datum | str | unitless | Name of the vertical datum used to define vertical elevation on the Earth. |
| projection | str | unitless | Name of the projected coordinate system of the original provided station position x, y coordinates. |
| distance_to_building | float32 | m | Distance to the nearest building, of the measuring instrument sample inlet. |
| distance_to_kerb | float32 | m | Distance to the street kerb, of the measuring instrument sample inlet. |
| distance_to_junction | float32 | m | Distance to the nearest road junction, of the measuring instrument sample inlet. |



| Variable | Data Type | Units | Description |
|---|---|---|---|
| distance_to_source | float32 | km | Distance to the main emission source, of the measuring instrument sample inlet. |
| street_width | float32 | m | Width of the street, where the measuring instrument is located. |
| street_type | str | unitless | Type of the street, where the measuring instrument is located. |
| daytime_traffic_speed | float32 | km hr$^{-1}$ | Average daytime speed of the passing traffic, where the measuring instrument is located. |
| daily_passing_vehicles | float32 | unitless | Daily average number of vehicles passing, where the measuring instrument is located. |
| data_level | str | unitless | Network provided data level of reported measurements. |
| climatology | str | unitless | Name of the climatology of which the observations pertain to. |
| station_name | str | unitless | Name of the measuring station. |
| city | str | unitless | Name of the city the station is located in. Calculated using the reverse_geocoder module (Thampi). |
| country | str | unitless | Name of the country the station is located in. Calculated using the reverse_geocoder module (Thampi). |
| administrative_country_division_1 | str | unitless | Name of the largest country administrative division in which the station lies. Calculated using the reverse_geocoder module (Thampi). |
| administrative_country_division_2 | str | unitless | Name of the second largest country administrative division in which the station lies. Calculated using the reverse_geocoder module (Thampi). |
| population | float32 | unitless | Population count of the nearest urban settlement. |
| representative_radius | float32 | km | Radius of representativity of the air dominantly measured at a station. |
| network | str | unitless | The reporting network name. |
| associated_networks | str | unitless | Names of associated networks that the station data is reported to, and the station references in said networks. Multiple networks are separated by ";". |
| **Standardised Network Provided Classifications** | | | |
| area_classification | str | unitless | Classification of the type of area a station is situated in. |
| station_classification | str | unitless | Classification of the type of air dominantly measured by a station. |



| Variable | Data Type | Units | Description |
|---|---|---|---|
| main_emission_source | str | unitless | Main emission source influencing air measured at a station. |
| land_use | str | unitless | Dominant land use in the area of a station. |
| terrain | str | unitless | Dominant terrain in the area of a station. |
| measurement_scale | str | unitless | Denotation of the geographic scope of the air measured at a station. |
| **Gridded Classifications** | | | |
| ESDAC_Iwahashi_landform_classification | str | unitless | Landform classification derived from slope gradient, surface texture and local convexity. |
| ESDAC_modal_Iwahashi_landform_classification_5km | str | unitless | Modal ESDAC Iwahashi landform classification, in a radius of 5km around the station. |
| ESDAC_modal_Iwahashi_landform_classification_25km | str | unitless | Modal ESDAC Iwahashi landform classification, in a radius of 25km around the station. |
| ESDAC_Meybeck_landform_classification | str | unitless | Landform classification derived from surface roughness. |
| ESDAC_modal_Meybeck_landform_classification_5km | str | unitless | Modal ESDAC Meybeck landform classification, in a radius of 5km around the station. |
| ESDAC_modal_Meybeck_landform_classification_25km | str | unitless | Modal ESDAC Meybeck landform classification, in a radius of 25km around the station. |
| GHSL_settlement_model_classification | str | unitless | Settlement type classification derived from population counts, population density and built-up area density. |
| GHSL_modal_settlement_model_classification_5km | str | unitless | Modal GHSL settlement model classification, in a radius of 5km around the station. |
| GHSL_modal_settlement_model_classification_25km | str | unitless | Modal GHSL settlement model classification, in a radius of 25km around the station. |
| Joly-Peuch_classification_code | float32 | unitless | Objective classification of the urban signature of a measured component at a station (most rural = 1, most urban = 10). Only available for some components: $O_3$, $NO_2$, $SO_2$, CO, PM10, PM2.5. |
| Koppen-Geiger_classification | str | unitless | Classification of the global climate types. |
| Koppen-Geiger_modal_classification_5km | str | unitless | Modal Koppen-Geiger classification, in a radius of 5km around the station. |
| Koppen-Geiger_modal_classification_25km | str | unitless | Modal Koppen-Geiger classification, in a radius of 25km around the station. |

*table continued on next page*





| Variable | Data Type | Units | Description |
|---|---|---|---|
| MODIS_MCD12C1_v6_IGBP_ land_use | str | unitless | Land use classification, derived from MODIS satellite imaging, using IGBP class definitions. |
| MODIS_MCD12C1_v6_modal_ IGBP_land_use_5km | str | unitless | Modal MODIS IGBP land use, in a radius of 5km around the station. |
| MODIS_MCD12C1_v6_modal_ IGBP_land_use_25km | str | unitless | Modal MODIS IGBP land use, in a radius of 25km around the station. |
| MODIS_MCD12C1_v6_UMD_ land_use | str | unitless | Land use classification, derived from MODIS satellite imaging, using UMD class definitions. |
| MODIS_MCD12C1_v6_modal_ UMD_land_use_5km | str | unitless | Modal MODIS UMD land use, in a radius of 5km around the station. |
| MODIS_MCD12C1_v6_modal_ UMD_land_use_25km | str | unitless | Modal MODIS UMD land use, in a radius of 25km around the station. |
| MODIS_MCD12C1_v6_LAI | str | unitless | Leaf Area Index (LAI) classification, derived from MODIS satellite imaging. |
| MODIS_MCD12C1_v6_modal_ LAI_5km | str | unitless | Modal MODIS LAI, in a radius of 5km around the station. |
| MODIS_MCD12C1_v6_modal_ LAI_25km | str | unitless | Modal MODIS LAI, in a radius of 25km around the station. |
| WMO_region | str | unitless | Classification of the global regions. |
| WWF_TEOW_terrestrial_ ecoregion | str | unitless | Classification of the global terrestrial ecoregions. |
| WWF_TEOW_biogeographical_ realm | str | unitless | Classification of the global biogeographical realms. |
| WWF_TEOW_biome | str | unitless | Classification of the global biomes. |
| UMBC_anthrome_classification | str | unitless | Anthropogenic land use classification. |
| UMBC_modal_anthrome_ classification_5km | str | unitless | Modal UMBC anthrome classification, in a radius of 5km around the station. |
| UMBC_modal_anthrome_ classification_25km | str | unitless | Modal UMBC anthrome classification, in a radius of 25km around the station. |
| **Gridded Products** | | | |
| EDGAR_v4.3.2_annual_average_ BC_emissions | float32 | kg m$^{-2}$ s$^{-1}$ | Annual average black carbon emissions. |

*table continued on next page*



| Variable | Data Type | Units | Description |
|---|---|---|---|
| EDGAR_v4.3.2_annual_average_ CO_emissions | float32 | kg m$^{-2}$ s$^{-1}$ | Annual average CO emissions. |
| EDGAR_v4.3.2_annual_average_ NH3_emissions | float32 | kg m$^{-2}$ s$^{-1}$ | Annual average NH$_3$ emissions. |
| EDGAR_v4.3.2_annual_average_ NMVOC_emissions | float32 | kg m$^{-2}$ s$^{-1}$ | Annual average NMVOC emissions. |
| EDGAR_v4.3.2_annual_average_ NOx_emissions | float32 | kg m$^{-2}$ s$^{-1}$ | Annual average NO$_x$ emissions. |
| EDGAR_v4.3.2_annual_average_ OC_emissions | float32 | kg m$^{-2}$ s$^{-1}$ | Annual average organic carbon emissions. |
| EDGAR_v4.3.2_annual_average_ PM10_emissions | float32 | kg m$^{-2}$ s$^{-1}$ | Annual average PM10 emissions. |
| EDGAR_v4.3.2_annual_average_ biogenic_PM2.5_emissions | float32 | kg m$^{-2}$ s$^{-1}$ | Annual average biogenic PM2.5 emissions. |
| EDGAR_v4.3.2_annual_average_ fossilfuel_PM2.5_emissions | float32 | kg m$^{-2}$ s$^{-1}$ | Annual average fossil fuel PM2.5 emissions. |
| EDGAR_v4.3.2_annual_average_ SO2_emissions | float32 | kg m$^{-2}$ s$^{-1}$ | Annual average SO$_2$ emissions. |
| ASTER_v3_altitude | float32 | m | Digital elevation model altitude, derived from TERRA satellite imaging. |
| ETOPO1_altitude | float32 | m | Digital elevation model altitude, derived from topography, bathymetry, and shoreline data. |
| ETOPO1_max_altitude_difference_ 5km | float32 | m | Altitude difference between the ETOPO1 altitude, and the minimum ETOPO1 altitude, in a radius of 5km around the station. |
| GHSL_built_up_area_density | float32 | % | Built up area density, as a percentage, derived from Landsat satellite imaging. |
| GHSL_average_built_up_area_ density_5km | float32 | % | Average GHSL built up area density, in a radius of 5km around the station. |
| GHSL_average_built_up_area_ density_25km | float32 | % | Average GHSL built up area density, in a radius of 25km around the station. |
| GHSL_max_built_up_area_ density_5km | float32 | % | Maximum GHSL built up area density, in a radius of 5km around the station. |



| Variable | Data Type | Units | Description |
|---|---|---|---|
| GHSL_max_built_up_area_density_25km | float32 | % | Maximum GHSL built up area density, in a radius of 25km around the station. |
| GHSL_population_density | float32 | people km$^{-2}$ | Population density, based on GPW population counts. |
| GHSL_average_population_density_5km | float32 | people km$^{-2}$ | Average GHSL population density, in a radius of 5km around the station. |
| GHSL_average_population_density_25km | float32 | people km$^{-2}$ | Average GHSL population density, in a radius of 25km around the station. |
| GHSL_max_population_density_5km | float32 | people km$^{-2}$ | Maximum GHSL population density, in a radius of 5km around the station. |
| GHSL_max_population_density_25km | float32 | people km$^{-2}$ | Maximum GHSL population density, in a radius of 25km around the station. |
| GPW_population_density | float32 | people km$^{-2}$ | Population density, derived from global census data. |
| GPW_average_population_density_5km | float32 | people km$^{-2}$ | Average GPW population density, in a radius of 5km around the station. |
| GPW_average_population_density_25km | float32 | people km$^{-2}$ | Average GPW population density, in a radius of 25km around the station. |
| GPW_max_population_density_5km | float32 | people km$^{-2}$ | Maximum GPW population density, in a radius of 5km around the station. |
| GPW_max_population_density_25km | float32 | people km$^{-2}$ | Maximum GPW population density, in a radius of 25km around the station. |
| NOAA-DMSP-OLS_v4_nighttime_stable_lights | float32 | unitless | Nighttime stable lights, derived from DMSP-OLS satellite imaging. The values are essentially a brightness index, ranging from 0 to 63. |
| NOAA-DMSP-OLS_v4_average_nighttime_stable_lights_5km | float32 | unitless | Average NOAA DMSP-OLS nighttime stable lights, in a radius of 5km around the station. |
| NOAA-DMSP-OLS_v4_average_nighttime_stable_lights_25km | float32 | unitless | Average NOAA DMSP-OLS nighttime stable lights, in a radius of 25km around the station. |
| NOAA-DMSP-OLS_v4_max_nighttime_stable_lights_5km | float32 | unitless | Maximum NOAA DMSP-OLS nighttime stable lights, in a radius of 5km around the station. |
| NOAA-DMSP-OLS_v4_max_nighttime_stable_lights_25km | float32 | unitless | Maximum NOAA DMSP-OLS nighttime stable lights, in a radius of 25km around the station. |
| OMI_level3_column_annual_average_NO2 | float32 | molecules cm$^{-2}$ | Column annual average $NO_2$, calculated from measurements from the OMI instrument on the AURA satellite. |

*table continued on next page*





| Variable | Data Type | Units | Description |
|---|---|---|---|
| OMI_level3_column_cloud_ screened_annual_average_NO2 | float32 | molecules cm$^{-2}$ | OMI column annual average NO$_2$, screened for cloud fraction less than 30 percent. |
| OMI_level3_tropospheric_column_ annual_average_NO2 | float32 | molecules cm$^{-2}$ | Tropospheric OMI column annual average NO$_2$. |
| OMI_level3_tropospheric_column_ cloud_screened_annual_average_ NO2 | float32 | molecules cm$^{-2}$ | Tropospheric OMI column annual average NO$_2$, screened for cloud fraction less than 30 percent. |
| GSFC_coastline_proximity | float32 | km | Proximity to the coastline. Negative distances represent locations over land, while positive distances represent locations over the ocean. |
| **Measurement Information** | | | |
| primary_sampling_type | str | unitless | Type of process used to sample air, by the primary sampling instrument. |
| primary_sampling_instrument_ name | str | unitless | Primary sampling instrument name. |
| primary_sampling_instrument_ reported_flow_rate | str | l min$^{-1}$ | Volume of fluid sampled per unit time, by the primary sampling instrument, as reported by the data provider. |
| primary_sampling_instrument_ documented_flow_rate | str | l min$^{-1}$ | Volume of fluid sampled per unit time, by the primary sampling instrument, as stated in the instrument documentation. |
| primary_sampling_process_details | str | unitless | Miscellaneous details about assumptions made in the standardisation of the primary sampling type / instrument. |
| primary_sampling_instrument_ manual_name | str | unitless | Name of the primary sampling instrument manual. |
| primary_sampling_further_details | str | unitless | Further details associated with the primary sampling type / instrument. |
| sample_preparation_types | str | unitless | Types of processes used to prepare sample for subsequent measurement. Multiple types are separated by ";" |
| sample_preparation_techniques | str | unitless | Specific techniques of utilised preparation types. Multiple techniques are separated by ";". |
| sample_preparation_process_details | str | unitless | Miscellaneous details about assumptions made in the standardisation of the sample preparation types / techniques. |

*table continued on next page*





| Variable | Data Type | Units | Description |
|---|---|---|---|
| sample_preparation_further_details | str | unitless | Further associated details associated with sample preparation types / techniques. |
| measurement_methodology | str | unitless | Methodology used for measuring component. |
| measuring_instrument_name | str | unitless | Measuring instrument name. |
| measuring_instrument_sampling_type | str | unitless | Type of process used to sample air, by the measuring instrument. |
| measuring_instrument_reported_flow_rate | str | l min$^{-1}$ | Volume of fluid sampled per unit time, by the measuring instrument, as reported by the data provider. |
| measuring_instrument_documented_flow_rate | str | l min$^{-1}$ | Volume of fluid sampled per unit time, by the measuring instrument, as stated in the instrument documentation. |
| measuring_instrument_process_details | str | unitless | Miscellaneous details about assumptions made in the standardisation of the measurement method / instrument. |
| measuring_instrument_manual_name | str | unitless | Name of the measuring instrument manual. |
| measuring_instrument_further_details | str | unitless | Further details associated with the measurement method / instrument. |
| measuring_instrument_reported_units | str | unitless | Units that the measured component are natively reported in. |
| measuring_instrument_reported_lower_limit_of_detection | float32 | standard component units | Lower limit of detection of the measuring instrument, as reported by the data provider. |
| measuring_instrument_documented_lower_limit_of_detection | float32 | standard component units | Lower limit of detection of the measuring instrument, as stated in the instrument documentation. |
| measuring_instrument_reported_upper_limit_of_detection | float32 | standard component units | Upper limit of detection of the measuring instrument, as reported by the data provider. |
| measuring_instrument_documented_upper_limit_of_detection | float32 | standard component units | Upper limit of detection of the measuring instrument, as stated in the instrument documentation. |

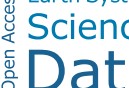

| Variable | Data Type | Units | Description |
|---|---|---|---|
| measuring_instrument_reported_ uncertainty | str | standard component units | Measurement uncertainty, as reported by the data provider. |
| measuring_instrument_ documented_uncertainty | str | standard component units | Measurement uncertainty, as stated in the instrument documentation. |
| measuring_instrument_reported_ accuracy | str | standard component units | Difference between the measurement and the actual value of the part that is measured, as reported by the data provider. |
| measuring_instrument_ documented_accuracy | str | standard component units | Difference between the measurement and the actual value of the part that is measured, as stated in the instrument documentation. |
| measuring_instrument_reported_ precision | str | standard component units | Measure of the variation seen when the same part is measured repeatedly with the same instrument, as reported by the data provider. |
| measuring_instrument_ documented_precision | str | standard component units | Measure of the variation seen when the same part is measured repeatedly with the same instrument, as stated in the instrument documentation. |
| measuring_instrument_reported_ zero_drift | str | standard component units | Measurement drift across the full scale caused by slippage, or due to undue warming up of the electronic circuits, as reported by the data provider. |
| measuring_instrument_ documented_zero_drift | str | standard component units | Measurement drift across the full scale caused by slippage, or due to undue warming up of the electronic circuits, as stated in the instrument documentation. |
| measuring_instrument_reported_ span_drift | str | standard component units | Measurement drift which proportionally increases along the upward scale, as reported by the data provider. |
| measuring_instrument_ documented_span_drift | str | standard component units | Measurement drift which proportionally increases along the upward scale, as stated in the instrument documentation. |





| Variable | Data Type | Units | Description |
|----------|-----------|-------|-------------|
| measuring_instrument_reported_zonal_drift | str | standard component units | Measurement drift which occurs only over a portion of the full scale, as reported by the data provider. |
| measuring_instrument_documented_zonal_drift | str | standard component units | Measurement drift which occurs only over a portion of the full scale, as stated in the instrument documentation. |
| measuring_instrument_reported_measurement_resolution | float32 | standard component units | Smallest level of change of a measured quantity that the instrument can detect, as reported by the data provider. |
| measuring_instrument_documented_measurement_resolution | float32 | standard component units | Smallest level of change of a measured quantity that the instrument can detect, as stated in the instrument documentation. |
| measuring_instrument_reported_absorption_cross_section | str | $cm^2$ | Assumed molecule cross-section for the component being measured (for optical measurement methods), as reported by the data provider. |
| measuring_instrument_documented_absorption_cross_section | str | $cm^2$ | Assumed molecule cross-section for the component being measured (for optical measurement methods), as stated in the instrument documentation. |
| measuring_instrument_inlet_information | str | unitless | Description of the sampling inlet of the measuring instrument. |
| measuring_instrument_calibration_scale | str | unitless | Name of the scale used for the calibration of the measuring instrument. |
| retrieval_algorithm | str | unitless | Name of the retrieval algorithm associated with measurement (for remote sampling). |
| network_provided_volume_standard_temperature | float64 | K | Temperature associated with the volume of the sampled gas. |
| network_provided_volume_standard_pressure | float64 | hPa | Pressure associated with the volume of the sampled gas. |
| **Contact Information** | | | |
| principal_investigator_name | str | unitless | Full name of the principal scientific investigator. |
| principal_investigator_institution | str | unitless | Institution of the principal scientific investigator. |
| principal_investigator_email_address | str | unitless | Email address of the principal scientific investigator. |



| Variable | Data Type | Units | Description |
|---|---|---|---|
| contact_name | str | unitless | Full name of the principal data contact. |
| contact_institution | str | unitless | Institution of the principal data contact. |
| contact_email_address | str | unitless | Email address of the principal data contact. |
| **Further Detail** | | | |
| network_sampling_details | str | unitless | Extra details about the sampling methods employed, from the data provider. |
| network_uncertainty_details | str | unitless | Extra details about the measurement uncertainties, from the data provider. |
| network_maintenance_details | str | unitless | Extra details about the operational maintenance at the station, from the data provider. |
| network_qa_details | str | unitless | Extra details about network quality assurance, from the data provider. |
| network_miscellaneous_details | str | unitless | Extra miscellaneous details from the data provider. |
| data_licence | str | unitless | Data licence of the ingested network data. |
| **Warnings** | | | |
| process_warnings | str | unitless | Process warnings accumulated through the GHOST pipeline. |



**Table A3.** GHOST standard component information, grouped per matrix. For each component, the chemical formula, long component name, standard units, minimum permitted measurement resolution, extreme lower limit, extreme upper limit, and extreme upper monthly median are given.

| GHOST Component Name | Chemical Formula | Long Component Name | Standard Units | Minimum Permitted Measurement Resolution | Extreme Lower Limit | Extreme Upper Limit | Extreme Upper Monthly Median |
|---|---|---|---|---|---|---|---|
| **gas** | | | | | | | |
| sconco3 | $O_3$ | ozone | nmol mol$^{-1}$ | 1.0 | 0.0 | 400.0 | 120.0 |
| sconcno | NO | nitrogen monoxide | nmol mol$^{-1}$ | 1.0 | 0.0 | 1200.0 | 250.0 |
| sconcno2 | $NO_2$ | nitrogen dioxide | nmol mol$^{-1}$ | 1.0 | 0.0 | 600.0 | 200.0 |
| sconcso2 | $SO_2$ | sulphur dioxide | nmol mol$^{-1}$ | 2.0 | 0.0 | 3000.0 | 750.0 |
| sconcco | CO | carbon monoxide | nmol mol$^{-1}$ | 20.0 | 0.0 | 30000.0 | 7500.0 |
| sconcch4 | $CH_4$ | methane | nmol mol$^{-1}$ | 20.0 | 0.0 | 50000.0 | 5000.0 |
| sconcc2h4 | $C_2H_4$ | ethene | pmol mol$^{-1}$ | 100.0 | 0.0 | 500000.0 | 50000.0 |
| sconcc2h6 | $C_2H_6$ | ethane | pmol mol$^{-1}$ | 100.0 | 0.0 | 500000.0 | 50000.0 |
| sconcc3h6 | $C_3H_6$ | propene | pmol mol$^{-1}$ | 100.0 | 0.0 | 500000.0 | 50000.0 |
| sconcc3h8 | $C_3H_8$ | propane | pmol mol$^{-1}$ | 100.0 | 0.0 | 500000.0 | 50000.0 |
| sconcisop | $C_5H_8$ | isoprene | pmol mol$^{-1}$ | 100.0 | 0.0 | 500000.0 | 50000.0 |
| sconcc6h6 | $C_6H_6$ | benzene | pmol mol$^{-1}$ | 100.0 | 0.0 | 500000.0 | 50000.0 |
| sconcc7h8 | $C_7H_8$ | toluene | pmol mol$^{-1}$ | 100.0 | 0.0 | 500000.0 | 50000.0 |
| sconcc10h16 | $C_{10}H_{16}$ | monoterpenes | pmol mol$^{-1}$ | 100.0 | 0.0 | 500000.0 | 50000.0 |
| sconcnmvoc | ——- | total non-methane volatile organic compounds | nmol mol$^{-1}$ | 20.0 | 0.0 | 20000.0 | 5000.0 |
| sconcvoc | ——- | total volatile organic compounds | nmol mol$^{-1}$ | 50.0 | 0.0 | 70000.0 | 10000.0 |
| sconnmhc | ——- | total non-methane hydrocarbons | nmol mol$^{-1}$ | 20.0 | 0.0 | 20000.0 | 5000.0 |
| sconchc | ——- | total hydrocarbons | nmol mol$^{-1}$ | 50.0 | 0.0 | 70000.0 | 10000.0 |
| sconcnh3 | $NH_3$ | ammonia | nmol mol$^{-1}$ | 1.0 | 0.0 | 1000.0 | 100.0 |



| GHOST Component Name | Chemical Formula | Long Component Name | Standard Units | Minimum Permitted Measurement Resolution | Extreme Lower Limit | Extreme Upper Limit | Extreme Upper Monthly Median |
|---|---|---|---|---|---|---|---|
| sconchno3 | $HNO_3$ | nitric acid | nmol mol$^{-1}$ | 0.1 | 0.0 | 25.0 | 5.0 |
| sconcpan | $C_2H_3NO_5$ | peroxyacetyl nitrate | nmol mol$^{-1}$ | 0.1 | 0.0 | 25.0 | 5.0 |
| sconchcho | $CH_2O$ | formaldehyde | nmol mol$^{-1}$ | 0.2 | 0.0 | 100.0 | 25.0 |
| sconchcl | HCl | hydrochloric acid | nmol mol$^{-1}$ | 0.1 | 0.0 | 25.0 | 5.0 |
| sconchf | HF | hydrofluoric acid | nmol mol$^{-1}$ | 1.0 | 0.0 | 1000.0 | 200.0 |
| sconch2s | $H_2S$ | hydrogen sulphide | nmol mol$^{-1}$ | 1.0 | 0.0 | 1000.0 | 200.0 |
| **pm** | | | | | | | |
| sconcal | Al | total particulate aluminium | ng m$^{-3}$ | 20.0 | 0.0 | 50000.0 | 5000.0 |
| sconcas | As | total particulate arsenic | ng m$^{-3}$ | 1.0 | 0.0 | 1000.0 | 200.0 |
| sconcbc | C | total particulate black carbon | $\mu$g m$^{-3}$ | 10.0 | 0.0 | 25000.0 | 2500.0 |
| sconcc | C | total particulate carbon | $\mu$g m$^{-3}$ | 20.0 | 0.0 | 50000.0 | 5000.0 |
| sconcca | $Ca^{2+}$ | total particulate calcium | $\mu$g m$^{-3}$ | 0.2 | 0.0 | 100.0 | 20.0 |
| sconccd | Cd | total particulate cadmium | ng m$^{-3}$ | 0.2 | 0.0 | 500.0 | 75.0 |
| sconccl | $Cl^-$ | total particulate chloride | $\mu$g m$^{-3}$ | 0.2 | 0.0 | 150.0 | 30.0 |
| sconccobalt | Co | total particulate cobalt | ng m$^{-3}$ | 0.1 | 0.0 | 50.0 | 5.0 |
| sconccr | Cr | total particulate chromium | ng m$^{-3}$ | 1.0 | 0.0 | 500.0 | 100.0 |
| sconccu | Cu | total particulate copper | ng m$^{-3}$ | 1.0 | 0.0 | 750.0 | 150.0 |
| sconcec | C | total particulate elemental carbon | $\mu$g m$^{-3}$ | 10.0 | 0.0 | 25000.0 | 2500.0 |





| GHOST Component Name | Chemical Formula | Long Component Name | Standard Units | Minimum Permitted Measurement Resolution | Extreme Lower Limit | Extreme Upper Limit | Extreme Upper Monthly Median |
|---|---|---|---|---|---|---|---|
| sconcfe | Fe | total particulate iron | ng m$^{-3}$ | 20.0 | 0.0 | 50000.0 | 5000.0 |
| sconchg | Hg | total particulate mercury | pg m$^{-3}$ | 10.0 | 0.0 | 30000.0 | 3000.0 |
| sconck | K$^+$ | total particulate potassium | $\mu$g m$^{-3}$ | 0.2 | 0.0 | 50.0 | 10.0 |
| sconcmg | Mg$^{2+}$ | total particulate magnesium | $\mu$g m$^{-3}$ | 0.2 | 0.0 | 50.0 | 10.0 |
| sconcmn | Mn | total particulate manganese | ng m$^{-3}$ | 2.0 | 0.0 | 5000.0 | 500.0 |
| sconcmsa | CH$_4$O$_3$S | total particulate methanesulfonic acid | $\mu$g m$^{-3}$ | 0.2 | 0.0 | 75.0 | 25.0 |
| sconcna | Na$^+$ | total particulate sodium | $\mu$g m$^{-3}$ | 0.2 | 0.0 | 150.0 | 30.0 |
| sconcnh4 | NH$_4^+$ | total particulate ammonium | $\mu$g m$^{-3}$ | 0.2 | 0.0 | 150.0 | 30.0 |
| sconcnh4no3 | NH$_4$NO$_3$ | total particulate ammonium nitrate | $\mu$g m$^{-3}$ | 0.2 | 0.0 | 150.0 | 30.0 |
| sconcni | Ni | total particulate nickel | ng m$^{-3}$ | 5.0 | 0.0 | 10000.0 | 1000.0 |
| sconcno3 | NO$_3^-$ | total particulate nitrate | $\mu$g m$^{-3}$ | 0.2 | 0.0 | 250.0 | 75.0 |
| sconcoc | C | total particulate organic carbon | $\mu$g m$^{-3}$ | 10.0 | 0.0 | 25000.0 | 2500.0 |
| sconcpb | Pb | total particulate lead | ng m$^{-3}$ | 50.0 | 0.0 | 60000.0 | 15000.0 |
| sconcse | Se | total particulate selenium | ng m$^{-3}$ | 0.2 | 0.0 | 150.0 | 30.0 |
| sconcso4 | SO$_4^{2-}$ | total particulate sulphate | $\mu$g m$^{-3}$ | 0.2 | 0.0 | 150.0 | 30.0 |
| sconcso4nss | SO$_4^{2-}$ | total particulate sulphate: non-sea salt | $\mu$g m$^{-3}$ | 0.2 | 0.0 | 150.0 | 30.0 |

*table continued on next page*



| GHOST Component Name | Chemical Formula | Long Component Name | Standard Units | Minimum Permitted Measurement Resolution | Extreme Lower Limit | Extreme Upper Limit | Extreme Upper Monthly Median |
|---|---|---|---|---|---|---|---|
| sconcso4ss | $SO_4^{2-}$ | total particulate sulphate: sea salt | $\mu g\ m^{-3}$ | 0.2 | 0.0 | 150.0 | 30.0 |
| sconcv | V | total particulate vanadium | $ng\ m^{-3}$ | 0.2 | 0.0 | 100.0 | 20.0 |
| sconczn | Zn | total particulate zinc | $ng\ m^{-3}$ | 20.0 | 0.0 | 30000.0 | 5000.0 |
| **pm10** | | | | | | | |
| pm10 | ——- | total PM10 | $\mu g\ m^{-3}$ | 20.0 | 0.0 | 50000.0 | 5000.0 |
| pm10al | Al | PM10 aluminium | $ng\ m^{-3}$ | 20.0 | 0.0 | 50000.0 | 5000.0 |
| pm10as | As | PM10 arsenic | $ng\ m^{-3}$ | 1.0 | 0.0 | 1000.0 | 200.0 |
| pm10bc | C | PM10 black carbon | $\mu g\ m^{-3}$ | 10.0 | 0.0 | 25000.0 | 2500.0 |
| pm10c | C | PM10 carbon | $\mu g\ m^{-3}$ | 20.0 | 0.0 | 50000.0 | 5000.0 |
| pm10ca | $Ca^{2+}$ | PM10 calcium | $\mu g\ m^{-3}$ | 0.2 | 0.0 | 100.0 | 20.0 |
| pm10cd | Cd | PM10 cadmium | $ng\ m^{-3}$ | 0.2 | 0.0 | 500.0 | 75.0 |
| pm10cl | $Cl^-$ | PM10 chloride | $\mu g\ m^{-3}$ | 0.2 | 0.0 | 150.0 | 30.0 |
| pm10cobalt | Co | PM10 cobalt | $ng\ m^{-3}$ | 0.1 | 0.0 | 50.0 | 5.0 |
| pm10cr | Cr | PM10 chromium | $ng\ m^{-3}$ | 1.0 | 0.0 | 500.0 | 100.0 |
| pm10cu | Cu | PM10 copper | $ng\ m^{-3}$ | 1.0 | 0.0 | 750.0 | 150.0 |
| pm10ec | C | PM10 elemental carbon | $\mu g\ m^{-3}$ | 10.0 | 0.0 | 25000.0 | 2500.0 |
| pm10fe | Fe | PM10 iron | $ng\ m^{-3}$ | 20.0 | 0.0 | 50000.0 | 5000.0 |
| pm10hg | Hg | PM10 mercury | $pg\ m^{-3}$ | 10.0 | 0.0 | 30000.0 | 3000.0 |
| pm10k | $K^+$ | PM10 potassium | $\mu g\ m^{-3}$ | 0.2 | 0.0 | 50.0 | 10.0 |
| pm10mg | $Mg^{2+}$ | PM10 magnesium | $\mu g\ m^{-3}$ | 0.2 | 0.0 | 50.0 | 10.0 |
| pm10mn | Mn | PM10 manganese | $ng\ m^{-3}$ | 2.0 | 0.0 | 5000.0 | 500.0 |
| pm10msa | $CH_4O_3S$ | PM10 methanesulfonic acid | $\mu g\ m^{-3}$ | 0.2 | 0.0 | 75.0 | 25.0 |
| pm10na | $Na^+$ | PM10 sodium | $\mu g\ m^{-3}$ | 0.2 | 0.0 | 150.0 | 30.0 |
| pm10nh4 | $NH_4^+$ | PM10 ammonium | $\mu g\ m^{-3}$ | 0.2 | 0.0 | 150.0 | 30.0 |





| GHOST Component Name | Chemical Formula | Long Component Name | Standard Units | Minimum Permitted Measurement Resolution | Extreme Lower Limit | Extreme Upper Limit | Extreme Upper Monthly Median |
|---|---|---|---|---|---|---|---|
| pm10nh4no3 | $NH_4NO_3$ | PM10 ammonium nitrate | $\mu g\ m^{-3}$ | 0.2 | 0.0 | 150.0 | 30.0 |
| pm10ni | Ni | PM10 nickel | $ng\ m^{-3}$ | 5.0 | 0.0 | 10000.0 | 1000.0 |
| pm10no3 | $NO_3^-$ | PM10 nitrate | $\mu g\ m^{-3}$ | 0.2 | 0.0 | 250.0 | 75.0 |
| pm10oc | C | PM10 organic carbon | $\mu g\ m^{-3}$ | 10.0 | 0.0 | 25000.0 | 2500.0 |
| pm10pb | Pb | PM10 lead | $ng\ m^{-3}$ | 50.0 | 0.0 | 60000.0 | 15000.0 |
| pm10se | Se | PM10 selenium | $ng\ m^{-3}$ | 0.2 | 0.0 | 150.0 | 30.0 |
| pm10so4 | $SO_4^{2-}$ | PM10 sulphate | $\mu g\ m^{-3}$ | 0.2 | 0.0 | 150.0 | 30.0 |
| pm10so4nss | $SO_4^{2-}$ | PM10 sulphate: non-sea salt | $\mu g\ m^{-3}$ | 0.2 | 0.0 | 150.0 | 30.0 |
| pm10so4ss | $SO_4^{2-}$ | PM10 sulphate: sea salt | $\mu g\ m^{-3}$ | 0.2 | 0.0 | 150.0 | 30.0 |
| pm10v | V | PM10 vanadium | $ng\ m^{-3}$ | 0.2 | 0.0 | 100.0 | 20.0 |
| pm10zn | Zn | PM10 zinc | $ng\ m^{-3}$ | 20.0 | 0.0 | 30000.0 | 5000.0 |
| **pm2.5** | | | | | | | |
| pm2p5 | ——- | total PM2.5 | $\mu g\ m^{-3}$ | 20.0 | 0.0 | 50000.0 | 5000.0 |
| pm2p5al | Al | PM2.5 aluminium | $ng\ m^{-3}$ | 20.0 | 0.0 | 50000.0 | 5000.0 |
| pm2p5as | As | PM2.5 arsenic | $ng\ m^{-3}$ | 1.0 | 0.0 | 1000.0 | 200.0 |
| pm2p5bc | C | PM2.5 black carbon | $\mu g\ m^{-3}$ | 10.0 | 0.0 | 25000.0 | 2500.0 |
| pm2p5c | C | PM2.5 carbon | $\mu g\ m^{-3}$ | 20.0 | 0.0 | 50000.0 | 5000.0 |
| pm2p5ca | $Ca^{2+}$ | PM2.5 calcium | $\mu g\ m^{-3}$ | 0.2 | 0.0 | 100.0 | 20.0 |
| pm2p5cd | Cd | PM2.5 cadmium | $ng\ m^{-3}$ | 0.2 | 0.0 | 500.0 | 75.0 |
| pm2p5cl | $Cl^-$ | PM2.5 chloride | $\mu g\ m^{-3}$ | 0.2 | 0.0 | 150.0 | 30.0 |
| pm2p5cobalt | Co | PM2.5 cobalt | $ng\ m^{-3}$ | 0.1 | 0.0 | 50.0 | 5.0 |
| pm2p5cr | Cr | PM2.5 chromium | $ng\ m^{-3}$ | 1.0 | 0.0 | 500.0 | 100.0 |
| pm2p5cu | Cu | PM2.5 copper | $ng\ m^{-3}$ | 1.0 | 0.0 | 750.0 | 150.0 |
| pm2p5ec | C | PM2.5 elemental carbon | $\mu g\ m^{-3}$ | 10.0 | 0.0 | 25000.0 | 2500.0 |

*table continued on next page*

| GHOST Component Name | Chemical Formula | Long Component Name | Standard Units | Minimum Permitted Measurement Resolution | Extreme Lower Limit | Extreme Upper Limit | Extreme Upper Monthly Median |
|---|---|---|---|---|---|---|---|
| pm2p5fe | Fe | PM2.5 iron | ng m$^{-3}$ | 20.0 | 0.0 | 50000.0 | 5000.0 |
| pm2p5hg | Hg | PM2.5 mercury | pg m$^{-3}$ | 10.0 | 0.0 | 30000.0 | 3000.0 |
| pm2p5k | K$^+$ | PM2.5 potassium | $\mu$g m$^{-3}$ | 0.2 | 0.0 | 50.0 | 10.0 |
| pm2p5mg | Mg$^{2+}$ | PM2.5 magnesium | $\mu$g m$^{-3}$ | 0.2 | 0.0 | 50.0 | 10.0 |
| pm2p5mn | Mn | PM2.5 manganese | ng m$^{-3}$ | 2.0 | 0.0 | 5000.0 | 500.0 |
| pm2p5msa | CH$_4$O$_3$S | PM2.5 methanesulfonic acid | $\mu$g m$^{-3}$ | 0.2 | 0.0 | 75.0 | 25.0 |
| pm2p5na | Na$^+$ | PM2.5 sodium | $\mu$g m$^{-3}$ | 0.2 | 0.0 | 150.0 | 30.0 |
| pm2p5nh4 | NH$_4^+$ | PM2.5 ammonium | $\mu$g m$^{-3}$ | 0.2 | 0.0 | 150.0 | 30.0 |
| pm2p5nh4no3 | NH$_4$NO$_3$ | PM2.5 ammonium nitrate | $\mu$g m$^{-3}$ | 0.2 | 0.0 | 150.0 | 30.0 |
| pm2p5ni | Ni | PM2.5 nickel | ng m$^{-3}$ | 5.0 | 0.0 | 10000.0 | 1000.0 |
| pm2p5no3 | NO$_3^-$ | PM2.5 nitrate | $\mu$g m$^{-3}$ | 0.2 | 0.0 | 250.0 | 75.0 |
| pm2p5oc | C | PM2.5 organic carbon | $\mu$g m$^{-3}$ | 10.0 | 0.0 | 25000.0 | 2500.0 |
| pm2p5pb | Pb | PM2.5 lead | ng m$^{-3}$ | 50.0 | 0.0 | 60000.0 | 15000.0 |
| pm2p5se | Se | PM2.5 selenium | ng m$^{-3}$ | 0.2 | 0.0 | 150.0 | 30.0 |
| pm2p5so4 | SO$_4^{2-}$ | PM2.5 sulphate | $\mu$g m$^{-3}$ | 0.2 | 0.0 | 150.0 | 30.0 |
| pm2p5so4nss | SO$_4^{2-}$ | PM2.5 sulphate: non-sea salt | $\mu$g m$^{-3}$ | 0.2 | 0.0 | 150.0 | 30.0 |
| pm2p5so4ss | SO$_4^{2-}$ | PM2.5 sulphate: sea salt | $\mu$g m$^{-3}$ | 0.2 | 0.0 | 150.0 | 30.0 |
| pm2p5v | V | PM2.5 vanadium | ng m$^{-3}$ | 0.2 | 0.0 | 100.0 | 20.0 |
| pm2p5zn | Zn | PM2.5 zinc | ng m$^{-3}$ | 20.0 | 0.0 | 30000.0 | 5000.0 |
| **pm1** | | | | | | | |
| pm1 | ——- | total PM1 | $\mu$g m$^{-3}$ | 20.0 | 0.0 | 50000.0 | 5000.0 |
| pm1al | Al | PM1 aluminium | ng m$^{-3}$ | 20.0 | 0.0 | 50000.0 | 5000.0 |
| pm1as | As | PM1 arsenic | ng m$^{-3}$ | 1.0 | 0.0 | 1000.0 | 200.0 |
| pm1bc | C | PM1 black carbon | $\mu$g m$^{-3}$ | 10.0 | 0.0 | 25000.0 | 2500.0 |

*table continued on next page*


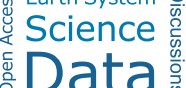

| GHOST Component Name | Chemical Formula | Long Component Name | Standard Units | Minimum Permitted Measurement Resolution | Extreme Lower Limit | Extreme Upper Limit | Extreme Upper Monthly Median |
|---|---|---|---|---|---|---|---|
| pm1c | C | PM1 carbon | $\mu$g m$^{-3}$ | 20.0 | 0.0 | 50000.0 | 5000.0 |
| pm1ca | Ca$^{2+}$ | PM1 calcium | $\mu$g m$^{-3}$ | 0.2 | 0.0 | 100.0 | 20.0 |
| pm1cd | Cd | PM1 cadmium | ng m$^{-3}$ | 0.2 | 0.0 | 500.0 | 75.0 |
| pm1cl | Cl$^-$ | PM1 chloride | $\mu$g m$^{-3}$ | 0.2 | 0.0 | 150.0 | 30.0 |
| pm1cobalt | Co | PM1 cobalt | ng m$^{-3}$ | 0.1 | 0.0 | 50.0 | 5.0 |
| pm1cr | Cr | PM1 chromium | ng m$^{-3}$ | 1.0 | 0.0 | 500.0 | 100.0 |
| pm1cu | Cu | PM1 copper | ng m$^{-3}$ | 1.0 | 0.0 | 750.0 | 150.0 |
| pm1ec | C | PM1 elemental carbon | $\mu$g m$^{-3}$ | 10.0 | 0.0 | 25000.0 | 2500.0 |
| pm1fe | Fe | PM1 iron | ng m$^{-3}$ | 20.0 | 0.0 | 50000.0 | 5000.0 |
| pm1hg | Hg | PM1 mercury | pg m$^{-3}$ | 10.0 | 0.0 | 30000.0 | 3000.0 |
| pm1k | K$^+$ | PM1 potassium | $\mu$g m$^{-3}$ | 0.2 | 0.0 | 50.0 | 10.0 |
| pm1mg | Mg$^{2+}$ | PM1 magnesium | $\mu$g m$^{-3}$ | 0.2 | 0.0 | 50.0 | 10.0 |
| pm1mn | Mn | PM1 manganese | ng m$^{-3}$ | 2.0 | 0.0 | 5000.0 | 500.0 |
| pm1msa | CH$_4$O$_3$S | PM1 methanesulfonic acid | $\mu$g m$^{-3}$ | 0.2 | 0.0 | 75.0 | 25.0 |
| pm1na | Na$^+$ | PM1 sodium | $\mu$g m$^{-3}$ | 0.2 | 0.0 | 150.0 | 30.0 |
| pm1nh4 | NH$_4^+$ | PM1 ammonium | $\mu$g m$^{-3}$ | 0.2 | 0.0 | 150.0 | 30.0 |
| pm1nh4no3 | NH$_4$NO$_3$ | PM1 ammonium nitrate | $\mu$g m$^{-3}$ | 0.2 | 0.0 | 150.0 | 30.0 |
| pm1ni | Ni | PM1 nickel | ng m$^{-3}$ | 5.0 | 0.0 | 10000.0 | 1000.0 |
| pm1no3 | NO$_3^-$ | PM1 nitrate | $\mu$g m$^{-3}$ | 0.2 | 0.0 | 250.0 | 75.0 |
| pm1oc | C | PM1 organic carbon | $\mu$g m$^{-3}$ | 10.0 | 0.0 | 25000.0 | 2500.0 |
| pm1pb | Pb | PM1 lead | ng m$^{-3}$ | 50.0 | 0.0 | 60000.0 | 15000.0 |
| pm1se | Se | PM1 selenium | ng m$^{-3}$ | 0.2 | 0.0 | 150.0 | 30.0 |
| pm1so4 | SO$_4^{2-}$ | PM1 sulphate | $\mu$g m$^{-3}$ | 0.2 | 0.0 | 150.0 | 30.0 |
| pm1so4nss | SO$_4^{2-}$ | PM1 sulphate: non-sea salt | $\mu$g m$^{-3}$ | 0.2 | 0.0 | 150.0 | 30.0 |

*table continued on next page*



| GHOST Component Name | Chemical Formula | Long Component Name | Standard Units | Minimum Permitted Measurement Resolution | Extreme Lower Limit | Extreme Upper Limit | Extreme Upper Monthly Median |
|---|---|---|---|---|---|---|---|
| pm1so4ss | $SO_4^{2-}$ | PM1 sulphate: sea salt | $\mu g\ m^{-3}$ | 0.2 | 0.0 | 150.0 | 30.0 |
| pm1v | V | PM1 vanadium | $ng\ m^{-3}$ | 0.2 | 0.0 | 100.0 | 20.0 |
| pm1zn | Zn | PM1 zinc | $ng\ m^{-3}$ | 20.0 | 0.0 | 30000.0 | 5000.0 |
| **aod** | | | | | | | |
| od380aero | —— | aerosol optical depth at 380nm | unitless | —— | 0.0 | 20.0 | —— |
| od440aero | —— | aerosol optical depth at 440nm | unitless | —— | 0.0 | 20.0 | —— |
| od500aero | —— | aerosol optical depth at 500nm | unitless | —— | 0.0 | 20.0 | —— |
| od500aerocoarse | —— | coarse mode aerosol optical depth at 500nm | unitless | —— | 0.0 | 20.0 | —— |
| od500aerofine | —— | fine mode aerosol optical depth at 500nm | unitless | —— | 0.0 | 20.0 | —— |
| fm500frac | —— | fine mode aerosol optical depth fraction at 500nm | unitless | —— | 0.0 | 1.0 | —— |
| od550aero | —— | aerosol optical depth at 550nm | unitless | —— | 0.0 | 20.0 | —— |
| od675aero | —— | aerosol optical depth at 675nm | unitless | —— | 0.0 | 20.0 | —— |
| od870aero | —— | aerosol optical depth at 870nm | unitless | —— | 0.0 | 20.0 | —— |
| od1020aero | —— | aerosol optical depth at 1020nm | unitless | —— | 0.0 | 20.0 | —— |
| ae440-870aero | —— | angstrom exponent between 440 and 870 nm | unitless | —— | 0.0 | 4.0 | —— |

*table continued on next page*



| GHOST Component Name | Chemical Formula | Long Component Name | Standard Units | Minimum Permitted Measurement Resolution | Extreme Lower Limit | Extreme Upper Limit | Extreme Upper Monthly Median |
|---|---|---|---|---|---|---|---|
| **extaod** | | | | | | | |
| extod440aero | ——- | extinction aerosol optical depth at 440nm | unitless | ——- | 0.0 | 20.0 | ——- |
| extod440aerocoarse | ——- | extinction coarse mode aerosol optical depth at 440nm | unitless | ——- | 0.0 | 20.0 | ——- |
| extod440aerofine | ——- | extinction fine mode aerosol optical depth at 440nm | unitless | ——- | 0.0 | 20.0 | ——- |
| extod675aero | ——- | extinction aerosol optical depth at 675nm | unitless | ——- | 0.0 | 20.0 | ——- |
| extod675aerocoarse | ——- | extinction coarse mode aerosol optical depth at 675nm | unitless | ——- | 0.0 | 20.0 | ——- |
| extod675aerofine | ——- | extinction fine mode aerosol optical depth at 675nm | unitless | ——- | 0.0 | 20.0 | ——- |
| extod870aero | ——- | extinction aerosol optical depth at 870nm | unitless | ——- | 0.0 | 20.0 | ——- |
| extod870aerocoarse | ——- | extinction coarse mode aerosol optical depth at 870nm | unitless | ——- | 0.0 | 20.0 | ——- |
| extod870aerofine | ——- | extinction fine mode aerosol optical depth at 870nm | unitless | ——- | 0.0 | 20.0 | ——- |





| GHOST Component Name | Chemical Formula | Long Component Name | Standard Units | Minimum Permitted Measurement Resolution | Extreme Lower Limit | Extreme Upper Limit | Extreme Upper Monthly Median |
|---|---|---|---|---|---|---|---|
| extod1020aero | ——- | extinction aerosol optical depth at 1020nm | unitless | ——- | 0.0 | 20.0 | ——- |
| extod1020aerocoarse | ——- | extinction coarse mode aerosol optical depth at 1020nm | unitless | ——- | 0.0 | 20.0 | ——- |
| extod1020aerofine | ——- | extinction fine mode aerosol optical depth at 1020nm | unitless | ——- | 0.0 | 20.0 | ——- |
| extae440-870aero | ——- | extinction angstrom exponent between 440 and 870 nm | unitless | ——- | 0.0 | 4.0 | ——- |
| **absaod** | | | | | | | |
| absod440aero | ——- | absorption aerosol optical depth at 440nm | unitless | ——- | 0.0 | 20.0 | ——- |
| absod675aero | ——- | absorption aerosol optical depth at 675nm | unitless | ——- | 0.0 | 20.0 | ——- |
| absod870aero | ——- | absorption aerosol optical depth at 870nm | unitless | ——- | 0.0 | 20.0 | ——- |
| absod1020aero | ——- | absorption aerosol optical depth at 1020nm | unitless | ——- | 0.0 | 20.0 | ——- |
| absae440-870aero | ——- | absorption angstrom exponent between 440 and 870 nm | unitless | ——- | 0.0 | 4.0 | ——- |





| GHOST Component Name | Chemical Formula | Long Component Name | Standard Units | Minimum Permitted Measurement Resolution | Extreme Lower Limit | Extreme Upper Limit | Extreme Upper Monthly Median |
|---|---|---|---|---|---|---|---|
| **ssa** | | | | | | | |
| sca440aero | ——- | single scattering albedo at 440nm | unitless | ——- | 0.0 | 1.0 | ——- |
| sca675aero | ——- | single scattering albedo at 675nm | unitless | ——- | 0.0 | 1.0 | ——- |
| sca870aero | ——- | single scattering albedo at 870nm | unitless | ——- | 0.0 | 1.0 | ——- |
| sca1020aero | ——- | single scattering albedo at 1020nm | unitless | ——- | 0.0 | 1.0 | ——- |
| **asy** | | | | | | | |
| asy440aero | ——- | asymmetry factor at 440nm | unitless | ——- | 0.0 | 2.0 | ——- |
| asy440aerocoarse | ——- | coarse mode asymmetry factor at 440nm | unitless | ——- | 0.0 | 2.0 | ——- |
| asy440aerofine | ——- | fine mode asymmetry factor at 440nm | unitless | ——- | 0.0 | 2.0 | ——- |
| asy675aero | ——- | asymmetry factor at 675nm | unitless | ——- | 0.0 | 2.0 | ——- |
| asy675aerocoarse | ——- | coarse mode asymmetry factor at 675nm | unitless | ——- | 0.0 | 2.0 | ——- |
| asy675aerofine | ——- | fine mode asymmetry factor at 675nm | unitless | ——- | 0.0 | 2.0 | ——- |
| asy870aero | ——- | asymmetry factor at 870nm | unitless | ——- | 0.0 | 2.0 | ——- |
| asy870aerocoarse | ——- | coarse mode asymmetry factor at 870nm | unitless | ——- | 0.0 | 2.0 | ——- |



| GHOST Component Name | Chemical Formula | Long Component Name | Standard Units | Minimum Permitted Measurement Resolution | Extreme Lower Limit | Extreme Upper Limit | Extreme Upper Monthly Median |
|---|---|---|---|---|---|---|---|
| asy870aerofine | ——- | fine mode asymmetry factor at 870nm | unitless | ——- | 0.0 | 2.0 | ——- |
| asy1020aero | ——- | asymmetry factor at 1020nm | unitless | ——- | 0.0 | 2.0 | ——- |
| asy1020aerocoarse | ——- | coarse mode asymmetry factor at 1020nm | unitless | ——- | 0.0 | 2.0 | ——- |
| asy1020aerofine | ——- | fine mode asymmetry factor at 1020nm | unitless | ——- | 0.0 | 2.0 | ——- |
| sphaero | ——- | sphericity factor | unitless | ——- | 0.0 | 100.0 | ——- |
| **rin** | | | | | | | |
| rinreal440 | ——- | real part of the refractive index at 440nm | unitless | ——- | 1.0 | 2.0 | ——- |
| rinreal675 | ——- | real part of the refractive index at 675nm | unitless | ——- | 1.0 | 2.0 | ——- |
| rinreal870 | ——- | real part of the refractive index at 870nm | unitless | ——- | 1.0 | 2.0 | ——- |
| rinreal1020 | ——- | real part of the refractive index at 1020nm | unitless | ——- | 1.0 | 2.0 | ——- |
| rinimag440 | ——- | imaginary part of the refractive index at 440nm | unitless | ——- | 0.0 | 0.1 | ——- |
| rinimag675 | ——- | imaginary part of the refractive index at 675nm | unitless | ——- | 0.0 | 0.1 | ——- |



| GHOST Component Name | Chemical Formula | Long Component Name | Standard Units | Minimum Permitted Measurement Resolution | Extreme Lower Limit | Extreme Upper Limit | Extreme Upper Monthly Median |
|---|---|---|---|---|---|---|---|
| rinimag870 | ——- | imaginary part of the refractive index at 870nm | unitless | ——- | 0.0 | 0.1 | ——- |
| rinimag1020 | ——- | imaginary part of the refractive index at 1020nm | unitless | ——- | 0.0 | 0.1 | ——- |
| **vconc** | | | | | | | |
| vconcaero | ——- | normalised total volume concentration (dV(r)/dln(r)) | $\mu m^3\ \mu m^{-2}$ | ——- | 0.0 | 20.0 | ——- |
| vconcaerocoarse | ——- | normalised total coarse mode volume concentration (dV(r)/dln(r)) | $\mu m^3\ \mu m^{-2}$ | ——- | 0.0 | 20.0 | ——- |
| vconcaerofine | ——- | normalised total fine mode volume concentration (dV(r)/dln(r)) | $\mu m^3\ \mu m^{-2}$ | ——- | 0.0 | 20.0 | ——- |
| **size** | | | | | | | |
| vconcaerobin1 | ——- | normalised volume concentration at 0.05 $\mu m$ (dV(r)/dln(r)) | $\mu m^3\ \mu m^{-2}$ | ——- | 0.0 | 2.0 | ——- |
| vconcaerobin2 | ——- | normalised volume concentration at 0.065604 $\mu m$ (dV(r)/dln(r)) | $\mu m^3\ \mu m^{-2}$ | ——- | 0.0 | 2.0 | ——- |
| vconcaerobin3 | ——- | normalised volume concentration at 0.086077 $\mu m$ (dV(r)/dln(r)) | $\mu m^3\ \mu m^{-2}$ | ——- | 0.0 | 2.0 | ——- |

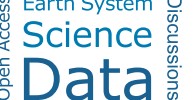

| GHOST Component Name | Chemical Formula | Long Component Name | Standard Units | Minimum Permitted Measurement Resolution | Extreme Lower Limit | Extreme Upper Limit | Extreme Upper Monthly Median |
|---|---|---|---|---|---|---|---|
| vconcaerobin4 | ——- | normalised volume concentration at 0.112939 $\mu$m (dV(r)/dln(r)) | $\mu$m$^3$ $\mu$m$^{-2}$ | ——- | 0.0 | 2.0 | ——- |
| vconcaerobin5 | ——- | normalised volume concentration at 0.148184 $\mu$m (dV(r)/dln(r)) | $\mu$m$^3$ $\mu$m$^{-2}$ | ——- | 0.0 | 2.0 | ——- |
| vconcaerobin6 | ——- | normalised volume concentration at 0.194429 $\mu$m (dV(r)/dln(r)) | $\mu$m$^3$ $\mu$m$^{-2}$ | ——- | 0.0 | 2.0 | ——- |
| vconcaerobin7 | ——- | normalised volume concentration at 0.255105 $\mu$m (dV(r)/dln(r)) | $\mu$m$^3$ $\mu$m$^{-2}$ | ——- | 0.0 | 2.0 | ——- |
| vconcaerobin8 | ——- | normalised volume concentration at 0.334716 $\mu$m (dV(r)/dln(r)) | $\mu$m$^3$ $\mu$m$^{-2}$ | ——- | 0.0 | 2.0 | ——- |
| vconcaerobin9 | ——- | normalised volume concentration at 0.439173 $\mu$m (dV(r)/dln(r)) | $\mu$m$^3$ $\mu$m$^{-2}$ | ——- | 0.0 | 2.0 | ——- |
| vconcaerobin10 | ——- | normalised volume concentration at 0.576227 $\mu$m (dV(r)/dln(r)) | $\mu$m$^3$ $\mu$m$^{-2}$ | ——- | 0.0 | 2.0 | ——- |





| GHOST Component Name | Chemical Formula | Long Component Name | Standard Units | Minimum Permitted Measurement Resolution | Extreme Lower Limit | Extreme Upper Limit | Extreme Upper Monthly Median |
|---|---|---|---|---|---|---|---|
| vconcaerobin11 | ——- | normalised volume concentration at 0.756052 $\mu$m (dV(r)/dln(r)) | $\mu$m$^3$ $\mu$m$^{-2}$ | ——- | 0.0 | 2.0 | ——- |
| vconcaerobin12 | ——- | normalised volume concentration at 0.991996 $\mu$m (dV(r)/dln(r)) | $\mu$m$^3$ $\mu$m$^{-2}$ | ——- | 0.0 | 2.0 | ——- |
| vconcaerobin13 | ——- | normalised volume concentration at 1.301571 $\mu$m (dV(r)/dln(r)) | $\mu$m$^3$ $\mu$m$^{-2}$ | ——- | 0.0 | 2.0 | ——- |
| vconcaerobin14 | ——- | normalised volume concentration at 1.707757 $\mu$m (dV(r)/dln(r)) | $\mu$m$^3$ $\mu$m$^{-2}$ | ——- | 0.0 | 2.0 | ——- |
| vconcaerobin15 | ——- | normalised volume concentration at 2.240702 $\mu$m (dV(r)/dln(r)) | $\mu$m$^3$ $\mu$m$^{-2}$ | ——- | 0.0 | 2.0 | ——- |
| vconcaerobin16 | ——- | normalised volume concentration at 2.939966 $\mu$m (dV(r)/dln(r)) | $\mu$m$^3$ $\mu$m$^{-2}$ | ——- | 0.0 | 2.0 | ——- |
| vconcaerobin17 | ——- | normalised volume concentration at 3.857452 $\mu$m (dV(r)/dln(r)) | $\mu$m$^3$ $\mu$m$^{-2}$ | ——- | 0.0 | 2.0 | ——- |





| GHOST Component Name | Chemical Formula | Long Component Name | Standard Units | Minimum Permitted Measurement Resolution | Extreme Lower Limit | Extreme Upper Limit | Extreme Upper Monthly Median |
|---|---|---|---|---|---|---|---|
| vconcaerobin18 | ——- | normalised volume concentration at 5.061260 $\mu$m (dV(r)/dln(r)) | $\mu$m$^3$ $\mu$m$^{-2}$ | ——- | 0.0 | 2.0 | ——- |
| vconcaerobin19 | ——- | normalised volume concentration at 6.640745 $\mu$m (dV(r)/dln(r)) | $\mu$m$^3$ $\mu$m$^{-2}$ | ——- | 0.0 | 2.0 | ——- |
| vconcaerobin20 | ——- | normalised volume concentration at 8.713145 $\mu$m (dV(r)/dln(r)) | $\mu$m$^3$ $\mu$m$^{-2}$ | ——- | 0.0 | 2.0 | ——- |
| vconcaerobin21 | ——- | normalised volume concentration at 11.432287 $\mu$m (dV(r)/dln(r)) | $\mu$m$^3$ $\mu$m$^{-2}$ | ——- | 0.0 | 2.0 | ——- |
| vconcaerobin22 | ——- | normalised volume concentration at 15.00um $\mu$m (dV(r)/dln(r)) | $\mu$m$^3$ $\mu$m$^{-2}$ | ——- | 0.0 | 2.0 | ——- |



**Table A4.** Definitions of the fields associated with each GHOST standard station classification metadata variable. Some of the fields also contain sub-fields where extra information from the data provider allows for finer grained classification.

| Field | Sub-field | Description |
|---|---|---|
| **area_classification** | | |
| urban | ——- | All areas with a level of urban influence. |
| urban | centre | Continuously built-up urban area, which is defined as the street front being built up by buildings with at least two floors. With the exception of parks, this area is not mixed with non-urbanised zones. |
| urban | suburban | A largely built-up urban area, this being defined as a contiguous settlement of detached buildings of any size with a building density less than that of in an urban-centre. The area is often interspersed with non-urbanised zones (e.g. lakes, woods). It must also be noted that "suburban" as defined here has a different meaning than in every day English i.e. "an outlying part of a city or town", suggesting that a suburban area is always attached to an urban-centre. A suburban area as defined here can be entirely detached from any urban-centre. |
| rural | ——- | All areas, that do not fulfil the criteria for an "urban" area are defined as "rural". |
| rural | near_city | Rural area which is within 10 km of an urban area / major pollution source. |
| rural | regional | Rural area which is 10-50 km from an urban area / major pollution source. |
| rural | remote | Rural area which is > 50 km from an urban area / major pollution source. |
| **station_classification** | | |
| background | ——- | Station located such that the air is representative of the average conditions within the area. Any pollution should not be dominated by a single source type (e.g. traffic), unless that source type is typical within the area. The station should usually be representative of a wider area of at least several square kilometres. |
| point_source | ——- | Station located such that the air is influenced by a major stationary emissions source (e.g. power plant), or influenced by traffic, rail, marine, or aviation sources. |
| point_source | industrial | Station located in close proximity to industrial sources of pollution. These sources can include: thermal power generation, district heating plants, refineries, waste incineration / treatment plants, dump sites, mining, airports, and ports. |
| point_source | traffic | Station located in close proximity to a road, and such that pollution levels are dominated by the emissions from road traffic. |

*table continued on next page*



| Field | Sub-field | Description |
|---|---|---|
| **main_emission_source** | | |
| agriculture | ——- | Emissions associated with agriculture |
| commercial_and_ residential_ combustion | ——- | Emissions associated with commercial and residential combustion. |
| extraction_of_ fossil_fuels | ——- | Emissions associated with the extraction of fossil fuels. |
| industrial_ combustion | ——- | Emissions associated with industrial combustion. |
| natural | ——- | Emissions from natural sources (e.g. terpenes from trees). |
| other_mobile_ sources_and_ machinery | ——- | Emissions from all other mobile sources than traffic, and from off-road vehicles and engines. |
| production_ processes | ——- | Emissions from processes associated with production and assembly. |
| power_production | ——- | Emissions from processes associated with the generation of power. |
| road_transport | ——- | Emissions from road traffic. |
| solvents | ——- | Emissions associated with use of solvents. |
| waste_treatment_ and_disposal | ——- | Emissions associated with waste treatment and disposal. |
| **land_use** | | |
| barren | ——- | Lands with exposed soil, sand or rocks, which never have more than 10% vegetated cover during any time of the year. |
| barren | beach | Land alongside a body of water which consists of loose particles, typically made from rock (e.g. sand or gravel). |
| barren | desert | A barren area of land, where little precipitation occurs, and consequently living conditions are hostile for plant and animal life. |
| barren | rock | Lands characterised by areas of bedrock exposure, scarps, talus, slides, volcanic material, rock glaciers, and other accumulations of rock without vegetative cover. |
| barren | soil | Lands with thin soil, without vegetation. |
| forest | ——- | Lands dominated by woody vegetation or trees, with > 60% cover, and height exceeding 2 m. Includes all evergreen needleleaf, evergreen broadleaf, deciduous needleleaf, deciduous broadleaf vegetation types. |



| Field | Sub-field | Description |
|---|---|---|
| open | ——- | Lands with herbaceous, other understory systems or woody vegetation less than 2m in height. |
| open | grassland | Lands with herbaceous types of cover. Tree and shrub cover is < 10%. |
| open | savanna | Lands with herbaceous and other understory systems, and forest canopy cover between 10% and 60%, and height exceeding 2m. |
| open | shrubland | Lands with woody vegetation less than 2m in height and with shrub canopy cover > 10%. The shrub foliage can be either evergreen or deciduous. |
| snow | ——- | Lands under snow / ice cover throughout the year. |
| urban | ——- | Land covered by buildings and other man-made structures. |
| urban | agricultural | Lands covered with temporary crops which have a harvest and a bare soil period. Also includes lands used for farming and raising of livestock. |
| urban | blighted | An area that by reason of deterioration, faulty planning, inadequate or improper facilities, deleterious land use or the existence of unsafe structures, or any combination of these factors, is detrimental to the safety, health or welfare of the community. |
| urban | commercial | Land dominated by real estate intended for use by for-profit businesses, such as office complexes, shopping centres, service stations, and restaurants. |
| urban | industrial | Land used for industrial purposes, e.g. manufacturing. |
| urban | military | Land used for solely military purposes. |
| urban | park | A large public garden, or area of land used for recreation. |
| urban | residential | Land used mainly for housing. |
| urban | transportation | All types of land use used for human transportation. This includes airports, roads, railway lines, and shipping ports. |
| water | ——- | Oceans, seas, lakes, reservoirs, and rivers. Can be either fresh or saltwater bodies. |
| wetland | ——- | Lands with a permanent mixture of water and herbaceous or woody vegetation. The vegetation can be present either in salt, brackish, or fresh water. |
| **terrain** | | |
| coastal | ——- | An area where the land meets the sea or ocean. |
| complex | ——- | A region having irregular topography (not including mountains or coastal). Complex terrain can include variations in land use, such as urban, irrigated, and unirrigated. |



| Field | Sub-field | Description |
|---|---|---|
| flat | ——- | Open terrain, country or ground which is mostly flat and free of obstructions such as trees and buildings. Examples include farmland or grassland. |
| mountain | ——- | A large landform that stretches above the surrounding land in a limited area, usually in the form of a peak. |
| rolling | ——- | Terrain where the natural slopes consistently rise and fall across a horizontal plane. |
| **measurement_scale** | | |
| micro | ——- | Representative for: 1m – 100m, i.e. a small street. |
| middle | ——- | Representative for: 100m – 0.5km, i.e. several city blocks. |
| neighbourhood | ——- | Representative for: 0.5km – 4km, i.e. some extended area of city that has relatively uniform land use. |
| city | ——- | Representative for: 4km – 50km, i.e. city like dimensions. |
| regional | ——- | Representative for: 50km – 100s km, i.e. a rural area of reasonably homogeneous geography, without large pollution sources. |





**Table A5.** Outline of the GHOST standard sampling types, with a description given for each type. These are set in the "primary_sampling_type" and / or "measuring_instrument_sampling_type" variables, dependent on the measurement process. For each type there are several standardised primary sampling instruments (83 in total across types), set in the "primary_sampling_instrument_name" variable. Measurements utilising a primary sampling instrument of a type that they are not associated with are given the "Erroneous Primary Sampling" (code 20) GHOST QA flag. Measurements utilising a primary sampling instrument whose type or name are unknown are given the "Unknown Primary Sampling Type" (code 14), and "Unknown Primary Sampling Instrument" (code 15) GHOST QA flags respectively. Any measurements where any assumptions are made regarding the primary sampling are given the "Assumed Primary Sampling" (code 11) GHOST QA flag.

| Sampling Type | Description |
|---|---|
| low volume continuous | Ambient air is continuously drawn in using a low volume sampler (typically sampling < 24,000L / 24-hours). These samplers can have in-built filters, designed to specifically retain certain components. |
| high volume continuous | Ambient air is continuously drawn in using a high volume sampler instrumentation (typically sampling > 100,000 L / 24-hours). These samplers can have in-built filters, designed to specifically retain certain components. |
| injection | The measuring instrument is injected with a limited quantity of air. The injected sample is typically pre-processed to aid the detection of a specific component. |
| continuous injection | The measuring instrument is periodically injected with limited quantities of air. The injected samples can either be from continuous automated collection, or from pre-processed loaded samples. |
| passive | Air is not drawn in, rather the sample is the ambient air which interacts with the measurement apparatus. |
| remote | The measuring instrument does not actively sample air, but uses advanced optical techniques to measure components in the air over long distances. |
| manual | No instrument is used to determine measured values, they are determined manually e.g. for some colorimetric methods measurement values are derived manually via the colour of the reagent after a reaction with a component of interest. |
| unknown | Sampling type is unknown. |





**Table A6.** Outline of the GHOST standard sample preparation types and techniques, with a description given for each type. These are set in the "sample_preparation_types", and "sample_preparation_techniques" variables. Each preparation type can have multiple sub-techniques. Measurements which use a preparation type that they are not associated with are given the "Erroneous Sample Preparation" (code 21) GHOST QA flag. When sample preparation of a given type / technique is utilised, but is unknown, then measurements are given the "Unknown Sample Preparation Type" (code 16), and "Unknown Sample Preparation Technique" (code 17) GHOST QA flags respectively. Any measurements where any assumptions are made regarding the sample preparation are given the "Assumed Sample Preparation" (code 12) GHOST QA flag.

| Preparation Type | Specific Techniques | Description |
|---|---|---|
| flask | ——- | Sample is collected in measurement flasks / canisters from ambient air, or filled by a pump. The canisters can be filled in a short window, or in quick bursts over a longer window to get a more representative sample. |
| bag | ——- | Sample is collected in gas sampling bags (typically teflon) from ambient air, or filled by a pump. These bags are a cheap alternative to canisters, with much reduced stability times. |
| preconcentration | ——- | Process of concentrating a sample before analysis, so that trace components can be more easily identified. This is done typically through absorption of the sample onto a cooled, sorbent-packed trap before thermal desorption to transfer the sample very quickly to the analytical system. |
| filter | ——- | Air is passed through a filtering system, selectively retaining compound(s) of interest. |
| filter pack | 1-stage filter pack, 2-stage filter pack, 3-stage filter pack, 4-stage filter pack | Air is passed through a filter pack, selectively retaining compound(s) of interest. Filter packs can contain multiple different filters, or stages, which target the retention of different components. |
| denuder | CEH DELTA, Riemer DEN2, UBA Olaf | Air is passed through a denuder before analysis to selectively retain compound(s) of interest. A denuder is cylindrical or annular conduit or tube internally coated with a reagent that selectively reacts with certain components. |
| sorbent trapping | diffusive sampler | Sample is passed through a sorbent material to trap and retain compound(s) of interest. Diffusive samplers use sorbent trapping to passively trap components over long time periods. |
| reagent reaction | Griess-Saltzman, Lyshkow, Jacobs-Hochheiser, Sodium Arsenite, TEA, TGS-ANSA, Sodium Phenolate, Nessler, Pararosaniline, Hydrogen Peroxide, Potassium Iodide, detection tube | Air is reacted with a liquid / solid chemical reagent to allow subsequent measurement of a specific compound. |



| Preparation Type | Specific Techniques | Description |
|---|---|---|
| intermediate measurement | ——- | A measurement is made using a certain method prior to a further method being used, e.g. measuring the PM size fraction concentration, before measuring the speciation of that size fraction. |
| unknown | ——- | Sample preparation type is unknown. |



**Table A7.** Outline of the GHOST standard measurement methods, set in the "measurement_methodology" variable. Associated with each method is an abbreviated code (e.g. UVP), which is also included in the "station_reference" variable (e.g. AHP_UVP). For each method the associated default sampling type, and sample preparation are stated, these set in the "measuring_instrument_sampling_type" and "sample_preparation_types" variables respectively. Stated also are the components that each method is known to measure, and the components which are accepted by GHOST QA to acceptably measure (i.e. without major known biases). For each method there are several standardised instruments that employ that method (508 in total across methods), set in the "measuring_instrument_name" variable. Components measured with a method either that they are not associated with, or not accepted by GHOST QA are given the "Erroneous Measurement Methodology" (code 22), and "Invalid QA Measurement Methodology" (code 23) GHOST QA flags respectively. Measurements for which the methodology or measuring instrument are unknown are given the "Unknown Measurement Method" (code 18), and "Unknown Measuring Instrument" (code 19) GHOST QA flags respectively. Any measurements where any assumptions are made regarding the method are given the "Assumed Measurement Methodology" (code 13) GHOST QA flag.

| Measurement Method | Sampling Type / Sample Preparation | Measured Components | QA Accepted Components |
|---|---|---|---|
| ultraviolet photometry (UVP) | low volume continuous | $O_3$ | $O_3$ |
| visible photometry (VP) | low volume continuous | $NO$, $NO_2$ | $NO$, $NO_2$ |
| ethylene chemiluminescence (ECL) | low volume continuous | $O_3$ | $O_3$ |
| eosin Y chemiluminescence (EYCL) | low volume continuous | $O_3$ | $O_3$ |
| rhodamine B chemiluminescence (RBC) | low volume continuous | $O_3$ | $O_3$ |
| chemiluminescence (internal molybdenum converter) (CL(IMC)) | low volume continuous | $NO$, $NO_2$, $O_3$ | $NO$, $O_3$ |
| chemiluminescence (external molybdenum converter) (CL(EMC)) | low volume continuous | $NO$, $NH_3$, $HNO_3$ | $NO$, $NH_3$, $HNO_3$ |
| chemiluminescence (internal photolytic converter) (CL(IPC)) | low volume continuous | $NO$, $NO_2$ | $NO$, $NO_2$ |
| chemiluminescence (internal molybdenum and quartz converters) (CL(IMQC)) | low volume continuous | $NO$, $NO_2$, $NH_3$, $HNO_3$ | $NO$, $NH_3$, $HNO_3$ |
| chemiluminescence (internal molybdenum converter and external quartz converter) (CL(IMC-EQC)) | low volume continuous | $NO$, $NO_2$, $NH_3$, $HNO_3$ | $NO$, $NH_3$, $HNO_3$ |
| chemiluminescence (internal molybdenum and stainless steel converters) (CL(IMSC)) | low volume continuous | $NO$, $NO_2$, $NH_3$, $HNO_3$ | $NO$, $NH_3$, $HNO_3$ |
| chemiluminescence (internal molybdenum converter and external stainless steel converter) (CL(IMC-ESC)) | low volume continuous | $NO$, $NO_2$, $NH_3$, $HNO_3$ | $NO$, $NH_3$, $HNO_3$ |





| Measurement Method | Sampling Type / Sample Preparation | Measured Components | QA Accepted Components |
|---|---|---|---|
| thermal reduction – chemiluminescence (TR-CL) | low volume continuous / filter | $NO_3^-$ | $NO_3^-$ |
| flame photometric detection (FPD) | low volume continuous | $SO_2$, $H_2S$, $K^+$, $SO_4^{2-}$ | $SO_2$, $H_2S$, $K^+$, $SO_4^{2-}$ |
| flame ionisation detection (FID) | low volume continuous | CO, $CH_4$, All VOC compounds, NMVOC, VOC, NMHC, HC, $CH_2O$ | VOC, HC |
| selective combustion – flame ionisation detection (SC-FID) | low volume continuous | CO, $CH_4$, All VOC compounds, NMVOC, VOC, NMHC, HC, $CH_2O$ | CO, $CH_4$, All VOC compounds, NMVOC, VOC, NMHC, HC, $CH_2O$ |
| conductimetry (CD) | low volume continuous / reagent reaction | $SO_2$, $NH_3$, $HNO_3$, HCl, $H_2S$ | $NH_3$, $HNO_3$, HCl |
| coulometry (CM) | low volume continuous / reagent reaction | $O_3$, NO, $NO_2$, $SO_2$, CO, $H_2S$ | ——- |
| polarography (PO) | injection | NO, $NO_2$, $SO_2$, $H_2S$ | ——- |
| capillary electrophoresis (CE) | injection | 10+ components | 10+ components |
| ultraviolet fluorescence (UVF) | low volume continuous | $SO_2$, $H_2S$ | $SO_2$, $H_2S$ |
| thermal reduction – ultraviolet fluorescence (TR-UVF) | low volume continuous / filter | $SO_4^{2-}$ | $SO_4^{2-}$ |
| laser-induced fluorescence (LIF) | low volume continuous | NO, $NO_2$ | NO, $NO_2$ |
| vacuum ultraviolet resonance fluorescence (VURF) | low volume continuous | CO | CO |
| cavity ringdown spectroscopy (CRDS) | low volume continuous | 10+ components | 10+ components |
| off-axis integrated cavity output spectroscopy (OA-ICOS) | low volume continuous | 10+ components | 10+ components |
| tunable diode laser absorption spectroscopy (TDLAS) | low volume continuous | 10+ components | 10+ components |
| cavity attenuated phase shift spectroscopy (CAPS) | low volume continuous | NO, $NO_2$ | NO, $NO_2$ |
| differential optical absorption spectroscopy (DOAS) | remote | 10+ components | 10+ components |
| electrochemical membrane diffusion (EMD) | low volume continuous | $NH_3$, $HNO_3$ | $NH_3$, $HNO_3$ |


Open Access    Earth System    Discussions
Science
Data

| Measurement Method | Sampling Type / Sample Preparation | Measured Components | QA Accepted Components |
|---|---|---|---|
| photoacoustic spectroscopy (PS) | low volume continuous | $NH_3$, $HNO_3$ | $NH_3$, $HNO_3$ |
| non-dispersive infrared absorption (luft) (NDIR-L) | low volume continuous | CO, $CH_4$ | CO, $CH_4$ |
| non-dispersive infrared absorption (gas-filter correlation) (NDIR-GFC) | low volume continuous | CO, $CH_4$ | CO, $CH_4$ |
| non-dispersive infrared absorption (cross-flow modulation) (NDIR-CFM) | low volume continuous | CO, $CH_4$ | CO, $CH_4$ |
| dual isotope fluorescence (DIF) | low volume continuous | CO | CO |
| fourier transform infrared spectroscopy (FTIR) | low volume continuous | CO, $CH_4$, All VOC compounds, NMVOC, VOC, NMHC, HC, $CH_2O$ | CO, $CH_4$, All VOC compounds, NMVOC, VOC, NMHC, HC, $CH_2O$ |
| gas chromatography – unknown detection (GC-UNK) | injection | CO, $CH_4$, All VOC compounds, NMVOC, VOC, NMHC, HC, $CH_2O$ | CO, $CH_4$, All VOC compounds, NMVOC, VOC, NMHC, HC, $CH_2O$ |
| gas chromatography – flame ionisation detection (GC-FID) | injection | CO, $CH_4$, All VOC compounds, NMVOC, VOC, NMHC, HC, $CH_2O$ | CO, $CH_4$, All VOC compounds, NMVOC, VOC, NMHC, HC, $CH_2O$ |
| gas chromatography – dual flame ionisation detection (GC-DFID) | injection | CO, $CH_4$, All VOC compounds, NMVOC, VOC, NMHC, HC, $CH_2O$ | CO, $CH_4$, All VOC compounds, NMVOC, VOC, NMHC, HC, $CH_2O$ |
| gas chromatography – electron capture detection (GC-ECD) | injection | CO, $CH_4$, All VOC compounds, NMVOC, VOC, NMHC, HC, $C_2H_3NO_5$, $CH_2O$ | CO, $CH_4$, All VOC compounds, NMVOC, VOC, NMHC, HC, $C_2H_3NO_5$, $CH_2O$ |
| gas chromatography – photoionisation detection (GC-PID) | injection | CO, $CH_4$, All VOC compounds, NMVOC, VOC, NMHC, HC, $CH_2O$ | CO, $CH_4$, All VOC compounds, NMVOC, VOC, NMHC, HC, $CH_2O$ |
| gas chromatography – mercuric oxide reduction detection (GC-HgO) | injection | CO | CO |
| gas chromatography – fourier transform infrared spectroscopy (GC-FTIR) | injection | CO, $CH_4$, All VOC compounds, NMVOC, VOC, NMHC, HC, $CH_2O$ | CO, $CH_4$, All VOC compounds, NMVOC, VOC, NMHC, HC, $CH_2O$ |

| Measurement Method | Sampling Type / Sample Preparation | Measured Components | QA Accepted Components |
|---|---|---|---|
| gas chromatography – mass spectrometry (GC-MS) | injection | 10+ components | 10+ components |
| pyrolysis – gas chromatography – mass spectrometry (Py-GC-MS) | injection | black C | black C |
| gas chromatography – direct temperature resolved mass spectrometry (GC-DTMS) | injection | CO, $CH_4$, All VOC compounds, NMVOC, VOC, NMHC, HC, $CH_2O$ | CO, $CH_4$, All VOC compounds, NMVOC, VOC, NMHC, HC, $CH_2O$ |
| gas chromatography – mass spectrometry – flame ionisation detection (GC-MS-FID) | injection | CO, $CH_4$, All VOC compounds, NMVOC, VOC, NMHC, HC, $CH_2O$ | CO, $CH_4$, All VOC compounds, NMVOC, VOC, NMHC, HC, $CH_2O$ |
| gas chromatography – mass spectrometry – photoionisation detection (GC-MS-PID) | injection | CO, $CH_4$, All VOC compounds, NMVOC, VOC, NMHC, HC, $CH_2O$ | CO, $CH_4$, All VOC compounds, NMVOC, VOC, NMHC, HC, $CH_2O$ |
| gas chromatography – electron capture detection – photoionisation detection (GC-ECD-PID) | injection | CO, $CH_4$, All VOC compounds, NMVOC, VOC, NMHC, HC, $C_2H_3NO_5$, $CH_2O$ | CO, $CH_4$, All VOC compounds, NMVOC, VOC, NMHC, HC, $C_2H_3NO_5$, $CH_2O$ |
| gas chromatography – flame ionisation detection – electron capture detection (GC-FID-ECD) | injection | CO, $CH_4$, All VOC compounds, NMVOC, VOC, NMHC, HC, $C_2H_3NO_5$, $CH_2O$ | CO, $CH_4$, All VOC compounds, NMVOC, VOC, NMHC, HC, $C_2H_3NO_5$, $CH_2O$ |
| gas chromatography – flame ionisation detection – photoionisation detection (GC-FID-PID) | injection | CO, $CH_4$, All VOC compounds, NMVOC, VOC, NMHC, HC, $CH_2O$ | CO, $CH_4$, All VOC compounds, NMVOC, VOC, NMHC, HC, $CH_2O$ |
| gas chromatography – fourier transform infrared spectroscopy – mass spectrometry (GC-FTIR-MS) | injection | CO, $CH_4$, All VOC compounds, NMVOC, VOC, NMHC, HC, $CH_2O$ | CO, $CH_4$, All VOC compounds, NMVOC, VOC, NMHC, HC, $CH_2O$ |
| gas chromatography – cold vapour atomic fluorescence spectroscopy (GC-CV-AFS) | injection | Cd, Hg | Cd, Hg |
| gas chromatography – sulphur chemiluminescence (GC-SC) | low volume continuous | $SO_2$, $H_2S$ | $SO_2$, $H_2S$ |



| Measurement Method | Sampling Type / Sample Preparation | Measured Components | QA Accepted Components |
|---|---|---|---|
| high performance liquid chromatography – unknown detection (HPLC-UNK) | injection | $CH_4$, $CH_2O$, Hg, $CH_4O_3S$, $NH_4^+$, $NH_4NO_3$, Ni, Pb, $SO_4^{2-}$ | $CH_4$, $CH_2O$, Hg, $CH_4O_3S$, $NH_4^+$, $NH_4NO_3$, Ni, Pb, $SO_4^{2-}$ |
| high performance liquid chromatography – mass spectrometry (HPLC-MS) | injection | $CH_2O$, Hg, $CH_4O_3S$, $NH_4^+$, $NH_4NO_3$, Ni, Pb, $SO_4^{2-}$ | $CH_2O$, Hg, $CH_4O_3S$, $NH_4^+$, $NH_4NO_3$, Ni, Pb, $SO_4^{2-}$ |
| high performance liquid chromatography – ultraviolet detection (HPLC-UV) | injection | $CH_4$, $CH_2O$, Hg, $CH_4O_3S$, $NH_4^+$, $NH_4NO_3$, Ni, Pb, $SO_4^{2-}$ | $CH_4$, $CH_2O$, Hg, $CH_4O_3S$, $NH_4^+$, $NH_4NO_3$, Ni, Pb, $SO_4^{2-}$ |
| high performance liquid chromatography – fluorescence detection (HPLC-FLD) | injection | $CH_2O$, Hg, $CH_4O_3S$, $NH_4^+$, $NH_4NO_3$, Ni, Pb, $SO_4^{2-}$ | $CH_2O$, Hg, $CH_4O_3S$, $NH_4^+$, $NH_4NO_3$, Ni, Pb, $SO_4^{2-}$ |
| high performance liquid chromatography – photodiode array detection (HPLC-PDA) | injection | $CH_2O$, Hg, $CH_4O_3S$, $NH_4^+$, $NH_4NO_3$, Ni, Pb, $SO_4^{2-}$ | $CH_2O$, Hg, $CH_4O_3S$, $NH_4^+$, $NH_4NO_3$, Ni, Pb, $SO_4^{2-}$ |
| high performance liquid chromatography – mass spectrometry – fluorescence detection (HPLC-MS-FLD) | injection | $CH_2O$, Hg, $CH_4O_3S$, $NH_4^+$, $NH_4NO_3$, Ni, Pb, $SO_4^{2-}$ | $CH_2O$, Hg, $CH_4O_3S$, $NH_4^+$, $NH_4NO_3$, Ni, Pb, $SO_4^{2-}$ |
| proton transfer reaction – unknown detection (PTR-UNK) | injection | CO, $CH_4$, All VOC compounds, NMVOC, VOC, NMHC, HC, $CH_2O$ | CO, $CH_4$, All VOC compounds, NMVOC, VOC, NMHC, HC, $CH_2O$ |
| proton transfer reaction – mass spectrometry (PTR-MS) | injection | CO, $CH_4$, All VOC compounds, NMVOC, VOC, NMHC, HC, $CH_2O$ | CO, $CH_4$, All VOC compounds, NMVOC, VOC, NMHC, HC, $CH_2O$ |
| colorimetry (CO) | injection | 10+ components | 10+ components |
| spectrophotometry (SP) | injection | 10+ components | 10+ components |
| second derivative spectrophotometry (SDS) | low volume continuous | NO, $NO_2$, $SO_2$, $NH_3$, $HNO_3$, HCl, $H_2S$ | $NH_3$, $HNO_3$, HCl |
| ion chromatography (IC) | injection | 10+ components | 10+ components |
| continuous flow analysis (CFA) | injection / reagent reaction | 10+ components | 10+ components |



| Measurement Method | Sampling Type / Sample Preparation | Measured Components | QA Accepted Components |
|---|---|---|---|
| titration (TI) | injection / reagent reaction | $SO_2$ | ——- |
| aerosol mass spectrometry (AMS) | low volume continuous / filter | $Cl^-$, $NO_3^-$, $NH_4^+$, $SO_4^{2-}$ | $Cl^-$, $NO_3^-$, $NH_4^+$, $SO_4^{2-}$ |
| gravimetry (GR) | manual / filter | PM10, PM2.5, PM1 | PM10, PM2.5, PM1 |
| tapered element oscillating microbalance – gravimetry (TEOM-GR) | low volume continuous / filter | PM10, PM2.5, PM1 | PM10, PM2.5, PM1 |
| tapered element oscillating microbalance – filter dynamics measurement system – gravimetry (TEOM-FDMS-GR) | low volume continuous / filter | PM10, PM2.5, PM1 | PM10, PM2.5, PM1 |
| quartz crystal microbalance – gravimetry (QCM-GR) | low volume continuous / filter | PM10, PM2.5, PM1 | PM10, PM2.5, PM1 |
| pressure drop tape sampling (PDTS) | low volume continuous / filter | PM10, PM2.5, PM1 | PM10, PM2.5, PM1 |
| beta-attenuation (BA) | low volume continuous / filter | PM10, PM2.5, PM1 | PM10, PM2.5, PM1 |
| nephelometry (NP) | low volume continuous / filter | PM10, PM2.5, PM1 | PM10, PM2.5, PM1 |
| nephelometry – laser spectrometry (NP-LS) | low volume continuous / filter | PM10, PM2.5, PM1 | PM10, PM2.5, PM1 |
| light scattering photometry (LSP) | low volume continuous / filter | PM10, PM2.5, PM1 | PM10, PM2.5, PM1 |
| optical particle counter (OPC) | low volume continuous / filter | PM10, PM2.5, PM1 | PM10, PM2.5, PM1 |
| beta-attenuation – nephelometry (BA-NP) | low volume continuous / filter | PM10, PM2.5, PM1 | PM10, PM2.5, PM1 |
| differential mobility particle sizer (DMPS) | low volume continuous / filter | PM10, PM2.5, PM1 | PM10, PM2.5, PM1 |
| scanning mobility particle sizer (SMPS) | low volume continuous / filter | PM10, PM2.5, PM1 | PM10, PM2.5, PM1 |
| thermal analysis (TA) | injection | C, elemental C, organic C | C, elemental C, organic C |
| thermal-optical analysis – unknown protocol (TOA-UNK) | injection | C, elemental C, organic C | C, elemental C, organic C |



| Measurement Method | Sampling Type / Sample Preparation | Measured Components | QA Accepted Components |
|---|---|---|---|
| thermal-optical analysis – EUSAAR2 (TOA-E) | injection | C, elemental C, organic C | C, elemental C, organic C |
| thermal-optical analysis – IMPROVE-A (TOA-I) | injection | C, elemental C, organic C | C, elemental C, organic C |
| thermal-optical analysis – NIOSH 5040 (TOA-N) | injection | C, elemental C, organic C | C, elemental C, organic C |
| aethalometer (ATH) | low volume continuous / filter | black C | black C |
| multi angle absorption photometer (MAAP) | low volume continuous / filter | black C | black C |
| particulate soot absorption photometer (PSAP) | low volume continuous / filter | black C | black C |
| continuous light absorption photometer (CLAP) | low volume continuous / filter | black C | black C |
| flame atomic absorption spectroscopy (F-AAS) | injection | 10+ components | 10+ components |
| graphite furnace atomic absorption spectroscopy (GF-AAS) | injection | 10+ components | 10+ components |
| cold vapour atomic absorption spectroscopy (CV-AAS) | injection | Cd, Hg | Cd, Hg |
| hydride generation atomic absorption spectroscopy (HG-AAS) | injection | As, Pb, Se | As, Pb, Se |
| flame atomic emission spectroscopy (F-AES) | injection | 10+ components | 10+ components |
| inductively coupled plasma atomic emission spectroscopy (ICP-AES) | injection | 10+ components | 10+ components |
| cold vapour atomic fluorescence spectroscopy (CV-AFS) | injection | Cd, Hg | Cd, Hg |
| inductively coupled plasma mass spectrometry (ICP-MS) | injection | 10+ components | 10+ components |
| X-ray fluorescence spectroscopy (XRFS) | injection | 10+ components | 10+ components |
| particle induced X-ray emission (PIXE) | injection | 10+ components | 10+ components |



| Measurement Method | Sampling Type / Sample Preparation | Measured Components | QA Accepted Components |
|---|---|---|---|
| photometry – direct (P-D) | remote | All aod matrix components | All aod matrix components |
| photometry – sky (P-S) | remote | All extaod, absaod, ssa, asy, rin, vconc, size matrix components | All extaod, absaod, ssa, asy, rin, vconc, size matrix components |
| unknown (UNK) | —— | —— | —— |





**Table A8.** Definitions of the standardised network QA flags, set in the "flag" variable. These flags represent a standardised version of all the different QA flags identified across the measurement networks. Whenever a flag is not active, a fill value (255) is set instead.

| Flag Code | Flag Name | Flag Code | Flag Name | Flag Code | Flag Name | Flag Code | Flag Name |
|---|---|---|---|---|---|---|---|
| colspan="8" align="center" | **Basic Flags** |||||||
| 0 | Valid Data | 1 | Preliminary Data | 2 | Missing Data | 3 | Invalid Data – Unspecified |
| 4 | Un-Flagged Data | | | | | | |
| colspan="8" align="center" | **Estimated Flags** |||||||
| 10 | Estimated Data – Unspecified | 11 | Estimated Data – Measured Negative Value | 12 | Estimated Data – No Value Detected | 13 | Estimated Data – Value Below Detection Limit |
| 14 | Estimated Data – Value Above Detection Limit | 15 | Estimated Data – Value Substituted from Secondary Monitor | 16 | Estimated Data – Multiple Parameters Aggregated | | |
| colspan="8" align="center" | **Extreme / Irregular Flags** |||||||
| 20 | Extreme / Irregular Data – Unspecified | 21 | Data Does Not Meet Internal Network Quality Control Criteria | 22 | High Variability of Data | 23 | Irregular Data Manually Screened and Accepted |
| 24 | Irregular Data Manually Screened and Rejected | 25 | Negative Value | 26 | No Value Detected | 27 | Reconstructed / Recalculated Data |
| 28 | Value Close to Detection Limit | 29 | Value Below Acceptable Range | 30 | Value Above Acceptable Range | 31 | Value Below Detection Limit |
| 32 | Value Above Detection Limit | | | | | | |
| colspan="8" align="center" | **Measurement Issue Flags** |||||||
| 40 | Measurement Issue – Unspecified | 41 | Chemical Issue | 42 | Erroneous Sampling Operation | 43 | Extreme Internal Instrument Meteorological Conditions |

*table continued on next page*

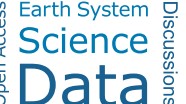

| Flag Code | Flag Name | Flag Code | Flag Name | Flag Code | Flag Name | Flag Code | Flag Name |
|---|---|---|---|---|---|---|---|
| 44 | Extreme Ambient Laboratory Meteorological Conditions | 45 | Extreme External Meteorological Conditions | 46 | Extreme Sample Transport Conditions | 47 | Invalid Flow Rate |
| 48 | Human Error | 49 | Matrix Effect | 50 | Mechanical Issue / Non-Operational Equipment | 51 | No Technician |
| 52 | Operational Maintenance Check Issue | 53 | Physical Issue With Filter | 54 | Power Failure | 55 | Sample Diluted for Analysis |
| 56 | Unmeasured Key Meteorological Parameter | 57 | Sample Not Analysed | | | | |

**Operational Maintenance Flags**

| Flag Code | Flag Name | Flag Code | Flag Name | Flag Code | Flag Name | Flag Code | Flag Name |
|---|---|---|---|---|---|---|---|
| 60 | Operational Maintenance – Unspecified | 61 | Calibration | 62 | Accuracy Check | 63 | Blank Check |
| 64 | Detection Limits Check | 65 | Precision Check | 66 | Retention Time Check | 67 | Span Check |
| 68 | Zero Check | 69 | Instrumental Inspection | 70 | Instrumental Repair | 71 | Quality Control Audit |

**Data Formatting Issue Flags**

| Flag Code | Flag Name | Flag Code | Flag Name | Flag Code | Flag Name | Flag Code | Flag Name |
|---|---|---|---|---|---|---|---|
| 80 | Data Formatting / Processing Issue | 81 | Corrected Data Formatting / Processing Issue | | | | |

**Representativity Flags**

| Flag Code | Flag Name | Flag Code | Flag Name | Flag Code | Flag Name | Flag Code | Flag Name |
|---|---|---|---|---|---|---|---|
| 90 | Aggregation / Representation Issue – Unspecified | 91 | Data Window Completeness < 90% | 92 | Data Window Completeness < 75% | 93 | Data Window Completeness < 66% |
| 94 | Data Window Completeness < 50% | 95 | Data Window Completeness < 25% | 96 | >= 75% of Measurements in Window Below Detection Limit | 97 | >= 50% of Measurements in Window Below Detection Limit |





| Flag Code | Flag Name | Flag Code | Flag Name | Flag Code | Flag Name | Flag Code | Flag Name |
|---|---|---|---|---|---|---|---|
| **Weather Flags** | | | | | | | |
| 100 | No Significant Weather | 101 | Precipitation – Unspecified Intensity | 102 | Precipitation – Light | 103 | Precipitation – Moderate |
| 104 | Precipitation – Heavy | 105 | Drizzle – Unspecified Intensity | 106 | Drizzle – Light | 107 | Drizzle – Moderate |
| 108 | Drizzle – Heavy | 109 | Freezing Drizzle – Unspecified Intensity | 110 | Freezing Drizzle – Light | 111 | Freezing Drizzle – Moderate |
| 112 | Freezing Drizzle – Heavy | 113 | Rain – Unspecified Intensity | 114 | Rain – Light | 115 | Rain – Moderate |
| 116 | Rain – Heavy | 117 | Rain Shower/s – Unspecified Intensity | 118 | Rain Shower/s – Light | 119 | Rain Shower/s – Moderate |
| 120 | Rain Shower/s – Heavy | 121 | Freezing Rain – Unspecified Intensity | 122 | Freezing Rain – Light | 123 | Freezing Rain – Moderate |
| 124 | Freezing Rain – Heavy | 125 | Freezing Rain Shower/s – Unspecified Intensity | 126 | Freezing Rain Shower/s – Light | 127 | Freezing Rain Shower/s – Moderate |
| 128 | Freezing Rain Shower/s – Heavy | 129 | Snow – Unspecified Intensity | 130 | Snow – Light | 131 | Snow – Moderate |
| 132 | Snow – Heavy | 133 | Snow Shower/s – Unspecified Intensity | 134 | Snow Shower/s – Light | 135 | Snow Shower/s – Moderate |
| 136 | Snow Shower/s – Heavy | 137 | Hail – Unspecified Intensity | 138 | Hail – Light | 139 | Hail – Moderate |
| 140 | Hail – Heavy | 141 | Hail Shower/s – Unspecified Intensity | 142 | Hail Shower/s – Light | 143 | Hail Shower/s – Moderate |
| 144 | Hail Shower/s – Heavy | 145 | Ice Pellets – Unspecified Intensity | 146 | Ice Pellets – Light | 147 | Ice Pellets – Moderate |
| 148 | Ice Pellets – Heavy | 149 | Ice Pellets Shower/s – Unspecified Intensity | 150 | Ice Pellets Shower/s – Light | 151 | Ice Pellets Shower/s – Moderate |
| 152 | Ice Pellets Shower/s – Heavy | 153 | Snow Pellets – Unspecified Intensity | 154 | Snow Pellets – Light | 155 | Snow Pellets – Moderate |
| 156 | Snow Pellets – Heavy | 157 | Snow Pellets Shower/s – Unspecified Intensity | 158 | Snow Pellets Shower/s – Light | 159 | Snow Pellets Shower/s – Moderate |





| Flag Code | Flag Name | Flag Code | Flag Name | Flag Code | Flag Name | Flag Code | Flag Name |
|---|---|---|---|---|---|---|---|
| 160 | Snow Pellets Shower/s – Heavy | 161 | Snow Grains – Unspecified Intensity | 162 | Snow Grains – Light | 163 | Snow Grains – Moderate |
| 164 | Snow Grains – Heavy | 165 | Diamond Dust – Unspecified Intensity | 166 | Diamond Dust – Light | 167 | Diamond Dust – Moderate |
| 168 | Diamond Dust – Heavy | 169 | Glaze | 170 | Rime | 171 | Thunderstorm |
| 172 | Funnel Cloud/s | 173 | Squalls | 174 | Tropical Cyclone (Cyclone / Hurricane / Typhoon) | 175 | Duststorm |
| 176 | Sandstorm | 177 | Dust/Sand Whirls | 178 | High Winds | | |
| **Local Contamination Flags** | | | | | | | |
| 180 | No Atmospheric Obscuration | 181 | Atmospheric Obscuration – Unknown | 182 | Dust | 183 | Blowing Dust |
| 184 | Drifting Dust | 185 | Sand | 186 | Blowing Sand | 187 | Drifting Sand |
| 188 | Blowing Snow | 189 | Drifting Snow | 190 | Fog | 191 | Freezing Fog |
| 192 | Ground Fog | 193 | Ice Fog | 194 | Haze | 195 | Mist |
| 196 | Sea Spray | 197 | Smoke | 198 | Volcanic Ash | 199 | No Local Contamination |
| 200 | Local Contamination – Unspecified | 201 | Agricultural Contamination | 202 | Bird-Dropping Contamination | 203 | Construction Contamination |
| 204 | Industrial Contamination | 205 | Insect Contamination | 206 | Internal Laboratory / Instrument Contamination | 207 | Pollen / Leaf Contamination |
| 208 | Traffic Contamination | | | | | | |
| **Exceptional Event Flags** | | | | | | | |
| 210 | Exceptional Event – Unspecified | 211 | Seismic Activity | 212 | Stratospheric Ozone Intrusion | 213 | Volcanic Eruptions |
| 214 | Wildfire | 220 | Chemical Spill / Industrial Accident | 221 | Cleanup After a Major Disaster | 222 | Demolition |

*table continued on next page*





| Flag Code | Flag Name | Flag Code | Flag Name | Flag Code | Flag Name | Flag Code | Flag Name |
|-----------|-----------|-----------|-----------|-----------|-----------|-----------|-----------|
| 223 | Fireworks | 224 | Infrequent Large Gathering | 225 | Terrorist Act | | |
| **Meteorological Infinite Flags** | | | | | | | |
| 230 | Visibility Distance Unlimited | 231 | Ceiling Height Unlimited | | | | |





**Table A9.** Definitions of GHOST QA flags, set in the "qa" variable, each derived from GHOST own quality control checks. Whenever a flag
is not active, a fill value (255) is set instead.

| QA Flag | QA Name | Description |
|---|---|---|
| **Basic Flags** | | |
| 0 | Missing Measurement | Measurement is missing (i.e. NaN), or have network QA flag stating missing measurement. |
| 1 | Infinite Value | Measurement is infinite. This happens when values are outside of the range that the float32 data type can handle (-3.4E+38 to +3.4E+38). |
| 2 | Negative Measurement | Measurement is negative (i.e. < 0.0), or have network QA flag stating negative measurement. |
| 3 | Zero Measurement | Measurement is zero, or have network QA flag stating no value detected. |
| 4 | Not Maximum Data Quality Level | Measurement is not of the highest data quality level available from data provider. |
| 5 | Preliminary Data | Measurement which is flagged in the network QA as preliminary. |
| 6 | Invalid Data Provider Flags – GHOST Decreed | Measurement is associated with network QA flag/s which have been decreed by the GHOST project architects to suggest the measurements are associated with substantial uncertainty / bias. |
| 7 | Invalid Data Provider Flags – Network Decreed | Measurement is associated with network QA flag/s which have been decreed by the reporting network to suggest the measurements are associated with substantial uncertainty / bias. |
| 8 | No Valid Data to Average | After screening by GHOST QA, no valid data remains to perform temporal average. |
| **Measurement Process Flags** | | |
| 10 | Methodology Not Mapped | The reported measurement methodology has not been able to be mapped to a standard methodology name. |
| 11 | Assumed Primary Sampling | A level of assumption has been made in determining the primary sampling type. |
| 12 | Assumed Sample Preparation | A level of assumption has been made in determining the sample preparation. |
| 13 | Assumed Measurement Methodology | A level of assumption has been made in determining the measurement methodology. |
| 14 | Unknown Primary Sampling Type | The specific name of the primary sampling type is unknown. |
| 15 | Unknown Primary Sampling Instrument | The specific name of the primary sampling instrument is unknown. |



| QA Flag | QA Name | Description |
|---|---|---|
| 16 | Unknown Sample Preparation Type | The specific name of the sample preparation type is unknown. |
| 17 | Unknown Sample Preparation Technique | The specific name of the sample preparation technique is unknown. |
| 18 | Unknown Measurement Method | The specific name of the measurement method is unknown. |
| 19 | Unknown Measuring Instrument | The specific name of measuring instrument is unknown. |
| 20 | Erroneous Primary Sampling | The primary sampling used is not appropriate to prepare the specific component for subsequent measurement. |
| 21 | Erroneous Sample Preparation | The sample preparation used is not appropriate to prepare the specific component for subsequent measurement. |
| 22 | Erroneous Measurement Methodology | The measurement methodology used is not known to be able to measure the specific component. |
| 23 | Invalid QA Measurement Methodology | The measurement methodology used has been decreed not to conform to minimum GHOST QA standards. |
| 24 | Corrected Parameter | Measurement has been corrected, or is of significantly higher quality than other types of measurements. |
| **Sample Gas Volume Flags** | | |
| 30 | Sample Gas Volume – Network Standard | The sample gas volume is assumed, using a known network standard temperature and pressure. |
| 31 | Sample Gas Volume – Unknown | The sample gas volume is unknown. |
| 32 | Unit Conversion – Network Standard Sample Gas Volume Assumption | Unit conversion has been done assuming the sample gas volume, using a known network standard temperature and pressure. |
| 33 | Unit Conversion – Educated Guess Sample Gas Volume Assumption | Unit conversion has been done making an educated guess at the temperature and pressure of the sample gas. |
| **Positional Metadata Doubt Flags** | | |
| 40 | Station Position Doubt – DEM Decreed | The validity of the reported station position is found to be in doubt, with the reported station altitude, differing by more than 50m in absolute terms from the ASTER v3 DEM altitude. |
| 41 | Station Position Doubt – Manually Decreed | There exists significant doubt about the accuracy of the station position, determined from empirical / word of mouth evidence. |
| **Data Product Flags** | | |
| 45 | Data Product | Data is a product that has been calculated from multiple components. |
| 46 | Insufficient Data to Calculate Data Product | There is insufficient valid data required to calculate data product. |

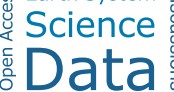

| QA Flag | QA Name | Description |
|---|---|---|
| **Local Condition Flags** | | |
| 50 | Local Precipitation | Network QA flag/s suggesting precipitation at the time of measurement. |
| 51 | Local Extreme Weather | Network QA flag/s suggesting extreme weather at the time of measurement. |
| 52 | Local Atmospheric Obscuration | Network QA flag/s suggesting atmospheric obscuration at the time of measurement. |
| 53 | Local Contamination | Network QA flag/s suggesting local contamination at the time of measurement. |
| 54 | Local Exceptional Event | Network QA flag/s suggesting exceptional event (either natural or anthropogenic) at the time of measurement. |
| **Timezone Flags** | | |
| 60 | Non-Integer Local Timezone (relative to UTC) | Determine if the local timezone of measurement station is non-integer, relative to UTC. |
| 61 | Timezone Doubt | Significant doubt exists regarding the local timezone of the reported data. |
| **Limit of Detection Flags** | | |
| 70 | Below Documented Lower Limit of Detection | Measurement is below or equal to the instrumental documented lower limit of detection. |
| 71 | Below Reported Lower Limit of Detection | Measurement is below or equal to the network reported lower limit of detection. |
| 72 | Below Preferential Lower Limit of Detection | Measurement is below or equal to the preferential lower limit of detection. This is the network reported limit if available, else it is the instrumental documented limit. |
| 73 | Above Documented Upper Limit of Detection | Measurement is above or equal to the instrumental documented upper limit of detection. |
| 74 | Above Reported Upper Limit of Detection | Measurement is above or equal to the network reported upper limit of detection. |
| 75 | Above Preferential Upper Limit of Detection | Measurement is above or equal to the preferential upper limit of detection. This is the network reported limit if available, else it is the instrumental documented limit. |
| **Measurement Resolution Flags** | | |
| 80 | Insufficient Measurement Resolution – Documented | The instrumental documented resolution of measurement is coarser than a set limit. |

*table continued on next page*





| QA Flag | QA Name | Description |
|---|---|---|
| 81 | Insufficient Measurement Resolution – Reported | The network reported resolution of measurement is coarser than a set limit. |
| 82 | Insufficient Measurement Resolution – Preferential | The preferential resolution of measurement is coarser than a set limit. This is the network reported resolution if available, else it is the instrumental documented resolution. |
| 83 | Insufficient Measurement Resolution – Empirical | The minimum difference between all measurements in a month is coarser than a set limit. Measurements are pre-screened by other GHOST QA (see Table A14). |
| **Recurring Value Flags** | | |
| 90 | Persistent Recurring Values – 5/6 | Persistently recurring values are symptomatic of when an instrument hits the detection limit, or is malfunctioning. If 5/6, 9/12 or 16/24 of consecutive values are non-NaN, and the same value, the whole series of consecutive values are flagged. |
| 91 | Persistent Recurring Values – 9/12 | |
| 92 | Persistent Recurring Values – 16/24 | |
| **Monthly Fractional Unique Value Flags** | | |
| 100 | Monthly Fractional Unique Values <= 1% | Monthly data with a low % of unique values is symptomatic of when an instrument hits the detection limit, or is malfunctioning. If the % of unique data in a month is less than a given %, then the entire month is flagged. Measurements are pre-screened by other GHOST QA (see Table A14). |
| 101 | Monthly Fractional Unique Values <= 5% | |
| 102 | Monthly Fractional Unique Values <= 10% | |
| 103 | Monthly Fractional Unique Values <= 30% | |
| 104 | Monthly Fractional Unique Values <= 50% | |
| 105 | Monthly Fractional Unique Values <= 70% | |
| 106 | Monthly Fractional Unique Values <= 90% | |
| **Data Outlier Flags** | | |
| 110 | Data Outlier – Exceeds Scientifically Decreed Lower / Upper Limit | Measurement exceeds scientifically decreed lower / upper bounds. |
| 111 | Data Outlier – Monthly Median Exceeds Scientifically Decreed Upper Limit | Monthly median is greater than a scientifically decreed upper limit. Measurements are pre-screened by other GHOST QA (see Table A14). |
| 112 | Data Outlier – Network Decreed | Network QA flag/s suggest measurement is outlying. |
| 113 | Data Outlier – Manually Decreed | Measurement has been manually found to be outlying. |
| 114 | Possible Data Outlier – Monthly Adjusted Boxplot | Measurement exceeds monthly adjusted boxplot inner fence (lower or upper). This is explained in more detail in Sect. 3.5.1. Measurements are pre-screened by other GHOST QA (see Table A14). |





| QA Flag | QA Name | Description |
|---|---|---|
| 115 | Probable Data Outlier – Monthly Adjusted Boxplot | Measurement exceeds monthly adjusted boxplot outer fence (lower or upper). This is explained in more detail in Sect. 3.5.1. Measurements are pre-screened by other GHOST QA (see Table A14). |
| **Monthly Distribution Consistency Flags** | | |
| 120 | Monthly Distribution Consistency – Zone 1 | Flags which indicate how consistent a monthly distribution of measurements is with other distributions for the same month, across the years. Zone 1 is when the distribution is extremely consistent, and Zone 10 is when the distribution is extremely atypical. This is explained in more detail in Sect. 3.5.2. Measurements are pre-screened by other GHOST QA (see Table A14). |
| 121 | Monthly Distribution Consistency – Zone 2 | |
| 122 | Monthly Distribution Consistency – Zone 3 | |
| 123 | Monthly Distribution Consistency – Zone 4 | |
| 124 | Monthly Distribution Consistency – Zone 5 | |
| 125 | Monthly Distribution Consistency – Zone 6 | |
| 126 | Monthly Distribution Consistency – Zone 7 | |
| 127 | Monthly Distribution Consistency – Zone 8 | |
| 128 | Monthly Distribution Consistency – Zone 9 | |
| 129 | Monthly Distribution Consistency – Zone 10 | |
| 130 | Monthly Distribution Consistency – Unclassified | |
| 131 | Systematic Inconsistent Monthly Distributions – 2/3 Months >= Zone 6 | |
| 132 | Systematic Inconsistent Monthly Distributions – 4/6 Months >= Zone 6 | |
| 133 | Systematic Inconsistent Monthly Distributions – 8/12 Months >= Zone 6 | |





**Table A10.** Definition of the default GHOST QA flags, used to prefilter data to create the "*GHOSTcomponentname*_prefiltered_defaultqa" data variable. The QA flag code and name are both stated.

| QA Flag | QA Name |
|---------|---------|
| 0 | Missing Measurement |
| 1 | Infinite Value |
| 2 | Negative Measurement |
| 6 | Invalid Data Provider Flags – GHOST Decreed |
| 8 | No Valid Data to Average |
| 20 | Erroneous Primary Sampling |
| 21 | Erroneous Sample Preparation |
| 22 | Erroneous Measurement Methodology |
| 72 | Below Preferential Lower Limit of Detection |
| 75 | Above Preferential Upper Limit of Detection |
| 82 | Insufficient Measurement Resolution – Preferential |
| 83 | Insufficient Measurement Resolution – Empirical |
| 110 | Data Outlier – Exceeds Scientifically Decreed Lower / Upper Limit |
| 111 | Data Outlier – Monthly Median Exceeds Scientifically Decreed Upper Limit |
| 112 | Data Outlier – Network Decreed |
| 113 | Data Outlier – Manually Decreed |
| 115 | Probable Data Outlier – Monthly Adjusted Boxplot |
| 132 | Systematic Inconsistent Monthly Distributions – 4/6 Months >= Zone 6 |
| 133 | Systematic Inconsistent Monthly Distributions – 8/12 Months >= Zone 6 |





**Table A11.** Description of the gridded metadata which are ingested in GHOST. This is an expanded version of Table 9, giving for each metadata type the temporal and spatial extent, the ellipsoid / projection, the horizontal / vertical datum, the native horizontal resolution, and native file format.

| Metadata Name | Temporal Extent | Spatial Extent | Ellipsoid / Projection | Horizontal / Vertical Datum | Native Resolution | Native File Format |
|---|---|---|---|---|---|---|
| ASTER v3 altitude (NASA et al., 2018) | 2000 – 2014 | -180:180°E -83:83°N | WGS 84 / ——- | World Geodetic System 1984 / EGM96 | 1" | netCDF4 |
| ETOPO1 altitude (NOAA NGDC, 2009) | 1940 – 2008 | -180:180°E -90:90°N | WGS 84 / ——- | World Geodetic System 1984 / tidal – mean sea level | 1' | netCDF3 |
| EDGAR v4.3.2 annual average emissions (Crippa et al., 2018; EC JRC and Netherlands PBL) | 1970, 1975, 1980, 1985, 1990, 1995, 2000, 2005, 2010, 2012 | -180:180°E -90:90°N | WGS 84 / ——- | World Geodetic System 1984 / ——- | 6' | netCDF3 |
| ESDAC Iwahashi landform classification (Iwahashi and Pike, 2007; ESDAC) | 2007 | -180:180°E -60:90°N | WGS 84 / ——- | World Geodetic System 1984 / ——- | 30" | TIF |
| ESDAC Meybeck landform classification (Meybeck et al., 2001; ESDAC) | 2001 | -180:180°E -56:61°N | WGS 84 / ——- | World Geodetic System 1984 / ——- | 30" | TIF |
| GPW population density, v3: CIESIN and CIAT (2005), v4: CIESIN (2018) | v3: 1990, 1995 v4: 2000, 2005, 2010, 2015 | v3: -180:180°E -58:85°N v4: -180:180°E -90:90°N | WGS 84 / ——- | World Geodetic System 1984 / ——- | v3: 2.5' v4: 30" | TIF |
| GHSL built up area density Corbane et al. (2018, 2019) | 1975, 1990, 2000, 2014 | -180:180°E -90:90°N | WGS 84 / World Mollweide | World Geodetic System 1984 / ——- | 250m | TIF |
| GHSL population density Freire et al. (2016); Schiavina et al. (2019) | 1975, 1990, 2000, 2015 | -180:180°E -90:90°N | WGS 84 / World Mollweide | World Geodetic System 1984 / ——- | 250m | TIF |



| Metadata Name | Temporal Extent | Spatial Extent | Ellipsoid / Projection | Horizontal / Vertical Datum | Native Resolution | Native File Format |
|---|---|---|---|---|---|---|
| GHSL settlement model classification Ehrlich et al. (2019); Pesaresi et al. (2019) | 1975, 1990, 2000, 2015 | -180:180°E -90:90°N | WGS 84 / World Mollweide | World Geodetic System 1984 / ——- | 1km | TIF |
| GSFC coastline proximity (NASA OBPG) | 2009 | -180:180°E -90:90°N | WGS 84 / ——- | World Geodetic System 1984 / ——- | 36" | TIF |
| Koppen-Geiger classification (Beck et al., 2018) | 1980 – 2016 | -180:180°E -90:90°N | WGS 84 / ——- | World Geodetic System 1984 / ——- | 30" | TIF |
| MODIS MCD12C1 v6 IGBP land use (Friedl and Sulla-Menashe, 2015) | 2001, 2005, 2010, 2015, 2018 | -180:180°E -90:90°N | WGS 84 / ——- | World Geodetic System 1984 / ——- | 3' | HDF4 |
| MODIS MCD12C1 v6 UMD land use (Friedl and Sulla-Menashe, 2015) | 2001, 2005, 2010, 2015, 2018 | -180:180°E -90:90°N | WGS 84 / ——- | World Geodetic System 1984 / ——- | 3' | HDF4 |
| MODIS MCD12C1 v6 LAI (Friedl and Sulla-Menashe, 2015) | 2001, 2005, 2010, 2015, 2018 | -180:180°E -90:90°N | WGS 84 / ——- | World Geodetic System 1984 / ——- | 3' | HDF4 |
| NOAA-DMSP-OLS v4 nighttime stable lights (NOAA and US Air Force Weather Agency) | 1992, 1995, 2000, 2005, 2010, 2013 | -180:180°E -65:75°N | WGS 84 / ——- | World Geodetic System 1984 / ——- | 30" | TIF |
| OMI level3 column annual average NO2 (Krotkov et al., 2017, 2019) | 2005, 2010, 2015, 2018 | -180:180°E -90:90°N | WGS 84 / ——- | World Geodetic System 1984 / ——- | 15' | HDF5 |
| OMI level3 column cloud screened annual average NO2 (Krotkov et al., 2017, 2019) | 2005, 2010, 2015, 2018 | -180:180°E -90:90°N | WGS 84 / ——- | World Geodetic System 1984 / ——- | 15' | HDF5 |



| Metadata Name | Temporal Extent | Spatial Extent | Ellipsoid / Projection | Horizontal / Vertical Datum | Native Resolution | Native File Format |
|---|---|---|---|---|---|---|
| OMI level3 tropospheric column annual average NO2 (Krotkov et al., 2017, 2019) | 2005, 2010, 2015, 2018 | -180:180°E -90:90°N | WGS 84 / ——- | World Geodetic System 1984 / ——- | 15' | HDF5 |
| OMI level3 tropospheric column cloud screened annual average NO2 (Krotkov et al., 2017, 2019) | 2005, 2010, 2015, 2018 | -180:180°E -90:90°N | WGS 84 / ——- | World Geodetic System 1984 / ——- | 15' | HDF5 |
| WMO region (WMO, a) | 2013 | -180:180°E -90:90°N | WGS 84 / ——- | World Geodetic System 1984 / ——- | ——- | GeoJSON |
| WWF TEOW terrestrial ecoregion (Olson et al., 2001) | 2006 | -180:180°E -90:83.623°N | WGS 84 / ——- | World Geodetic System 1984 / ——- | ——- | Shapefile |
| WWF TEOW biogeographical realm (Olson et al., 2001) | 2006 | -180:180°E -90:83.623°N | WGS 84 / ——- | World Geodetic System 1984 / ——- | ——- | Shapefile |
| WWF TEOW biome (Olson et al., 2001) | 2006 | -180:180°E -90:83.623°N | WGS 84 / ——- | World Geodetic System 1984 / ——- | ——- | Shapefile |
| UMBC anthrome classification (Ellis et al., 2010; University of Maryland Baltimore County) | 2000 | -180:180°E -90:90°N | WGS 84 / ——- | World Geodetic System 1984 / ——- | 5' | netCDF3 |





**Table A12.** Outline of the key metadata variables (grouped per type) used for the assessment of duplicate metadata columns, in Stage 1 of the GHOST pipeline (standardisation). A metadata column is identified as being "duplicate" if none of the key variables do not change from the previous column.

| Metadata Group Type | Metadata Variables |
|---|---|
| station information | longitude, latitude, altitude, sampling_height, measurement_altitude, distance_to_building, distance_to_kerb, distance_to_junction, distance_to_source, street_width, street_type, daytime_traffic_speed, daily_passing_vehicles, ellipsoid, horizontal_datum, vertical_datum, projection, data_level, climatology, station_name, city, country, population, representative_radius, associated_networks |
| station classifications | area_classification, station_classification, main_emission_source, land_use, terrain, measurement_scale |
| measurement information | primary_sampling_type, primary_sampling_instrument_name, primary_sampling_instrument_reported_flow_rate, sample_preparation_types, sample_preparation_techniques, measurement_methodology, measuring_instrument_name, measuring_instrument_sampling_type, measuring_instrument_reported_flow_rate, measuring_instrument_reported_lower_limit_of_detection, measuring_instrument_reported_upper_limit_of_detection, measuring_instrument_reported_uncertainty, measuring_instrument_reported_accuracy, measuring_instrument_reported_precision, measuring_instrument_reported_measurement_resolution, measuring_instrument_reported_absorption_cross_section, measuring_instrument_calibration_scale, network_provided_volume_standard_temperature, network_provided_volume_standard_pressure |



**Table A13.** Definitions of the dependencies for the temporal filling of metadata variables, in Stage 2 of the GHOST pipeline (station data concatenation), to prevent incompatibilities in concurrent metadata variables. This essentially means for all metadata variables in a group, that each variable can only be filled temporally (going either forwards or backwards in time), if none of the dependent variables have changed between metadata columns. Because of the importance of positional variables being set (e.g. latitude), filling is attempted to be done through several passes, using progressively less stringent dependencies, until ultimately requiring zero dependencies. The "non-filled" group outlines variables that filling is not performed for due to being highly time sensitive.

| Metadata Group Type | Dependent Variables | Metadata Variables |
| --- | --- | --- |
| longitude | 1. latitude<br>2. non-dependent | longitude |
| latitude | 1. longitude<br>2. non-dependent | latitude |
| altitude | 1. longitude, latitude, measurement_altitude<br>2. longitude, latitude, sampling_height<br>3. longitude, latitude<br>4. non-dependent | altitude |
| sampling height | 1. longitude, latitude, measurement_altitude<br>2. longitude, latitude, altitude<br>3. longitude, latitude<br>4. non-dependent | sampling_height |
| measurement altitude | 1. longitude, latitude, altitude<br>2. longitude, latitude, sampling_height<br>3. longitude, latitude<br>4. non-dependent | measurement_altitude |



| Metadata Group Type | Dependent Variables | Metadata Variables |
| --- | --- | --- |
| position dependent | longitude, latitude | area_classification, station_classification, main_emission_source, land_use, terrain, measurement_scale, representative_radius, distance_to_building, distance_to_kerb, distance_to_junction, distance_to_source, street_width, street_type, ellipsoid, horizontal_datum, vertical_datum, projection, climatology, station_name, city, country, associated_networks |
| primary sampling type dependent | primary_sampling_type | primary_sampling_instrument_name |
| primary sampling instrument dependent | primary_sampling_instrument_name | primary_sampling_instrument_documented_flow_rate, primary_sampling_instrument_reported_flow_rate, primary_sampling_instrument_manual_name |
| sample preparation type dependent | sample_preparation_types | sample_preparation_techniques |
| measurement methodology dependent | measurement_methodology | measuring_instrument_name |
| measuring instrument dependent | measuring_instrument_name | measuring_instrument_documented_flow_rate, measuring_instrument_reported_flow_rate, measuring_instrument_manual_name, measuring_instrument_reported_units, measuring_instrument_reported_lower_limit_of_detection, measuring_instrument_documented_lower_limit_of_detection, measuring_instrument_reported_upper_limit_of_detection, measuring_instrument_documented_upper_limit_of_detection, measuring_instrument_reported_uncertainty, measuring_instrument_documented_uncertainty, measuring_instrument_reported_accuracy, measuring_instrument_documented_accuracy, measuring_instrument_reported_precision, measuring_instrument_documented_precision, measuring_instrument_reported_zero_drift, measuring_instrument_documented_zero_drift, measuring_instrument_reported_span_drift, measuring_instrument_documented_span_drift, measuring_instrument_reported_zonal_drift, measuring_instrument_documented_zonal_drift, measuring_instrument_reported_measurement_resolution, measuring_instrument_documented_measurement_resolution, measuring_instrument_reported_absorption_cross_section, measuring_instrument_documented_absorption_cross_section |

*table continued on next page*



| Metadata Group Type | Dependent Variables | Metadata Variables |
| :---: | :---: | :--- |
| non-filled | —————————— | daytime_traffic_speed, daytime_passing_vehicles, population |





**Table A14.** Outline of all GHOST QA checks, in Stage 4 of the GHOST pipeline (quality assurance), which pre-screen data by other GHOST QA before calculation.

| QA Check | Pre-screen QA Flag Codes |
|---|---|
| Empirical measurement resolution (code 83) | 0, 1, 6, 72, 75, 110, 112, 113 |
| Unique values (codes 100 – 106) | 0, 1, 6, 72, 75, 110, 112, 113 |
| Non-feasible monthly median (code 111) | 0, 1, 6, 72, 75, 110, 112, 113 |
| Monthly adjusted boxplot (codes 114 and 115) | 0, 1, 6, 72, 75, 110, 112, 113 |
| Monthly distribution consistency (codes 120 – 133) | 0, 1, 6, 20, 21, 72, 75, 100, 110, 112, 113 |



**Table A15.** Outline of the different GHOST QA flag groupings in Stage 6 of the GHOST pipeline (temporal averaging), detailing how GHOST QA flags are treated whenever measurements are averaged in a window. When averaging measurements some GHOST QA flags are applied to screen invalid data, whereas the rest of the flags are only retained if they appear more than not across the window.

| Flag Grouping | Description | QA Flag Codes |
|---|---|---|
| Invalid QA | Flags are applied to screen data, ensuring the subsequent temporal average is sensible. | 0, 1, 6, 46, 72, 75, 110, 112, 113 |
| Modal QA | Flags for which a modal determination is performed i.e. if each flag appears more than not across the associated measurements, they are kept for the averaged period, otherwise they are dropped. | 2, 3, 4, 5, 7, 10, 11, 12, 13, 14, 15, 16, 17, 18, 19, 20, 21, 22, 23, 24, 30, 31, 32, 33, 40, 41, 45, 50, 51, 52, 53, 54, 60, 61, 70, 71, 73, 74, 80, 81, 82, 83, 90, 91, 92, 100, 101, 102, 103, 104, 105, 106, 111, 114, 115, 120, 121, 122, 123, 124, 125, 126, 127, 128, 129, 130, 131, 132, 133 |





*Author contributions.* DB is the sole developer of the project, and drafted the paper. OJ and CPGP helped design the framework of the paper, and acquired funding. SB and MG helped resolve data rights issues. JK helped make the link with WIGOS. All authors contributed to revising the paper.

*Competing interests.* The authors declare no competing interests.

*Acknowledgements.* The authors gratefully acknowledge all data providers for the substantial work done in establishing and maintaining the measuring stations that provide the data used in this work. We would also like to warmly thank all data providers who met with GHOST authors through this work, and for all support given, from helping resolve data rights issues, to giving suggestions for improvements.

The authors would also like to acknowledge C. Lund Myhre and M. Fiebig for the fruitful discussions during the preparation of the manuscript.

BSC co-authors acknowledge the computing resources of MareNostrum, and the technical support provided by the Barcelona Supercomputing Center (AECT-2020-1-0007, AECT-2021-1-0027, AECT-2022-1-0008, and AECT-2022-3-0013). We also acknowledge the Red Temática ACTRIS España (CGL2017-90884-REDT), and the H2020 project ACTRIS IMP (#871115).

The research leading to these results has received funding from the grant RTI2018-099894-BI00 funded by MCIN/AEI/ 10.13039/501100011033 (BROWNING), the EU H2020 Framework Programme under grant agreement nº GA 821205

(FORCES), the European Research Council under the Horizon 2020 research and innovation programme through the ERC Consolidator Grant grant agreement No. 773051 (FRAGMENT), the AXA Research Fund (AXA Chair on Sand and Dust Storms at the Barcelona Supercomputing Center), and the Department of Research and Universities of the Government of Catalonia through the Atmospheric Composition Research Group (code 2021 SGR 01550).





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
