# Peer review of "GHOST: A globally harmonised dataset of surface atmospheric composition measurements"

_Earth System Science Data, 2023_

## Author Comment (AC1)

We greatly thank the reviewer for their feedback on our manuscript. Please find our response here to their comments.

Dene Bowdalo

**Reviewer 1**

The GHOST harmonised dataset is a valuable contribution to the surface atmospheric composition measurement community in that it brings together key data from 38 observing networks in a standardized way. The dataset has the potential to streamline data workflows for the atmospheric composition measurement community and also opens the possibility of conducting longer scale spatial and temporal analyses. I rated the uniqueness of this manuscript as 'good' due to the fact that this dataset could be replicated if needed. However, the cost and effort required to replicate the dataset would be high, making the open sharing of this data important. The usefulness of this manuscript and the corresponding dataset is excellent. I especially appreciated the thorough definitions of the variables and metadata included in the appendix. As a researcher, I wish that more datasets had such clear and thorough documentation on variables and metadata. The manuscript and dataset is complete and also includes access to the code.

**1.1.** The presentation quality should be improved as the manuscript is lengthy and long articles are not expected for ESSD. More specifically, section 3 (GHOST processing workflow) should be streamlined and made more concise in order to improve the presentation quality. With some revisions, especially to section 3, this manuscript will be a nice contribution to the Earth science community.

> With regards to the length of the manuscript and section 3 in general, we acknowledge that this is a longer than typical manuscript for ESSD, however we feel that the detail is necessary for a few key reasons. Firstly, this is one of the largest harmonisation efforts of this kind that we know of, and thus for the sake of reproducibility we felt it critical to layout each step of the processing workflow, so the work can in theory be entirely reproduced. We attempted to be as concise as possible in each subsection of section 3, moving as much supplementary information as possible to be the appendix where appropriate, however the amount of detailed thought we put into creating each stage of the workflow resulted in a large number of subsections. Despite this, we feel section 3 importantly emphasises the scientific rigour employed in creating the dataset, and can only serve to enhance confidence in the quality of the dataset as a result.

---

## Author Comment (AC2)

We greatly thank the reviewer for their feedback on our manuscript. Please find our response here to their comments.

Dene Bowdalo

**Reviewer 2**

This is an excellent paper and an important contribution to our field. Atmospheric composition observations are important to society and are needed to quantity trends and current levels of pollutants in the atmosphere, to evaluate models, to assimilate with models to provide optimal estimates of the state of the atmosphere, among others. But atmospheric composition measurements occur in many different and disparate locations, and it is often very difficult to discover and access the data. The approach developed and described in this paper goes a long way to improving the discovery and use of atmospheric composition data. It is a major development, and the authors are to be congratulated.
The paper is well written, and the relevant information is made available, and methods described in appropriate detail.

**1.1.** My only comment is related to how can this workflow be made even more useful. Specifically, much of the atmospheric composition data sits in individual measurement sites and data available and described in publications. How easy is it for a single measurement site to make their data available to GHOST? Can they provide the DOI and meta data etc. in an easy way?

> Our goal with this work was to process data from the major public reporting networks. In main this decision was taken due to the lack of homogeneity in reporting formats from across individual measurement stations. Parsing 38 networks was itself an extremely onerous task, having to do so for even more formats we felt was too much of a challenge. With that said, we would naturally love to incorporate as much data as we can in the dataset. With hope, if this project is welcomed and supported by the community and can gain more support, this would allow us to potentially look at creating standard templates for data providers to contribute data.